

# Description of the MIROC-ES2L Earth system model and evaluation of its climate–biogeochemical processes and feedbacks

Tomohiro Hajima[1], Michio Watanabe[1], Akitomo Yamamoto[1], Hiroaki Tatebe[1], Maki A. Noguchi[1], Manabu Abe[1], Rumi Ohgaito[1], Akinori Ito[1], Dai Yamazaki[2], Hideki Okajima[1], Akihiko Ito[3,1], Kumiko Takata[3], Koji Ogochi[1], Shingo Watanabe[1], Michio Kawamiya[1]

[1]Research Center for Environmental Modeling and Application, Japan Agency for Marine-Earth Science and Technology, 3173-25 Showamachi, Kanazawaku, Yokohama, Kanagawa 236-0001, Japan

[2]Research Center for Advanced Science and Technology, The University of Tokyo, Tokyo, 153-8505, Japan

[3]National Institute for Environmental Studies, Tsukuba, 305-8506, Japan

*Correspondence to*: Tomohiro Hajima (hajima@jamstec.go.jp)

Abstract

This study developed a new Model for Interdisciplinary Research on Climate, Earth System version2 for Long-term simulations (MIROC-ES2L) Earth system model (ESM) using a state-of-the-art climate model as the physical core. This 20    model embeds a terrestrial biogeochemical component with explicit carbon–nitrogen interaction to account for soil nutrient control on plant growth and the land carbon sink. The model's ocean biogeochemical component is largely updated to simulate biogeochemical cycles of carbon, nitrogen, phosphorus, iron, and oxygen such that oceanic primary productivity can be controlled by multiple nutrient limitations. The ocean nitrogen cycle is coupled with the land component via river discharge processes, and external inputs of iron from pyrogenic and lithogenic sources are considered. 25    Comparison of a historical simulation with observation studies showed the model could reproduce reasonable historical changes in climate, the carbon cycle, and other biogeochemical variables together with reasonable spatial patterns of distribution of the present-day condition. The model demonstrated historical human perturbation of the nitrogen cycle through land use and agriculture, and it simulated the resultant impact on the terrestrial carbon cycle. Sensitivity analyses in preindustrial conditions revealed modeled ocean biogeochemistry could be changed regionally (but substantially) by 30    nutrient inputs from the atmosphere and rivers. Through an idealized experiment of a 1%$CO_2$ increase scenario, we found the transient climate response (TCR) in the model is 1.5 K, i.e., approximately 70% that of our previous model. The cumulative airborne fraction (AF) is also reduced by 15% because of the intensified land carbon sink, resulting in an AF close to the multimodel mean of the Coupled Model Intercomparison Project Phase 5 (CMIP5) ESMs. The transient climate response to cumulative carbon emission (TCRE) is 1.3 K EgC$^{-1}$, i.e., slightly smaller than the average of the





CMIP5 ESMs, suggesting "optimistic" model performance in future climate projections. This model and the simulation
results are contributing to the Coupled Model Intercomparison Project Phase 6 (CMIP6). The ESM could help further
understanding of climate–biogeochemical interaction mechanisms, projections of future environmental changes, and
exploration of our future options regarding sustainable development by evolving the processes of climate,
biogeochemistry, and human activities in a holistic and interactive manner.

## 1. Introduction

Originally, global climate projections using climate models were based on simulations by atmosphere-only physical
models (Manabe et al., 1965). Numerical climate models evolved through the integration or improvement of component
models on ocean circulation (Manabe and Bryan, 1969), land hydrological processes (Sellers et al., 1986), sea ice

dynamics (e.g., Meehl and Washington, 1995), and aerosols (e.g., Takemura et al., 2000), most of which focus on physical
aspects that affect how climate is formed. Cox et al. (2000) attempted to couple a carbon cycle model and a climate model
to investigate the roles of biophysical and biogeochemical (carbon cycle) feedbacks on climate. Their results showing that
such interactions are significant in projecting future climate aroused interest in the processes and feedbacks beyond those
incorporated in traditional climate models. Models that incorporate biogeochemical processes, such as the one by Cox et al.

(2000), are often called Earth system models (ESMs). Currently, the most comprehensive state-of-the-art ESMs include
component models of the land and ocean carbon cycle, atmospheric chemistry, dynamic vegetation, and other
biogeochemical cycles (e.g., Watanabe et al., 2011, Collins et al., 2011).

Among many processes and possible interactions in the Earth system, the carbon cycle and its feedback on climate
remain the focus of simulation studies using ESMs because of the importance of anthropogenic $CO_2$ as the primary driver

for climate change and the complexity of the natural carbon cycle that determines its fate. As ESMs simulate explicit
climate–carbon interactions, ESMs can simulate the temporal evolution of atmospheric $CO_2$ concentration and the
resultant climate change using anthropogenic $CO_2$ emissions as an input (Friedlingstein et al., 2006, 2014). It is also
possible to make climate projections using prescribed $CO_2$ concentrations, and the diagnosed $CO_2$ fluxes in the
simulations can be used for calculating the level of anthropogenic $CO_2$ emissions compatible with prescribed $CO_2$

pathways (Jones et al., 2013). Furthermore, simulation results can be diagnosed in terms of the relationship between
anthropogenic $CO_2$ emissions and global temperature rise, i.e., the so-called transient climate response to cumulative
carbon emissions (TCRE) (Allen et al., 2009; Matthews et al., 2009). The ESMs of the Coupled Model Intercomparison
Project Phase 5 (CMIP5) revealed that the relationship is approximately linear (Gillett et al., 2013), which facilitates
estimation of the total amount of anthropogenic $CO_2$ emissions to restrict global warming to below a specific mitigation

target.

The feedback of the carbon cycle on climate is manifest through regulation of the atmospheric $CO_2$ concentration,
which can be decomposed into two feedback processes. The first process is the carbon cycle response to $CO_2$ increase.
Elevated $CO_2$ concentration accelerates vegetation growth that intensifies the land carbon sink. In addition, increased
levels of atmospheric $CO_2$ accelerate $CO_2$ dissolution into the surface water of the ocean, which eventually propagates into

the deeper ocean via ocean circulation and biological processes. Consequently, an increase of atmospheric $CO_2$ triggered
by external forcing (e.g., anthropogenic emissions) can be mitigated partly by natural $CO_2$ uptake, forming a negative
feedback loop between atmospheric $CO_2$ concentration and natural carbon uptake, i.e., the so-called $CO_2$–carbon feedback
(Gregory et al., 2009) or carbon–concentration feedback (Boer and Arora et al., 2009). The second feedback process is the





carbon cycle response to global warming. Global warming induces loss of carbon from the land to the atmosphere by
accelerating ecosystem respiration (Arora et al., 2013; Todd-Brown et al., 2014; Friedlingstein et al., 2014), while ocean
surface warming reduces the solubility of $CO_2$ in seawater. Intensification of upper-ocean stratification and weakening of
the biological pump by global warming also prevent effective transportation of dissolved carbon into the deeper ocean
(Frölicher et al., 2015; Yamamoto et al., 2018). Global warming might lead to localized intensification of the natural
carbon sink (e.g., lengthening of the growing season and exposure of the ocean surface through melting of sea ice).
However, state-of-the-art ESMs have projected global natural carbon loss due to warming, suggesting a positive feedback
loop between climate change and natural carbon uptake, i.e., the so-called climate–carbon feedback (Friedlingstein et al.,
2006; Arora et al., 2013).

Quantification of the strength of the carbon cycle feedbacks and their comparison among ESMs were first made by
Friedlingstein et al. (2006), who showed that all ESMs agreed with the positive sign of the climate–carbon feedbacks for
both land and ocean. The latest comparison using CMIP5 ESMs was made by Arora et al. (2013). They found that the
widest spread between the models was in the land carbon response to $CO_2$ increase, while the second greatest spread was
in the land carbon response to warming. Two of the ESMs in their analysis employed explicit carbon–nitrogen (C–N)
interactions in the land component for considering the limitation of soil N on land $CO_2$ uptake, and these two models
showed the smallest land carbon response to $CO_2$ increase. Although it was pointed out later that the lowest response of
the two C–N models was not necessarily induced by N limitation (Hajima et al., 2014b), the comparison study by Arora et
al. (2013) aroused interest in terrestrial biogeochemical feedbacks other than the carbon cycle. The importance of N
limitation on the land carbon sink has also been suggested following simulation studies using offline land models (e.g.,
Thornton et al., 2007; Sokolov et al., 2008; Zaehle and Friend, 2010) and diagnostic analyses using the simulation output
of ESMs (e.g. Wieder et al., 2015).

Compared with land, the oceans showed better agreement among the CMIP5 ESMs (Arora et al., 2013) in terms of the
strength of both $CO_2$–carbon and climate–carbon feedbacks. However, the ESMs showed substantial discrepancies in the
spatiotemporal patterns of ocean $CO_2$ uptake, even in historical simulations. In particular, in the Southern Ocean, although
the models indicated dominance of the region in relation to anthropogenic carbon uptake (Frölicher et al., 2015), the
seasonality of the atmosphere–ocean $CO_2$ flux and the cumulative values in that region showed divergent patterns among
the models (Anav et al., 2013; Frölicher et al., 2015).

The ecological response of the ocean in ESMs remains far from certain. A benchmark study by Anav et al. (2013)
revealed that all CMIP5 ESMs underestimate net primary productivity (NPP) in the high latitudes of the Northern
Hemisphere, where seawater temperature and N availability likely limit primary production (e.g., Moore et al., 2013).
They also found that most models overestimate NPP in the Southern Hemisphere high latitudes, where nutrient supply is
sufficient because of strong upwelling but iron supply is limited (Moore et al., 2013). Globally, the CMIP5 ESMs simulate
NPP with different magnitudes, even in the preindustrial condition, and the global NPP response among the models to past
and future climate change is largely divergent (Laufkötter et al., 2015), as is the sinking particle flux (Fu et al., 2016).
Although such problems regarding oceanic NPP might be attributable partly to inaccurate reproduction of oceanic physical
fields by the models (Frölicher et al., 2015; Laufkötter et al., 2015), it is critical in simulations to reproduce accurately the
relative abundances of nutrients in the euphotic zone and their availability to microorganisms. In particular, nutrients in
the upper ocean are sustained by upwelling from the deeper ocean and nutrient inputs from external sources. Some studies
suggest that nutrient availability to marine ecosystems could decline in the future through reduction of nutrient upwelling
because of intensified stratification (e.g., Ono et al., 2008; Whitney et al., 2013; Yasunaka et al., 2016). Conversely, other



studies suggest that nutrient supply through atmospheric deposition and river discharge processes could be amplified in
the future because of human activities (Gruber and Galloway, 2008; Mahowald et al., 2009) unless robust mitigation
policies are adopted. Thus, to project the effects of biogeochemical feedback on climate, it is necessary to consider the
response of ecological processes to changing nutrient inputs as well as the response to climate change.

On the basis of the above, we previously reviewed the CMIP5 exercises and we discussed the perspective for new ESM
development (Hajima et al., 2014a). In our ESM development, we prioritized the incorporation of explicit C–N interaction
in the land biogeochemical component. The terrestrial nitrogen cycle regulates the carbon cycle by modulating soil
nutrient availability to plants, regulating leaf N concentration and photosynthetic capacity, and changing the C:N ratio in
plants and soils. In particular, $CO_2$ stimulation of plant growth (the so-called $CO_2$ fertilization effect) is the main driver of
terrestrial $CO_2$–carbon feedback, while N limitation on plant growth might regulate the feedback strength (Arora et al.,
2013; Hajima et al., 2014a; Hajima et al., 2014b). Thus, consideration of C–N coupling in the terrestrial ecosystem in an
ESM will enable change in the capacity of the land carbon sink following a change of N dynamics induced by human
perturbation (e.g., fertilizers) and/or atmospheric N deposition.

For the ocean, the biogeochemical component in our previous model (the MIROC-ESM; Watanabe et al., 2011) was
unchanged from that used for the first stage of the Coupled Climate Carbon Cycle Model Intercomparison Project
(C4MIP: Friedlingstein et al., 2006; Yoshikawa et al., 2008). The ocean component simulated C and N cycles only, using
simple parameterizations of ocean ecosystem dynamics with four types of N tracer and five C tracers (Watanabe et al.,
2011) with fixed C:N ratios of the organic components. Furthermore, the ocean N cycle in the model was closed to other
subsystems, i.e., there was no N input into the ocean (e.g., biological N fixation, atmospheric N deposition, and riverine N
input) or flux out of the system (e.g., outgassing and sedimentation). To account for changing inputs of N nutrients into
the ocean in the simulations, we gave second priority to the coupling of the ocean N cycle to other subsystems by
incorporating N exchange processes between the ocean and other components in the new ESM. The ocean N fixer (i.e.,
diazotrophs) can be regulated strongly by P availability (Shinozaki et al., 2018); therefore, inclusion of the ocean P cycle
should be adopted together with improvement of the N cycle. In addition, as the denitrification process is regulated
strongly by the level of oxygen in seawater, it was also decided to include the oxygen cycle in the new model. Inclusion of
the oxygen cycle provides potential to project future oceanic deoxygenation that is likely to threaten the habitable zone of
marine ecosystems, driven by changes in oxygen solubility, mixing, circulation, and respiration due to global warming
(Oschlies et al., 2018; Yamamoto et al., 2015).

The third priority in developing a new ESM was incorporation of Fe cycle processes. Fe is an essential micronutrient
for phytoplankton. Thus, any model lacking consideration of the Fe cycle potentially overestimates primary productivity,
especially in regions where subsurface macronutrient supply is enhanced but Fe availability is limited, e.g., the main
oceanic upwelling "high-nutrient, low-chlorophyll" regions (Martin and Gordon, 1988; Moore et al., 2013). Similar to the
N cycle, the ocean Fe cycle is also an open system. One of its main external sources is dissolved Fe from continental
margins and from hydrothermal vents along mid-ocean ridges (Tagliabue et al., 2017). Thus, continental and hydrothermal
Fe supply is important in terms of determining the background Fe concentration in seawater. In addition, the ocean Fe
cycle is also connected to the land through the atmosphere (Jickells et al., 2005; Mahowald et al., 2009; Ito et al., 2019).
Fe-containing aerosols are emitted from dry land surfaces, open biomass burning, and fossil fuel combustion, and they are
delivered to marine ecosystems via dry and wet deposition processes. These processes have been perturbed by climate
change, land use change (LUC), and air pollution (Jickells et al., 2005; Mahowald et al., 2009; Ito et al., 2019). Thus,





consideration of atmospheric Fe deposition in particular is necessary to reflect the anthropogenic impact on future marine ecosystem dynamics via Fe cycle processes.

Here, we present a description of a new ESM, the Model for Interdisciplinary Research on Climate, Earth System version2 for Long-term simulations (MIROC-ESL2), which considers explicit carbon and nitrogen cycles for land, and carbon, nitrogen, iron, phosphate, and oxygen cycles for the ocean. In the model, the biogeochemical components are coupled interactively with physical climate components, enabling consideration of climate–biogeochemical feedbacks. The model description and experimental settings are presented in Sect. 2. The basic performance of the model, evaluated

by executing a historical simulation and comparison of the results with observation-based studies, is presented in Sect. 3.1. To evaluate the sensitivity of the biogeochemical processes, experiments for sensitivity analysis were performed and the results compared with existent studies. In particular, global temperature response to cumulative anthropogenic $CO_2$ emissions in the new model was quantified and compared with that of the CMIP5 ESMs, to characterize the general features of the new model in relation to existing ESMs. The results of the sensitivity analyses are presented in Sect. 3.2.

Finally, a summary and perspectives obtained from this study are summarized in Sect. 4.

## 2. Methods

### 2.1. Model configurations

To describe comprehensively the MIROC-ES2L structure (Fig. 1), we first present the physical core of MIROC5.2,

which is an updated version of MIROC5 used for the CMIP5 exercises. Only a brief summary is presented here because a detailed description on the modeling of MIROC5 can be found in Watanabe et al. (2010) and an account of a simulation study performed by MIROC5.2 can be found in Tatebe et al. (2018). In addition, a description of MIROC6, which shares almost the same structure and many of the characteristics of MIROC5.2, except for the atmospheric spatial resolution and cumulus treatments, can be found in Tatebe et al. (2019). In this paper, description of the land and ocean biogeochemistry

is presented in detail because those two components represent the main modifications from the previous version of the ESM (i.e., the MIROC-ESM; Watanabe et al., 2011).

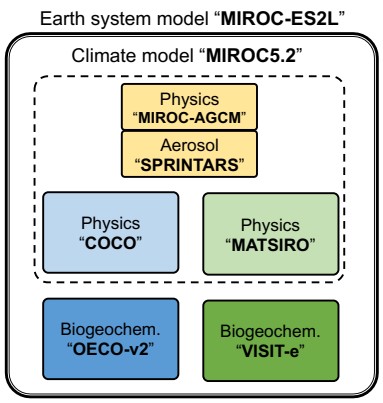

**Figure 1**





Schematic of component models in the new MIROC-ES2L Earth system model. The physical core of the model is
MIROC5.2, which consists of an atmospheric climate model (CCSR-NIES AGCM or MIROC-AGCM) with an aerosol
module (SPRINTARS), an ocean physics model (COCO) with a sea ice model, and a land physics model (MATSIRO)
with a river routine. The land biogeochemistry component (VISIT-e) simulates carbon and nitrogen cycles with an LUC
routine, and the ocean biogeochemistry component (OECO) simulates the cycles of carbon, nitrogen, iron, phosphorus,
and oxygen.

### 2.1.1.    Physical core

The MIROC5.2 physical core comprises component models of the atmosphere, ocean, and land. The atmospheric
model is based on a spectral dynamical core, originally named the Center for Climate System Research–National Institute
for Environmental Studies atmospheric general circulation model (CCSR-NIES AGCM; Numaguchi et al., 1997), which is
coupled interactively with an aerosol component model called the Spectral Radiation-Transport Model for Aerosol
Species (SPRINTARS; Takemura et al., 2000, 2005). For the ocean, the CCSR Ocean Component (COCO) model
(Hasumi, 2006) is used in conjunction with a sea ice component model. For land, the Minimal Advanced Treatments of
Surface Interaction and Runoff (MATSIRO) model (Takata et al., 2003) is coupled to simulate the atmosphere–land
boundary conditions and freshwater input into the ocean. Considering the application possibility of the ESM to long-term
climate simulations of more than hundreds of years, e.g., paleoclimate studies (Ohgaito et al., 2013; Yamamoto et al.,
2019), the horizontal resolution of the atmosphere is set to have T42 spectral truncation, which is approximately 2.8°
intervals for latitude and longitude. The vertical resolution is 40 layers up to 3 hPa with a hybrid σ–p coordinate, as in
MIROC5. The horizontal coordination for the ocean is changed from the bipolar system employed in MIROC5 to a
tripolar system in MIROC5.2. The vertical levels are increased from 44 to 62 with a hybrid σ–z coordinate system. For
land, the same horizontal resolution as used for the atmosphere is employed, and the vertical soil structure of the model
has six layers down to the depth of 14 m. Subgrid fractions for two land use types (agriculture plus managed pasture and
others) are considered for the physical processes.

For the AGCM, the schemes used for the dynamical core, radiation, cumulus convection, and cloud microphysics are
mostly the same as in MIROC5; the major update of processes mainly concerns the aerosol module. The version used here
treats atmospheric organic matter (OM) as one of the prognostic variables, and emission of primary OM and precursors for
secondary OM are diagnosed in the component. For land, the scheme for subgrid snow distribution is replaced by one
incorporating a physically based approach (Nitta et al., 2014; Tatebe et al., 2019), and snow-derived wetland is newly
considered to reduce the hot bias found in the European region during spring–summer (Nitta et al., 2017; Tatebe et al.,
2019). The ocean and sea ice components are mostly the same as in MIROC5.

### 2.1.2.    Land biogeochemical processes

The model of the land ecosystem/biogeochemistry component in MIROC-ES2L is the Vegetation Integrative SImulator
for Trace gases model (VISIT; Ito and Inatomi, 2012a). This model simulates carbon and nitrogen dynamics on land
(schematics can be found in Ito and Oikawa 2002 for the carbon cycle and Supplementary Fig. 1 for the nitrogen cycle). It



has been used for ecological studies of site–global scale (e.g., Ito and Inatomi, 2012b), impact assessments of climate change (e.g., Warszawski et al., 2013; Ito et al., 2016), prior to $CO_2$ flux inversion studies (e.g., Maksyutov et al., 2013;

Niwa et al., 2017), and contemporary assessments of $CO_2$, $CH_4$, and $N_2O$ emissions in the Global Carbon Projects (Le Quéré et al., 2016; Saunois et al., 2016; Tian et al., 2018). The early version of the model (Sim-CYCLE; Ito and Oikawa, 2002) was actually used as the land carbon cycle component in the first stage of the C4MIP project (Friedlingstein et al., 2006; Yoshikawa et al., 2008). The model covers major processes relevant to the global carbon cycle. Photosynthesis or gross primary productivity (GPP) is simulated based on the Monsi–Saeki theory (Monsi and Saeki, 1953), which provides

a conventional scheme to simulate leaf-level photosynthesis in a semiempirical manner and for upscaling to canopy-level primary productivity. The allocation of photosynthate between carbon pools in vegetation (e.g., leaf, stem, and root) is regulated dynamically following phenological stages. Transfer of vegetation carbon into litter/soil pools is simulated using constant turnover rates and, in deciduous forests, seasonal leaf shedding occurs at the end of the growing period. The model focuses on biogeochemical processes and it does not explicitly simulate dynamic change in vegetation composition;

therefore, these processes are simulated using a static biome distribution. The carbon stored in litter (i.e., foliage, stem, and root litter) and humus (i.e., active, slow, and passive) pools is decomposed and released as $CO_2$ to the atmosphere, under the influence of soil water and temperature. Further details on the carbon cycle processes in the model can be found in Ito and Oikawa (2002).

For the nitrogen cycle, the model considers two major nitrogen influxes into the ecosystem: biological nitrogen fixation

(BNF) simulated based on the scheme of Cleveland et al. (1999) and external nitrogen sources such as fertilizer and atmospheric nitrogen deposition, which are prescribed in forcing data. The fluxes of nitrogen out of the land ecosystem are $N_2$ and $N_2O$ production processed by nitrification and denitrification in soils, based on the scheme of Parton et al. (1996), leaching of inorganic nitrogen from soils, which is affected by the amount of soil nitrate and runoff rate, and $NH_3$ volatilization from soils (Lin et al., 2000; Thornley, 1998). Within the vegetation–soil system, organic nitrogen in the soil

is supplied from litter fall, whereas inorganic nitrogen is released through soil decomposition processes (soil mineralization) and stored as two chemical forms ($NO_3^-$ and $NH_4^+$). Inorganic nitrogen is taken up by plants, allocated to two vegetation pools (canopy and structural pools), and immobilized into a microbe pool. Finally, mineral nitrogen is lost via biotic/abiotic processes as mentioned above.

Although the original land component model covers most major carbon/nitrogen processes, for the purposes of

inclusion in the new ESM and making fully coupled climate–carbon/nitrogen projections, the land model was modified for this study (hereafter, the modified version is called VISIT-e). Thus, the modified model represents the tight interaction between carbon and nitrogen in plants. This is because the original model has only loose interaction between these two cycles and thus it cannot predict precisely the nitrogen limitation on primary productivity. To achieve this, the photosynthetic capacity in VISIT-e is modified to be controlled by the amount of nitrogen in leaves (leaf nitrogen

concentration), which is determined by the balance between the nitrogen demand of plants and potential supply from the soil. Thus, if sufficient inorganic nitrogen is not available for plants, leaf nitrogen concentration is gradually lowered, leading to the reductions of photosynthetic capacity and the plant production rate. This process is required to simulate the observed down-regulation in elevated $CO_2$ experiments (e.g., Norby et al., 2010; Zaehle et al., 2014). Other modifications regarding the nitrogen cycle are described in Appendix A.

Although the original VISIT incorporates LUC and associated $CO_2$ emission processes, to take full advantage of the latest LUC forcing dataset (Land-use harmonization 2; Ma et al., 2019), additional LUC-related processes have been newly introduced in VISIT-e. The model assumes five types of tile in each land grid (i.e., primary vegetation, secondary



vegetation, urban, cropland, and pasture) with the same structure of carbon/nitrogen pools. All processes are calculated separately for each tile (i.e., no lateral interaction), and then the variables in the tile are summed after weighting by the

areal fraction of each land use type. The LUC impact is modeled assuming two types of land use impact on the biogeochemistry. The first impact considers status-driven LUC processes, which affect land biogeochemistry even when the areal fractions of the tiles are fixed. For example, even when a simulation is conducted with fixed areal fractions (e.g., a spin-up run under 1850 conditions), crop harvesting, nitrogen fixation by N-fixing crops, and the decay of OM in product pools occur. The second type of land use impact includes transition-driven processes that happen only when areal

changes occur among the tiles. For example, when an areal fraction is changed within a year (e.g., conversion of forest to urban land use), carbon and nitrogen in the harvested biomass are translocated between product pools. When cropland is abandoned and the area is reclassified as secondary forest, the apparent mean mass density of secondary forest is first diluted because of the increase in less-vegetated area, and then secondary forest starts regrowth toward a new stabilization state. Further detailed description on LUC modeling is given in Appendix A.

The land ecosystem component runs with a daily time step in the ESM. It has fixed spatial distribution patterns of 12 vegetation categories (see Supplementary Fig. 2), and the land biogeochemistry is affected by daily averaged atmospheric conditions ($CO_2$ concentration, downward shortwave radiation, air temperature, and air pressure) and land abiotic conditions (soil water, soil temperature, and runoff rate as the base flow) simulated by the physical core of the ESM. In turn, daily averaged land variables simulated by VISIT-e are used by other components of the ESM. For example, the

simulated leaf area index (LAI) is referenced in the physical core of the model to simulate physical dynamics on the land surface (e.g., evapotranspiration, albedo, and surface roughness). Furthermore, the rate of net atmosphere–land $CO_2$ fluxes is used in the calculation of the atmospheric $CO_2$ concentration, and inorganic N leached from the soil is transported rivers and subsequently used as an input of N nutrients to the ocean ecosystem. We note that the chemical state of N in rivers is assumed conserved during transportation, and that biogeochemical processes such as outgassing or sedimentation in

freshwater systems are neglected in the present model. In addition, although the model can simulate terrestrial carbon loss by erosion and the dissolution of organic carbon in the ocean, to close the global mass conservation of carbon and nitrogen in simulations, the processes are forced to be inactive in this study. Finally, although $N_2O$ and $NH_3$ emissions are simulated, the emission fluxes are simply diagnosed and they do not produce any changes in the atmospheric radiation balance or air quality.


### 2.1.3. Ocean biogeochemical processes

The new ocean biogeochemical component model OECO2 (see Supplementary Fig. 3 for a schematic), is a nutrient–phytoplankton–zooplankton–detritus-type model that is an extension of the previous model (Watanabe et al., 2011). Although only an overview of OECO2 is presented here, a detailed description can be found in Appendix B.

In OECO2, ocean biogeochemical dynamics are simulated based on 13 types of oceanic tracer. Three of them are associated with cycles of macronutrients (nitrate and phosphate) and a micronutrient (dissolved Fe). The model has four organic tracers of "ordinary" nondiazotrophic phytoplankton, diazotrophic phytoplankton (nitrogen fixer), zooplankton, and particulate detritus. All OM in these four tracers is assumed to have identical nutrient, oxygen, and micronutrient iron composition following the Redfield ratio concept with a constant elemental stoichiometric ratio for carbon, i.e., C:N:P:O =

106:16:1:138 (Takahashi et al., 1985) and C:Fe = $150:10^{-3}$ (Gregg et al., 2003). Four other types of tracer are associated





with carbon and/or calcium, i.e., dissolved inorganic carbon (DIC), total alkalinity, calcium, and calcium carbonate. The two other tracers are oxygen and nitrous oxide.

The nitrogen cycle processes in OECO2 are similar to those in the previous version (Yoshikawa et al., 2008; Watanabe et al., 2011), except the new model accounts for nitrogen influxes such as nitrogen deposition from the atmosphere (as
external forcing), input of inorganic nitrogen from land via rivers, and BNF by diazotrophic phytoplankton. In addition, denitrification processes are also modeled as the dominant process of oceanic nitrogen loss, with explicit distinction between the gaseous forms of $N_2O$ and $N_2$ (see below for nitrogen fixation and denitrification processes). Loss of nitrogen through the sedimentation process is also considered. The phosphorus cycle is newly embedded in the model to represent strong phosphorous limitation on the growth of diazotrophic phytoplankton. The structure of the phosphorus cycle is
generally similar to that of nitrogen, except the riverine input of phosphate is the only process that introduces phosphorus into the ocean. As the land ecosystem model cannot simulate the phosphorus cycle, the flux of phosphorous from rivers is diagnosed from the nitrogen flux, assuming that the phosphate brought to the river mouth satisfies the N:P ratio of 16:1, similar to the Redfield ratio.

The structure of the ocean iron cycle is also similar to that of nitrogen, except the following processes are modeled as
iron input into the ocean. Two major sources of iron deposition from the atmosphere are included in the new model: lithogenic and pyrogenic sources. Mineral dust emission is diagnosed by the aerosol component module, depending on the near-surface wind speed, soil dryness, and bare ground cover, while iron emitted from biomass burning and the consumption of fossil fuel and biofuel follows external forcing. The latter emission dataset used in this study is shown in Supplementary Fig. 4. The iron emissions from pyrogenic sources are estimated based on the iron content and emissions
of particulate matter (Ito et al., 2018). A shift from coal to oil combustion is considered in relation to shipping (Fletcher, 1997; Endresen et al., 2007). The iron content of mineral dust is prescribed at 3.5% (Duce and Tindale, 1991). The iron deposition from biomass burning is calculated from black carbon (BC) deposition and a ratio of 0.04 gFe gBC$^{-1}$ in fine particles at emission (Ito 2011). The emission, transportation, and deposition processes are simulated explicitly by the atmospheric aerosol component. The iron from different sources has different solubility in seawater and thus different
amounts of iron are available for phytoplankton. The solubility of iron is prescribed at 79% for oil combustion, 11% for coal combustion, and 18% for biomass burning (Ito, 2013). The solubility of iron for mineral dust is prescribed at 2% (Jickells et al., 2005).

In addition to the Fe input from the atmosphere, recent studies suggest contributions of Fe supply from sediment and hydrothermal vents to ecosystem activities (Tagliabue et al., 2017). The contributions of these two natural Fe sources to
the determination of atmospheric $CO_2$ concentration and export production are similar to or greater than that of dust (Tagliabue et al., 2014). Therefore, these three Fe sources are also considered in the new ESM (Appendix B).

Ocean ecosystem dynamics are simulated based on the nutrient cycles of nitrate, phosphorous, and iron. Nutrient concentration, in conjunction with the controls of seawater temperature and availability of light, regulates the primary productivity of the two types of phytoplankton. The model assumes that diazotrophic phytoplankton can prosper in regions
where phosphate is available but nitrate concentration is small (<0.05 μmol L$^{-1}$). In the model, zooplankton is assumed independent of abiotic conditions (e.g., seawater temperature) and dependent on biotic conditions (phytoplankton and zooplankton concentrations), as in the previous model. The denitrification process is modeled to occur only in suboxic waters (<5 μmol L$^{-1}$) (Schmittner et al., 2008) and it is suppressed in water with low nitrate concentration (<1 μmol L$^{-1}$). Detritus contains nitrate, phosphorus, iron, oxygen, and carbon, most of which is remineralized while sinking downward.





The detritus that reaches the ocean floor is removed from the system; however, a fraction of OM in the sediment is assumed to return to the bottom layer of the water column at a constant rate in each location (Kobayashi and Oka, 2018).

The ocean carbon cycle is formed by atmosphere–ocean $CO_2$ exchange, inorganic carbon chemistry, OM dynamics driven by marine ecosystem activities, and transportation and reallocation processes of ocean carbon within the interior. The formulations of atmosphere–ocean gas exchange, carbon chemistry, and related parameters follow protocols from the

Ocean Model Intercomparison Project (OMIP; Orr et al., 2017). Production of DIC and total alkalinity is controlled by changes in inorganic nutrients and $CaCO_3$, following Keller et al. (2012).

Finally, the flux of dimethyl sulfide (DMS) from the ocean, which is produced by plankton and is a precursor of atmospheric sulfate aerosols, is diagnosed in the original aerosol module from the surface downward shortwave radiation flux. In MIROC-ES2L, this emission scheme is modified and the flux is calculated from the sea surface DMS

concentration that is diagnosed from the simulated surface water chlorophyll concentrations and the corresponding mixed-layer depth (Appendix B). In the present model, this is the only pathway via which ocean biogeochemistry affects climate, except for atmosphere–ocean $CO_2$ exchange.

### 2.2. Experiments, forcing, and metrics

#### 2.2.1.  Experiments and forcing

To evaluate the performance and sensitivities of MIROC-ES2L, we conducted four groups of experiments comprising eleven experiments in total (Tables 1 and 2). The first group was a control run that consisted of two types of experiment: a normal control run (CTL) in which the external forcing was set to preindustrial conditions and an alternative control run (CTL-D) used for sensitivity analysis of the ocean biogeochemistry, which is described later.

The second group, used for historical simulations, consisted of three types of experiment during the period 1850–2014. All three experiments were driven by the Coupled Model Intercomparison Project Phase 6 (Eyring et al., 2016) official forcing datasets (version 6.2.1; details on forcing datasets used in the simulations are summarized in Appendix C), and the $CO_2$ concentration was prescribed in the simulations (i.e., so-called concentration-driven experiments). The first comprised a conventional historical simulation (HIST), and the simulation result is used for direct comparison with

observation-based studies to evaluate model performance. The second was a special experiment named HIST-NOLUC, designed for evaluation of the impact of LUC on the climate and biogeochemistry. In this experiment, land use and agricultural management (fertilizer application) were fixed at preindustrial levels. This experimental configuration is the same as the LUMIP experiment in CMIP6 named land-noLu (Lawrence et al., 2016). The third experiment (HIST-BGC) was the same as HIST, except only carbon cycle processes detect the $CO_2$ increase (named in C4MIP of CMIP6 as hist-

bgc; Jones et al., 2016). Thus, there was no $CO_2$-induced global warming in the experiment.

The third experimental group was used to evaluate the climate and carbon cycle feedbacks. This group comprised three types of idealized experiment, following experimental designs proposed by Eyring et al. (2016) and Jones et al. (2016). In the three experiments, $CO_2$ concentration was prescribed to increase at the rate of 1.0% per year from the preindustrial state throughout the 140-year period (i.e., the concentration finally reached a value of approximately 1140 ppmv), while

other external forcing was maintained at the preindustrial condition. The three experiments were configured as follows: (1) 1PPY: a normal experiment in which both climate and biogeochemical processes respond to the $CO_2$ increase; (2) 1PPY-BGC: the same as 1PPY but only carbon cycle processes detect the $CO_2$ increase; and (3) 1PPY-RAD: the same as 1PPY





but only atmospheric radiation processes detect the $CO_2$ increase. In 1PPY-BGC, carbon cycle processes respond to the $CO_2$ increase without $CO_2$-induced global warming; thus, the result of this simulation is used to quantify $CO_2$–carbon
feedback. In 1PPY-RAD, as there is no direct $CO_2$ stimulation on the carbon cycle, climate change is the only cause of carbon cycle variation. Thus, this simulation result is utilized for evaluating climate–carbon feedback.

The final group comprised a set of experiments for evaluating ocean biogeochemistry, focusing mainly on the processes newly introduced in MIROC-ES2L. This group consisted of three types of experiment. The first experiment (NO-NR) was configured similarly to the CTL run, except the ocean component did not receive any riverine N input. Through this
experiment, the impact of riverine N on ocean biogeochemistry could be evaluated. The second experiment (NO-NRD) was the same as NO-NR, except atmospheric N deposition additionally had no effect on ocean biogeochemistry. By evaluating the difference between NO-NR and NO-NRD, the impact of nitrogen deposition on ocean biogeochemistry could be evaluated. The final experiment (NO-FD) was configured with atmospheric Fe deposition onto the ocean surface switched off. To detect slight signals of ocean biogeochemistry arising from switching off the three processes (i.e.,
riverine N, N deposition, and Fe deposition), it was necessary to maintain consistency in the ocean physical fields between the experiments because a slight difference of the ocean physical fields produces perturbation on ocean biogeochemistry, which would be noise in the analyses. In MIROC-ES2L, ocean DMS emission is the feedback process of ocean biogeochemistry on the atmospheric physical processes; thus, biogeochemical change induced by the switching-off manipulations must change the DMS emission, leading to inconsistency in the physical fields between the experiments. To
avoid this occurrence, the DMS emission scheme in all three experiments was reverted to that used in the original aerosol component model, which is independent of the ocean ecosystem state (Appendix B). Similarly, the special control run (CTL-D), which was based on CTL, also had the DMS emission scheme changed to the same as NO-NR, NO-NRD, and NO-FD.

For conducting the experiments described above, preindustrial spin-up was performed in advance. Land and ocean
biogeochemical components were decoupled from the ESM, and the spin-up run was conducted for 3000 years for the ocean component and 30,000 years for land, by prescribing model-derived physical fields and other external forcing for the component models. In the final phase of the spin-up procedure, continuous spin-up, forced by the 1850-year condition of CMIP6 forcing, was performed for the entire system for 2483 years (Supplementary Fig. 5). All the experiments listed in Table 1 were initiated from the final condition of this spin-up procedure.


**Table 1**

Summary of details of the experiments.





| Experimental Group | Experiment | Purpose | Configurations | Duration [yrs] |
|---|---|---|---|---|
| Control | CTL | Control run | CO2 conc. and other forcings are fixed at pre-industrial level | 165 |
| | CTL-D | Control run for NO-NR, NO-NRD, and NO-FD | Same as CTL, but DMS emission follows the scheme of original aerosol module | 100 |
| Historical | HIST | Evaluation of model performance | Following CMIP6-DECK historical run | 165 (1850-2014) |
| | HIST-NOLUC | Evaluation of land-use change impact on carbon cycle | LUC and fertilizer are fixed at pre-industrial level | 165 (1850-2014) |
| | HIST-BGC | Evaluation of response of carbon cycle to CO2 increase | Same as HIST but only biogeochemical processes "see" the CO2 increase | 165 (1850-2014) |
| 1%CO2 | 1PPY | Evaluation of sensitivities of climate and carbon | Prescribed CO2 increased with 1.0 [% yr-1] | 140 |
| | 1PPY-BGC | Evaluation of response of carbon cycle to CO2 increase | Same as 1PPY but only biogeochemical processes "see" the CO2 increase | 140 |
| | 1PPY-RAD | Evaluation of response of carbon cycle to climate change | Same as 1PPY but only atmospheric radiative processes "see" the CO2 increase | 140 |
| OBGC | NO-NR | Evaluation of impacts of riverine N to ocean | Same as CTL-D but ocean doesn't get impact from riverine N | 100 |
| | NO-NRD | Evaluation of impacts of deposition N to ocean, by combining with NO-NR | Same as NO-NR but ocean doesn't get impact from Fe deposition | 100 |
| | NO-FD | Evaluation of impacts of deposition Fe to ocean | Same as CTL-D but ocean doesn't get impact from de eposition | 100 |

**Table 2**

Biogeochemical configurations in experiments, summarized as biogeochemical process settings. Bold characters represent the major differences between experiments within an experimental group.

| Experimental Group | Experiments | Impact on Land/Ocean BGC[*] | | | Impact on Ocean BGC[†] | | | DMS scheme[‡] |
|---|---|---|---|---|---|---|---|---|
| | | CO₂ | Climate | LUC | River N | Dep. N | Dep. Fe | |
| Control | CTL | – | – | – | O | O | O | **TypeA** |
| | CTL-D | – | – | – | O | O | O | **TypeB** |
| Historical | HIST | **O** | **O** | **O** | O | O | O | TypeA |
| | HIST-NOLUC | **O** | **O** | – | O | O | O | TypeA |
| | HIST-BGC | **O** | – | **O** | O | O | O | TypeA |
| 1%CO2 | 1PPY | **O** | **O** | – | O | O | O | TypeA |
| | 1PPY-BGC | **O** | **O** | – | O | O | O | TypeA |
| | 1PPY-RAD | **O** | – | – | O | O | O | TypeA |
| OBGC | NO-NR | – | – | – | – | **O** | **O** | **TypeB** |
| | NO-NRD | – | – | – | – | – | **O** | **TypeB** |
| | NO-FD | – | – | – | **O** | **O** | – | **TypeB** |

*If the biogeochemical process in an experiment was affected by $CO_2$, climate, or land use change, the letter "O" is

present; otherwise, the symbol "–" is used.





†If the ocean biogeochemistry process detected fluxes of riverine nitrogen, atmospheric nitrogen deposition, or atmospheric iron deposition, the letter "O" is present; otherwise, the symbol "–" is used.

‡The TypeA DMS emission scheme is the default scheme in MIROC-ES2L, where DMS emission is simulated as being dependent on the ocean biogeochemical status and the mixed-layer depth. TypeB is a scheme employed in the

original aerosol component model in which DMS emission is calculated as independent of ocean status.

### 2.2.2. Evaluation of climate and carbon cycle response to $CO_2$

To evaluate the climate and carbon cycle response to $CO_2$ increase, we used the metrics of transient climate response (TCR), airborne fraction of $CO_2$ (AF), and TCRE, which have been utilized previously to characterize the entire climate–

carbon cycle response to $CO_2$ increase in other models (Matthews et al., 2009; Hajima et al., 2012; Gillett et al., 2013). Similar analysis is made in this study and the result is presented in Sect. 3.2.

First, TCRE is defined as the ratio of global mean near-surface air temperature change (T) to cumulative anthropogenic carbon emission (CE) at the level of doubled $CO_2$ concentration from the preindustrial state (hereafter, $2xCO_2^{PI}$):

$$TCRE = T/CE, \tag{1}$$

which can be written as follows:

$$TCRE = (CA/CE) \times (T/CA), \tag{2}$$

where CA is the atmospheric carbon increase until reaching $2xCO_2^{PI}$. The first factor on the right-hand side (CA/CE) is identical to the definition of the cumulative airborne fraction of anthropogenic carbon emission:

$$CA/CE = AF. \tag{3}$$

The second factor (T/CA) can be represented by TCR as follows:

$$T/CA = TCR/CA, \tag{4}$$

which is because TCR is defined as T at $2xCO_2^{PI}$. Thus, Eq. 2 can be expressed as follows:

$$TCRE = AF \times (TCR/CA). \tag{5}$$

The result of the 1PPY simulation was used to evaluate TCRE, TCR, and AF. As CA is prescribed in the simulation,

CE can be diagnosed by CE = CA + CL + CO, where CL and CO represent the change in land and ocean carbon storage, respectively. As shown in Matthews et al. (2009), AF summarizes the carbon cycle response to anthropogenic CE. The second factor in Eq. 5 (TCR/CA) captures the global climate response to $CO_2$ increase in the models. TCRE quantifies the entire climate–carbon cycle response to anthropogenic $CO_2$ emission in the model.

To evaluate the strength of carbon cycle feedbacks in the model, the feedback strength is quantified by the so-called β

and γ quantities (Friedlingstein et al., 2006; Arora et al., 2013). The former is a feedback parameter for $CO_2$–carbon feedback (carbon cycle response to $CO_2$ increase), which can be calculated as follows:

$$\beta_L = (CL^{1PPY-BGC} - CL^{CTL})/CA^{1PPY}, \tag{6}$$

$$\beta_O = (CO^{1PPY-BGC} - CO^{CTL})/CA^{1PPY}, \tag{7}$$





where subscripts L and O represent land and ocean, respectively, and the superscripts represent the experiment used for the calculation.

The quantity γ is a feedback parameter for climate–carbon feedback (carbon cycle response to climate change), which can be calculated using the results of the 1PPY-RAD and CTL simulations:

$$\gamma_L = (CL^{1PPY\text{-}RAD} - CL^{CTL})/CA^{1PPY} , \tag{8}$$

$$\gamma_O = (CO^{1PPY\text{-}RAD} - CO^{CTL})/CA^{1PPY} . \tag{9}$$

## 3. Results and discussion

### 3.1. Model performance in historical simulation

#### 3.1.1. Global climate: net radiation balance and global temperature

To evaluate the physical fields reproduced by MIROC-ES2L, the temporal evolutions of the global mean net radiation balance at the top of atmosphere (TOA), near-surface air temperature (SAT), sea surface temperature (SST), and upper-ocean (0–700 m) temperature were compared with observation datasets and the results are shown in Fig. 2. The model simulates a reasonably steady state of net TOA radiation balance in the CTL run, showing a trend of $-4.6 \times 10^{-5}$ W m$^{-2}$ yr$^{-1}$ during the 165-year period. When comparing net TOA radiation balance of the HIST simulation with satellite measurement (CERES EBAF-TOA edition 4.0, constrained by in situ measurements; Loeb et al., 2012, 2018), the model result is $-0.63$ W m$^{-2}$ (negative means net incoming radiation) during 2001–2010, which is within the range of $-0.5 \pm 0.43$ W m$^{-2}$ estimated by Loeb et al. (2012) for the corresponding period (Fig. 2a).

Following the net increase of incoming radiation, the SAT anomaly becomes larger in the latter half of the 20th century (Fig. 2b). The warming trend during 1951–2011 is simulated as 0.1 K per decade, which is consistent with that of HadCRUT4 (version 4.6; Morice et al., 2012) of 0.11 K per decade (Stocker et al., 2013). Observation datasets of SST (HadSST version 3.1.1; Kennedy et al., 2011) and upper-ocean temperature (Levitus et al., 2012) clearly display increasing trends in the corresponding period, which are reproduced successfully by the model (Fig. 2c and 2d). In addition to the warming trend in the latter half of the 20th century, the model captures the slowdown of SAT increase both in the 1950s and in the 1960s. These changes are likely induced by increased anthropogenic aerosol emissions and resultant cooling through indirect aerosol effects, together with cooling attributable to large volcanic eruptions in the 1960s (Wilcox et al., 2013; Nozawa et al., 2005). However, distinct discrepancies between the model result and HadCRUT4 are found for SAT and SST, in the 1860s and particularly in the 1900s. This might be due to inevitable asynchronization between the simulation and observations on the phasing of the internal variability of climate. Kosaka et al. (2016) reported that there should have been four major cooling events due to tropical Pacific variability in the 20th century, one of which was found in the 1900s. They reported the other three events were around 1940, 1970, and 2000; however, discrepancies arising from these three events are not so evident in this study, likely because of the single ensemble simulation. It is also confirmed that the model exhibits short-term response of the TOA radiation balance following episodic volcanic events (Fig. 2a, vertical dashed lines), with resultant cooling of SAT and SST (Fig. 2a–c) and further propagation into the deeper ocean with extended cooling duration (Fig. 2d). Overall, the historical SAT increase in MIROC-ES2L, taking the difference between the average of 1850–1900 and 2003–2012, is 0.69 K, while the HadCRUT4-





based estimate by Stocker et al. (2013) is 0.78 K for the corresponding period. It is considered that the model shows good

performance in reproducing global physical fields. This is likely attributable to the inherited robust performance of the

physical core of the model (MIROC5.2), because MIROC-ES2L has only two feedback pathways of biophysical processes

on climate (DMS emission from the ocean and terrestrial processes associated with LAI dynamics) when the model is

driven by a prescribed $CO_2$ concentration. Both processes are likely to change the physical fields locally.

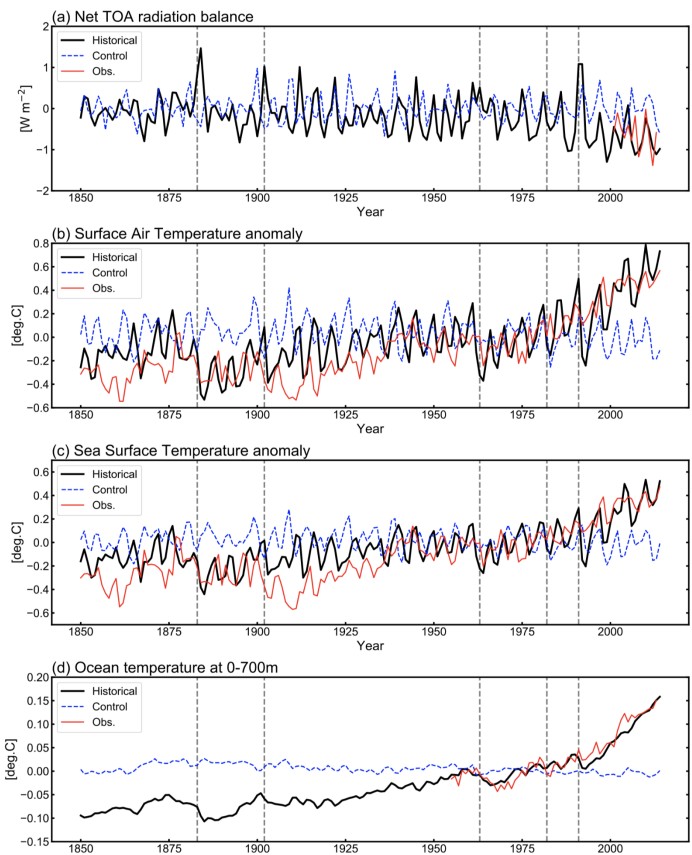

**Figure 2**

Comparison of HIST simulation results by MIROC-ES2L with observations: (a) net radiation balance at the top of the

atmosphere (TOA; upward positive), (b) global mean surface air temperature, (c) global mean sea surface temperature,

and (d) global mean ocean temperature at 0–700 m depth. Black, red, and blue lines represent historical simulations,

historical observations, and pi-control simulations, respectively. Vertical dashed lines represent the timing of major

volcanic eruptions (i.e., Krakatau (1883), Santa Maria (1902), Agung (1963), El Chichon (1982), and Pinatubo (1991)). In

panel (a), the simulation results are presented as anomalies from the 1850–2014 average of the CTL run. In panels (b), (c),





and (d), the results are presented as the anomaly from the 1961–1990 averages. Observation data for the radiation balance were obtained from the global product of CERES EBAF-TOA edition 4.0. Observation data for SAT and SST were obtained from HadCRUT4 (Morice et al., 2012) version 4.6 and HadSST (Kennedy et al., 2011) version 3.1.1,

respectively. Ocean temperature anomaly, updated from Levitus et al. (2012), is used for the comparison of ocean temperature at 0–700 m depth during the period 1955–2014.

### 3.1.2. Global carbon budget

The simulated net $CO_2$ uptake by land and ocean in cumulative values (i.e., changes in total carbon of land and ocean)

is shown in Fig. 3a and 3b, respectively. For land, the CTL run shows slight reduction of carbon of 7.6 PgC during the 165 years (i.e., 4.6 PgC per century), which is within the acceptable range for the CMIP6 exercise (10 PgC per century; Jones et al., 2016). The dashed gray line in Fig. 3a is the result from HIST-NOLUC, showing a natural land carbon sink in MIROC-ES2L of 166 PgC during 1850–2014. This is well comparable with the estimate of 185 ± 50 PgC by Le Quéré et al. (2018) for the same period (vertical gray bar in Fig. 3a), which was obtained from multiple offline terrestrial ecosystem

models with fixed land use. In addition, LUC is one of the factors to change drastically the historical land carbon amount, because positive (negative) LUC emission is linked directly with reduction (increase) of land carbon. Based on bookkeeping methods, Le Quéré et al. (2018) estimated the cumulative CE derived from LUC during 1850–2014 as 195 ± 75 PgC, whereas the simulated cumulative emission by MIROC-ES2L that is diagnosed by the difference in land carbon amount between HIST-NOLUC and HIST is 156 PgC.

Through being affected by both environmental and LUCs, MIROC-ES2L demonstrates in the HIST simulation that land carbon is reduced by approximately 60 PgC from the beginning of the simulation up until the middle of the 20th century (black line in Fig. 3a). This reduction should reflect LUC during this period because HIST-NOLUC does not show such a trend of decrease in the corresponding period (dashed gray line in Fig. 3a). From the 1960s, the model shows continuous carbon sequestration on land, resulting in positive net $CO_2$ uptake of 2.4 PgC yr$^{-1}$ in the 2000s (Table 3). This

continuous increase in the latter half of the 20th century is due to the combined effects of $CO_2$ fertilization, vegetation recovery associated with LUC, and the increase of nitrogen input via deposition and the use of fertilizer. This is displayed clearly in Fig. 3c, where the historical land carbon change is decomposed into the responses to (1) $CO_2$ increase (blue line, diagnosed by "HIST-NOLUC + HIST-BGC – HIST"; see Table 2), (2) climate change (red line, by "HIST – HIST-BGC"), and (3) LUC (green line, by "HIST – HIST-NOLUC"). In the latter half of the 20th century, land carbon sequestration

accelerated by $CO_2$ stimulation is clear, while climate change and the resultant terrestrial carbon loss also become evident. In addition, land carbon reduction induced by LUC is slightly alleviated in the corresponding period. During the historical period, MIROC-ES2L simulates total land carbon change of 44 PgC. This number drops to within the range of −10 ± 90 PgC (vertical black bar in Fig. 3a), where estimation uncertainties arising from both the terrestrial natural carbon sink and LUC emission by Le Quéré et al. (2018) are considered (calculated as $(\sigma_{LUC}^2 + \sigma_{SINK}^2)^{0.5}$, where $\sigma_{LUC}$ and $\sigma_{SINK}$ represent

the uncertainty range of LUC emission and the land sink, respectively, in Le Quéré et al., 2018).

For the ocean, the model shows a trend of increase of carbon accumulation in the CTL run (Fig. 3b). This is partly due to carbon removal by the sedimentation process that is newly introduced into MIROC-ES2L. In this process, an amount of carbon is extracted from the ocean bottom, which should be compensated by an equivalent input of carbon from the atmosphere through gas exchange processes. In the CTL run, the rate of carbon extracted from the ocean bottom is 0.068

PgC yr$^{-1}$ (Table 4), suggesting that the process removes 11 PgC throughout the entire simulation period of CTL (165





years). It is noted that Cias et al. (2013) suggested that the ocean was a net source of $CO_2$ in the preindustrial era to an amount of 0.7 PgC yr$^{-1}$, whereas our model shows it as a net sink in the same condition. This is likely attributable to the lack of a process of riverine carbon input in our model. For example, Cias et al. (2013) estimated that rivers obtain an external input of carbon of 0.9 PgC yr$^{-1}$ from the lithosphere, of which 0.2 PgC yr$^{-1}$ is removed by sedimentation and 0.7

PgC yr$^{-1}$ is lost to the atmosphere via gaseous exchange. The sedimentation process cannot explain all of the increase of oceanic carbon in the CTL run (30 PgC). Therefore, the remainder should be attributed to other reasons, e.g., the shortness of the spin-up period or imperfect mass conservation in the component model.

The HIST run shows the cumulative carbon uptake by the ocean, which is driven predominantly by $CO_2$ increase (Fig. 3b and 3d). In comparison with land, ocean carbon shows a relatively small response to climate change (red line in Fig. 3d), which is consistent with analysis of the carbon cycle feedback in an idealized scenario (Arora et al., 2013).

Furthermore, the model shows weak or almost no response against LUC (green line in Fig. 3d), although the ocean component in the model actually receives increased nitrogen input from rivers attributable to LUC and agriculture (Fig. 4, Table 4). It suggests that the increase of riverine nitrogen input due to LUC and agriculture would not have had global-scale impact on ocean carbon uptake in the historical period. The model simulates cumulative carbon uptake of 163 PgC

for 1850–2014, which is within the range of 150 ± 20 PgC (vertical black bar in Fig. 3b) reported by Le Quéré et al. (2018).

Overall, MIROC-ES2L qualitatively captures well the temporal evolution of carbon dynamics in the historical period, and the cumulative carbon uptake by both land and ocean is within the range of the estimates by Le Quéré et al. (2018). However, the model might overestimate net carbon uptake by the land and/or ocean, or underestimate LUC emissions,

because the diagnosed $CO_2$ concentration in the HIST run is 376 ppmv (Appendix D), which is lower (by 22 ppmv) than that actually monitored. This bias of diagnosed $CO_2$ concentration might be alleviated partially if the model were driven by anthropogenic $CO_2$ emissions, because in the emission-driven mode, lower atmospheric $CO_2$ concentration can weaken the land/ocean sink through the negative $CO_2$–carbon feedback. In addition, in the emission-driven mode, land and ocean are mutually interlinked via the atmospheric $CO_2$ concentration; thus, strong bias of $CO_2$ flux in one component can be

modulated by the other. This mechanism might reduce the bias of $CO_2$ fluxes of land and ocean simultaneously, or it might exacerbate $CO_2$ flux by imposing the flux bias of one onto the other. For more detail, simulations and analyses based on emission-driven configurations are necessary, as designed in C4MIP (Jones et al., 2016).





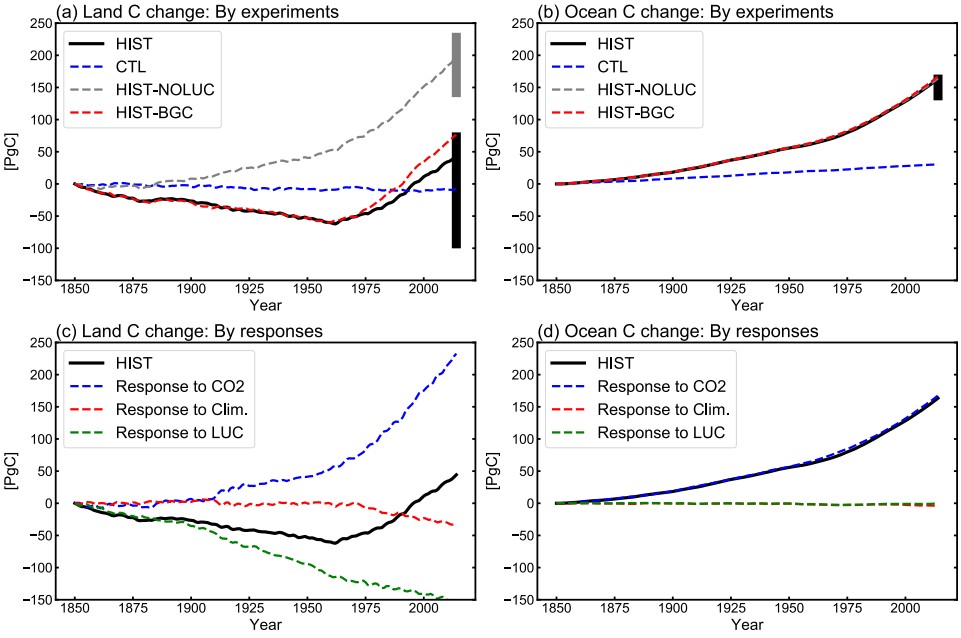


**Figure 3**

Land and ocean carbon change (i.e., cumulative net carbon uptake by land and ocean) in historical simulations. Upper panels present simulation results of historical (HIST, black lines), historical without land-use change (HIST-NOLUC, dashed gray), historical without global climate change (HIST-BGC, dashed red), and control (CTL, dashed blue) runs. For land calculation, carbon amount change in product pools for land use is taken into account. Vertical bars represent uncertainty ranges estimated from Le Quéré et al. (2018). Black bars correspond to HIST (1850–2014) run result, and gray bar represents the uncertainty range for natural carbon sink of land, which corresponds to HIST-NOLUC run in this study. In lower panels, HIST run result is shown again (black lines) together with decomposed response of land/ocean carbon against $CO_2$ increase (dashed blue), climate change (dashed red), and LUC (dashed green). Note that the ocean in MIROC-ES2L takes into account carbon removal via sedimentation process onto ocean floor; thus, the model exhibits continuous carbon uptake, even in the CTL experiment.

**Table 3**

Key variables of global land biogeochemistry: preindustrial condition (165 years) and the 2000s in the historical run (HIST).

*1: Net carbon uptake is calculated as the net ecosystem productivity minus the carbon emissions from product pools for land use.

*2: BNF by agriculture is also included.

*3: Net nitrogen uptake is calculated by annual changes in total nitrogen storage.



|  | Preindustrial | 2000s |
|---|---|---|
| Gross primary productivity (PgC yr[-1]) | 108.8 | 123.8 |
| Net primary productivity (PgC yr[-1]) | 57.7 | 67.2 |
| Heterotrophic respiration (PgC yr[-1]) | 56.9 | 59.4 |
| Net carbon uptake[*1] (PgC yr[-1]) | 0.0 | 2.4 |
| Vegetation carbon (PgC) | 538.4 | 543.3 |
| Soil organic carbon (PgC) | 1484.2 | 1491.0 |
|  |  |  |
| Biological fixation[*2] (TgN yr[-1]) | 97.2 | 135.9 |
| Deposition (TgN yr[-1]) | 19.6 | 65.5 |
| Fertilizer (TgN yr[-1]) | 0.0 | 114.0 |
| $N_2$ emission (TgN yr[-1]) | 71.5 | 110.8 |
| $N_2O$ emission (TgN yr[-1]) | 9.5 | 13.7 |
| $NH_3$ emission (TgN yr[-1]) | 1.9 | 19.5 |
| N leaching (TgN yr[-1]) | 17.3 | 33.4 |
| Net ecosystem nitrogen uptake[*3] (TgN yr[-1]) | 0.7 | 37.0 |
| Vegetation nitrogen (PgN) | 4.0 | 3.9 |
| Soil total nitrogen (PgN) | 74.9 | 75.3 |

**Table 4**

Key global ocean biogeochemical fluxes and concentrations under preindustrial control simulation and the 2000s.

|  | Preindustrial | 2000s |
|---|---|---|
| Net primary productivity (PgC yr[-1]) | 28.3 | 28.6 |
| Sinking particulate organic carbon at 100 m (PgC yr[-1]) | 7.8 | 7.9 |





| | | |
|---|---|---|
| Nitrogen fixation (TgN yr$^{-1}$) | 129.1 | 125.9 |
| Nitrogen deposition (TgN yr$^{-1}$) | 14.2 | 35.2 |
| Riverine nitrogen input (TgN yr$^{-1}$) | 17.5 | 33.9 |
| Denitrification (TgN yr$^{-1}$) | 142.2 | 164.5 |
| N$_2$O emission (TgN yr$^{-1}$) | 4.5 | 4.4 |
| Nitrogen flux into the sediment (TgN yr$^{-1}$) | 0.012 | 0.013 |
| N cycle imbalance (TgN yr$^{-1}$) | 14.1 | 26.1 |
| Atmosphere–ocean CO$_2$ flux (PgC yr$^{-1}$) | -0.15 | -2.37 |
| Carbon flux into sediment (PgC yr$^{-1}$) | 0.068 | 0.073 |
| Mean O$_2$ concentration (mmol m$^{-3}$) | 191 | 189.9 |
| Hypoxic volume ($10^{15}$ m$^3$; [O$_2$] < 80 mmol m$^{-3}$) | 34.2 | 34.3 |
| Suboxic volume ($10^{15}$ m$^3$; [O$_2$] < 5 mmol m$^{-3}$) | 2.3 | 2.7 |

### 3.1.3. Global nitrogen budget

MIROC-ES2L can simulate the global nitrogen cycle under interaction with climate and the carbon cycle, and the global N budget of land and ocean in the HIST simulation is shown in Fig. 4 as component fluxes. Comparison of the terrestrial nitrogen budget in the 2000s with the preindustrial condition (Table 3) reveals the inputs of nitrogen via deposition and fertilizer, which are controlled by forcing data, increase by 46 and 114 TgN yr$^{-1}$, respectively. In addition, BNF is also increased by 40% (39 TgN yr$^{-1}$), which is caused by areal expansion of agriculture for N-fixing crops (Fig. 4,

Supplementary Fig. 6). Previous studies have shown similar levels of increase. For example, Gruber and Galloway (2008) reported a value of 35 TgN yr$^{-1}$, and the absolute magnitude of agricultural BNF in the present-day condition was estimated as 50–70 TgN yr$^{-1}$ by Herridge et al. (2008) and 40 TgN yr$^{-1}$ by Galloway et al. (2008).

    For terrestrial nitrogen efflux, Gruber and Galloway (2008) reported N$_2$ emission in the unperturbed state was 100 TgN yr$^{-1}$, i.e., larger than found in this study (72 TgN yr$^{-1}$). However, in the present-day condition, they estimated the absolute

magnitude of N$_2$ emission as 115 TgN yr$^{-1}$, which is reasonably close to our model result (111 TgN yr$^{-1}$). MIROC-ES2L simulates the historical increase of N$_2$O emission from soil as 4.5 TgN yr$^{-1}$ from the preindustrial condition to the 2000s, which is comparable with the estimate of approximately 4 TgN yr$^{-1}$ for 1861–2015 derived from a model comparison study (Tian et al., 2018). However, the absolute magnitude of terrestrial N$_2$O emission fluxes in preindustrial and present-day conditions are likely overestimated (Table 3; Hashimoto, 2012).

Although it is difficult to obtain observation-based estimates on how much nitrogen was accumulated by the land ecosystem in the historical period, the model demonstrates net nitrogen uptake by land in the 2000s as 37 TgN yr$^{-1}$ (Table



3). This positive uptake is likely caused by increased total nitrogen input into the land ecosystem, as well as increased nitrogen demand by plants and soils under elevated $CO_2$ concentrations. Actually, 1PPY-BGC, in which nitrogen deposition, fertilizer application, and global warming are forced to maintain preindustrial levels and where only $CO_2$

perturbs the biogeochemistry, shows net increase of total nitrogen in the land ecosystem (data not shown), suggesting atmospheric $CO_2$ increase stimulates ecosystem nitrogen demand. We note that the model demonstrates nitrogen loss by LUC at a rate of >50 TgN yr$^{-1}$ (Fig. 4). It is because the harvested biomass in the model is translocated to product pools, and the nitrogen contained in the biomass is assumed lost with implicit chemical form, together with carbon loss as $CO_2$.

Compared with land, the model simulates relatively stable dynamics of the oceanic nitrogen budget but with larger
interannual variation (Fig. 4b). In the 2000s, oceanic BNF is simulated as 126 TgN yr$^{-1}$, which is almost at the same level (slightly below) that of the preindustrial state, i.e., 129 TgN yr$^{-1}$ (Table 4). This number is close to previously reported estimates of approximately 130 TgN yr$^{-1}$ (Eugster and Gruber, 2012). The invariant behavior of BNF in the model suggests the historical change in nitrogen input into the ocean is attributable primarily to two external sources: deposition and riverine input. Nitrogen deposition into the ocean, which is prescribed in the forcing data, shows an increase from 14
TgN yr$^{-1}$ in the preindustrial condition to 35 TgN yr$^{-1}$ in the 2000s. Riverine nitrogen input is shown to increase from 18 TgN yr$^{-1}$ in the preindustrial condition to 34 TgN yr$^{-1}$ in the 2000s (this is discussed further in Sect. 3.1.6 and Sect. 3.2.3). In this study, the gross nitrogen input into the ocean in the present-day condition is simulated as 195 TgN yr$^{-1}$. The value is reasonably close to the estimate of 200 TgN yr$^{-1}$ by Wang et al. (2019) and that of 209 TgN yr$^{-1}$ by Galloway et al. (2004); however, it is smaller than other published estimates (e.g., 294 TgN yr$^{-1}$; Codispoti et al. (2001) and 270 TgN yr$^{-1}$;
Gruber and Galloway (2008)). Denitrification, the main source of ocean nitrogen loss, is simulated as 142 TgN yr$^{-1}$ for the preindustrial condition and 165 TgN yr$^{-1}$ for the 2000s. These values are within the wide range of total denitrification rate estimated by previous studies, i.e., 145–450 TgN yr$^{-1}$ (Eugster and Gruber, 2012). It should be noted that the present model used in this study does not include sedimentary denitrification. Thus, the expected N flux by sedimentary denitrification is imposed on water-column denitrification, and the rate of water-column denitrification is likely
overestimated. Overall, the model exhibits oceanic N imbalance of 26.1 TgN yr$^{-1}$ in the present-day condition (Fig. 4, Table 4).

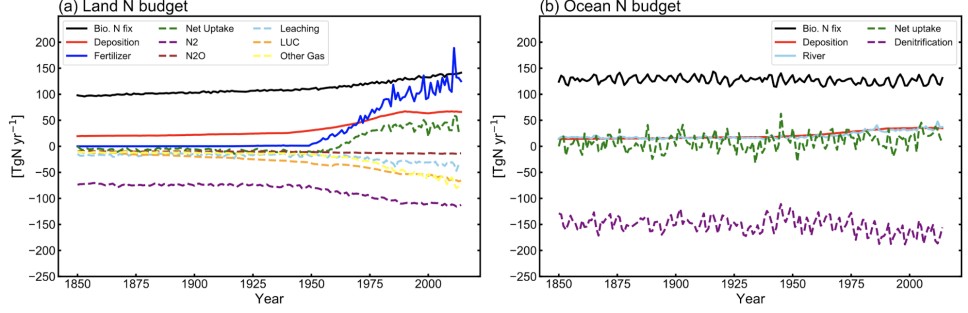

**Figure 4**

Global nitrogen budget of (a) land and (b) ocean in the HIST simulation. Solid lines represent the nitrogen input into the land/ocean, and dashed lines represent its fate. Positive (negative) values mean flux into (out of) the land/ocean. In





panel (a), BNF (black line) takes into account both natural and agricultural fluxes. LUC (dashed orange line) is an emission derived from the decay of LUC-product pools. Other gases (yellow line) represent the sum of $NH_3$ emission and

flux from abiotic sources. For the ocean, denitrification (purple line) includes both $N_2$ and $N_2O$ emissions. The rate of nitrogen loss by the sedimentation process onto the ocean floor is not shown in the figure because of the small size of the flux (<0.015 TgN yr$^{-1}$). All nitrogen gas emissions are diagnosed and thus have no effect on the radiative balance in the atmosphere or on air quality change.

### 3.1.4.  Climate: atmosphere and ocean physical fields

Here, we present an overview of the performance of the mean state of the physical fields, atmosphere, and land/oceanic basic variables of the model in comparison with various observational based data. The variables examined here are SAT, precipitation, SST, sea ice concentration and land snow cover, and mixed-layer depth, all of which are representative physical states associated with biogeochemical processes. Shown in Fig. 5 is the climatology of SAT (air temperature at 2-

m height) averaged over 1989–2009 for annual, December–February (DJF), and June–August (JJA) means, and the biases in comparison with the ERA-Interim dataset (Dee et al., 2011). The comparison suggests that the model performs well (biases <2°C) over the tropics and most of the global area in terms of both annual mean and seasonality. However, obvious warm biases exist over the Southern Ocean and Antarctica. This is a general tendency of state-of-the-art general circulation models, and both MIROC5 (Watanabe et al., 2010) and MIROC6 (Tatebe et al., 2019) also suffer this problem.

The warm bias in the Southern Ocean can be attributed to the difficulty of expressing cloud cover in the region (Hyder et al., 2018) and poor representations of the mixed-layer depth and deep convection in the open ocean attributable to the lack of modeled mesoscale processes in the Antarctic Circumpolar Current (Tatebe et al., 2019). A related warm bias in SST over the Southern Ocean is also confirmed, which is discussed later.


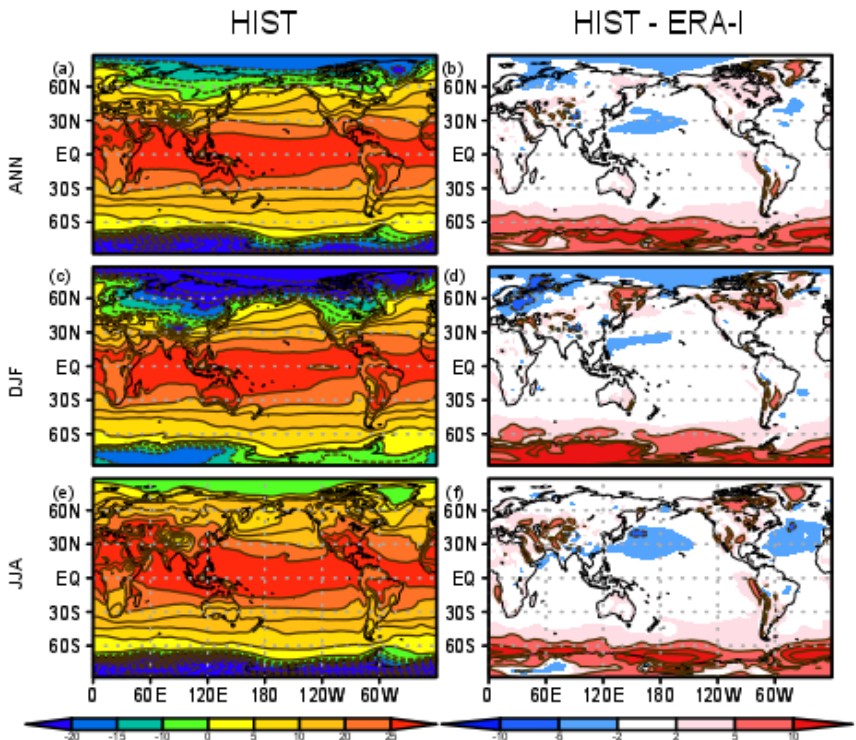

**Figure 5**

 Air temperature at 2-m height (°C) in the HIST simulation presented as 1989–2009 climatology and the bias compared with the ERA-Interim dataset (Dee et al., 2011) for (a) and (b) annual, (c) and (d) DJF, and (e) and (f) JJA means.


 Figure 6 shows the precipitation distribution in the HIST experiment in comparison with the Global Precipitation Climatology Project (GPCP) dataset (Adler et al., 2003). In general, the precipitation distribution is reasonably well represented in the model. The Intertropical Convergence Zone (ITCZ) is reproduced well in the experiment, except that the simulated South Pacific Convergence Zone is shifted equatorward relative to the GPCP, which is the so-called double

675 ITCZ syndrome (Bellucci et al., 2010). Over continental areas, the model is effective in capturing the spatial pattern of both the annual mean precipitation and the seasonality. However, positive precipitation biases are evident in arid and semiarid regions of central Asia, Australia, Africa, and the western side of North America.



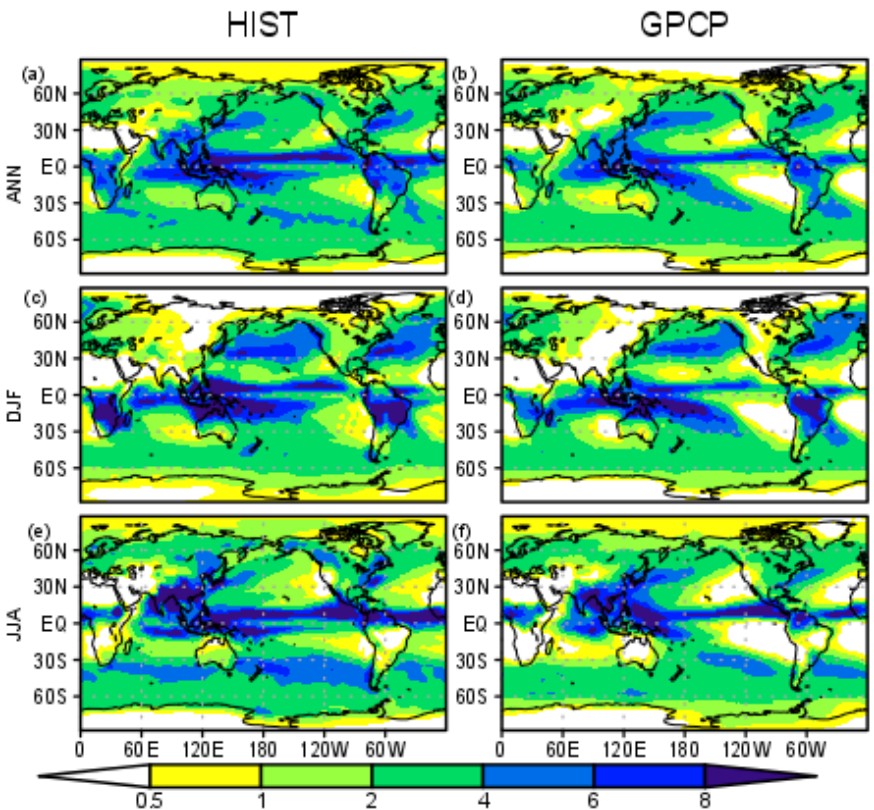

**Figure 6**

Precipitation distributions (mm d$^{-1}$) in the HIST simulation and GPCP dataset, (Adler et al., 2003) for (a) and (b) annual,
(c) and (d) DJF, and (e) and (f) JJA means averaged over 1981–2000.

It is important to represent SST well in terms of simulating climate change (Ohgaito and Abe-Ouchi, 2009) and
reproducing biogeochemical fields on the ocean surface. Figure 7 presents the modeled SST and bias in comparison with
the World Ocean Atlas 2013 (WOA2013; Locarnini et al., 2013). Generally, the model performs well, confirmed by the
large extent of the area with minimal bias (colored white in Fig. 7). However, obvious bias is evident, e.g., the warm bias
in the Southern Ocean, as already explained above (Fig. 5). A cold bias is also evident over the northwestern Pacific
Ocean, which is attributable partially to the mixed layer being too deep (as shown in Fig. 9). One other possible reason for
this bias is the resolution of the model being too coarse to reproduce properly the Western Boundary Currents and their
extensions.

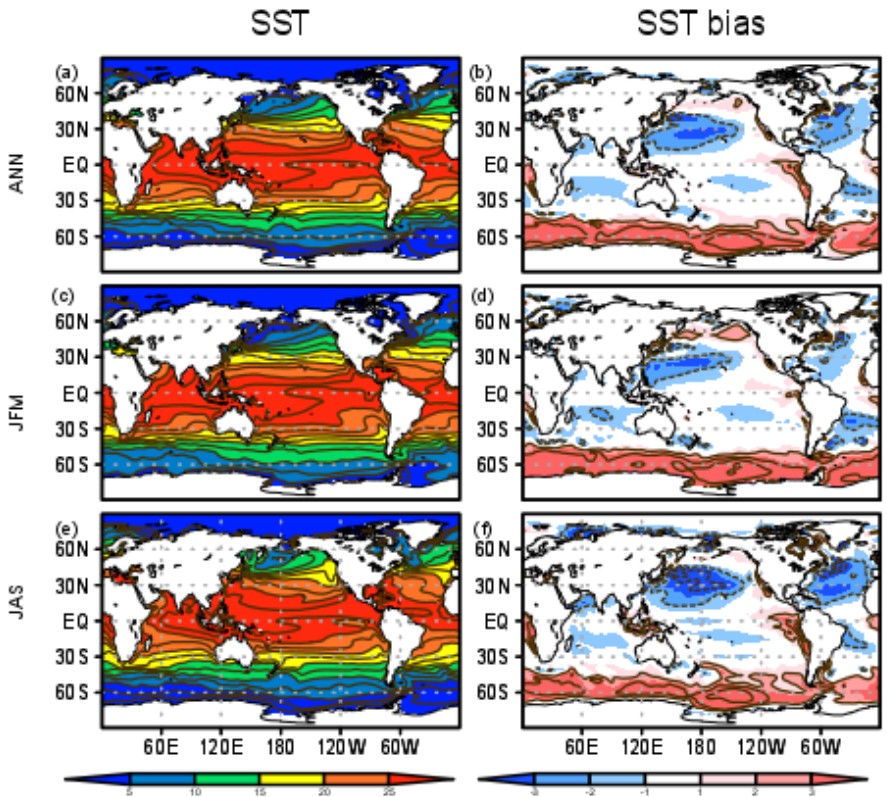

**Figure 7**

SST (°C) in the HIST simulation presented as 1955–2012 climatology and the bias in comparison with WOA2013 (Locarnini et al., 2013) for (a) and (b) annual, (c) and (d) JFM, and (e) and (f) JAS means.

      The model performance in simulating sea ice concentration and snow cover over land for both March and September is shown in Fig. 8, in comparison with observational data (Special Sensor Microwave Imager (SSM/I; Kaleschke et al.,

2001) for sea ice concentration and Moderate-resolution Imaging Spectroradiometer (MODIS; Hall et al., 2006) for snow cover). Sea ice extent is represented well for both months, although the summertime concentration peak is slightly smaller than observed. The extent of the snow-covered area is also represented well, likely owing to the updated scheme for subgrid snow representation (Nitta et al., 2014; Tatebe et al., 2019). However, the fine structure of the snow cover is lost in the simulation, which is likely attributable to the coarse resolution of the modeled atmosphere and land. The reasonable

performance in reproducing land snow seasonality also implies reasonable performance in reproducing LAI seasonality in the boreal region, because snowmelt (accumulation) and leaf flush (shedding) processes are associated with each other (Supplementary Fig. 7).

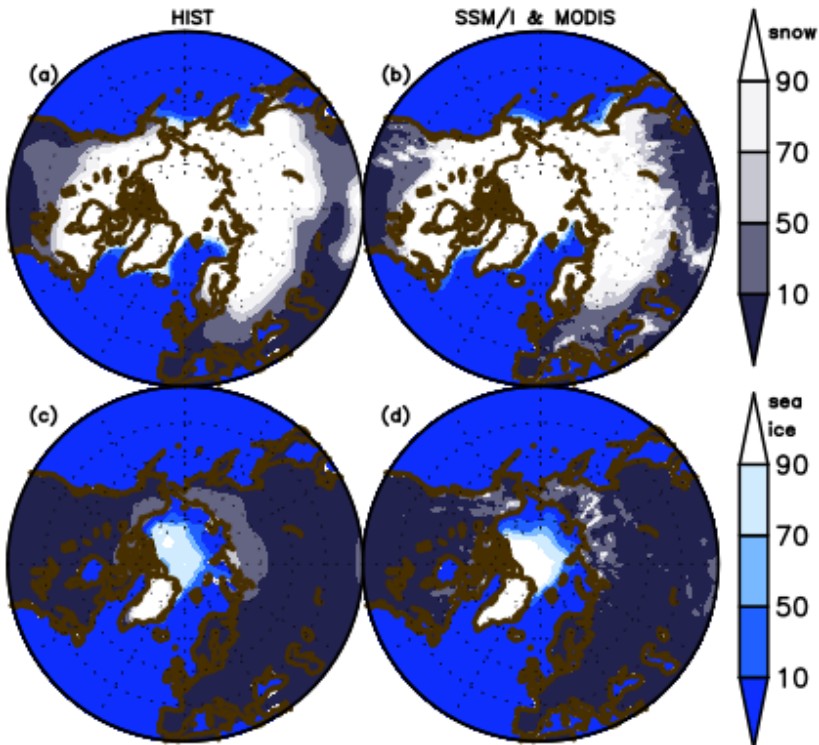

**Figure 8**

Northern Hemisphere sea ice concentration and land snow fraction (%) in the HIST simulation presented as 2003–2013 climatology and in comparison with SSM/I (Kaleschke et al., 2001) and MODIS (Hall et al., 2006) data for (a) and (b) March and (c) and (d) September.

Figure 9 shows the mixed-layer depth in comparison with the Mixed-Layer dataset of Argo, Grid Point Value (MILA_GPV; Hosoda et al., 2010). It is confirmed that the HIST simulation captures well both the spatial pattern and the seasonality change in mixed-layer depth. In the Northern Hemisphere winter, the structure of the deep mixed layer over the western North Pacific Ocean is consistent with observations, although the actual depth is overestimated. There could be various reasons for such discrepancy, e.g., wind stress or ocean temperature profile. The deep mixed layer in the high-latitude regions of the North Atlantic Ocean is also consistent with observations. In addition, the shallow mixed layer in low latitudes is generally captured well by the simulation, and the depth that is maintained at around 100 m over the Southern Ocean is consistent with observations. In austral winter, MILA_GPV shows the mixed layer develops to more than 200 m over the Indian Ocean and the Pacific sector of the Southern Ocean, whereas it is shallow (around 50 m) in the tropics and the Northern Hemisphere (Fig. 9d). The model captures the general pattern in austral winter, although the extent of the simulated deeper mixed-layer depth of more than 200 m in the Southern Ocean is larger than that of MILA_GPV (Fig. 9c).







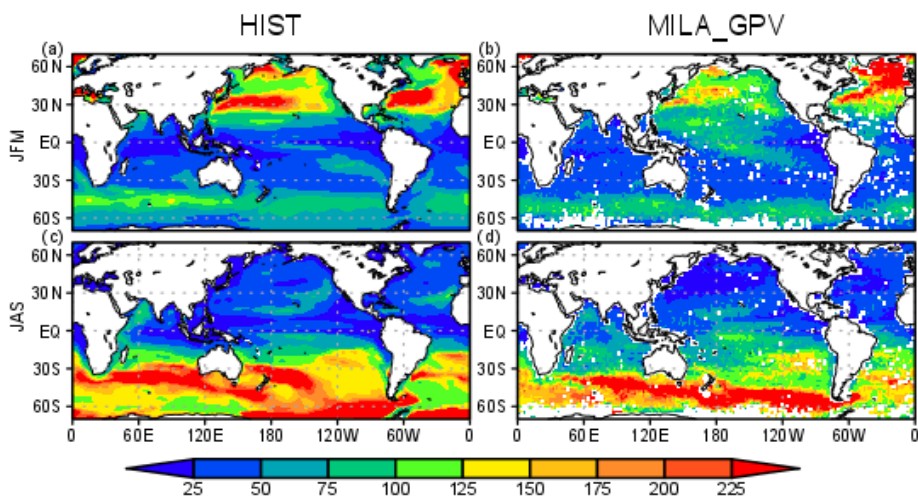

**Figure 9**

Mixed-layer depth (m) in the HIST simulation presented as 2000–2010 climatology and comparison with MILA_GPV data set (Hosoda et al., 2010) for (a) and (b) JFM and (c) and (d) JAS means.

### 3.1.5.  Land biogeochemistry

Model performance in relation to land biogeochemistry is evaluated based on the spatial distributions of three fundamental variables of the land carbon cycle in comparison with observation-based products. First, GPP in the HIST simulation is compared with the global product by Jung et al. (2011) (Fig. 10a–c). The model simulates higher productivity (>2000 gC m$^{-2}$) in the tropical forests of central Africa, Southeast Asia, and Southern America, although the productivity in these regions is generally underestimated in comparison with the observation-based product. This underestimation is likely attributable to the use of the parameter values of photosynthetic capacities ($K_{PSAT1}$ and $K_{PSAT2}$ in Appendix A) from Kattge et al. (2009). This is because Kattge et al. (2009) also showed such substantial depression of photosynthetic capacity in the tropics. The model captures well the moderate productivity of vegetation in savanna regions such as the eastern side of South America and the marginal region surrounding central Africa. Moderate GPP is also found in the Northern Hemisphere in the region 20°–45°N, where a large proportion of land cover is dominated by both natural and agricultural vegetation (Supplementary Fig. 2). The GPP gradient from moderate to lower GPP in boreal to tundra regions of Eurasia and North America is captured well by the model. The model estimates global GPP at 124 PgC yr$^{-1}$ in the 2000s (Table 3), which is within the range of 106–140 PgC yr$^{-1}$ produced by the CMIP5 ESMs and is reasonably close to the value of 119 PgC yr$^{-1}$ derived from an observation product (1986–2005 average; Anav et al., 2013).

To evaluate the simulated vegetation carbon, we compare the model results of forest carbon, not total vegetation carbon, with those of Kindermann et al. (2008) (Fig. 10d–f). The model reproduces the reasonably high density of biomass in tropical forests, although the values are smaller than the observation product (Fig. 10f). This is attributable partly to the





underestimation of GPP in this region, as described above. In high-latitude regions of the Northern Hemisphere (around 50°N), the model overestimates biomass density, particularly in terms of the evergreen coniferous forests that extend

across western Siberia and North America. Considering the GPP in these regions is captured reasonably well by the model (Fig. 10a and 10b), the overestimation of boreal forest biomass is likely due to the underestimated turnover rate of forest carbon. Slight overestimation of biomass is also found in the region where intensive cultivation has occurred, i.e., Europe, Southeast–East Asia, and eastern America. The model estimates global vegetation carbon including all types of vegetation at 543 PgC (Table 3).

In Fig. 10g–i, the model results of soil organic carbon (SOC) are compared with two different types of SOC product: harmonized soil property values for broadscale modeling (WISE30sec) by Batjes (2016) and NCSCDv2 by Hugelius et al. (2013). The former is a global dataset that represents soil column SOC down to the depth of 2 m, whereas the latter targets only the high-latitudinal region of the Northern Hemisphere at different soil depths (~1, ~2, and ~3 m). Comparison with WISE30sec confirms that the model successfully captures the spatial distribution of lower carbon accumulation in arid and

tropical regions and higher SOC in boreal regions in the Northern Hemisphere. However, the simulated zonal mean SOC in the boreal regions is about half that of WISE30sec (Fig. 10i). This is likely attributable to different treatment of the vertical profile of SOC, i.e., WISE30sec covers the total SOC down to 2 m depth, while the model simulates SOC dynamics with an implicit vertical profile, considering only the difference of upper SOC as litter form and lower SOC as humus. The model result in the boreal region is comparable with the NCSCDv2 estimation for 1-m depth. We note, as

mentioned by Todd-Brown et al. (2012), large uncertainty remains in the estimation of SOC amount, especially in boreal regions. Globally, SOC is simulated as 1491 PgC (Table 3) in this study, which is smaller than the value of 2060 ± 215 PgC of WISE30sec (Batjes, 2016) but comparable with the range of 890–1660 PgC, as estimated by Todd-Brown et al. (2012) based on the Harmonized World Soil Database v1.2 (FAO/IIASA/ISRIC/ISS-CAS/JRC, 2012).

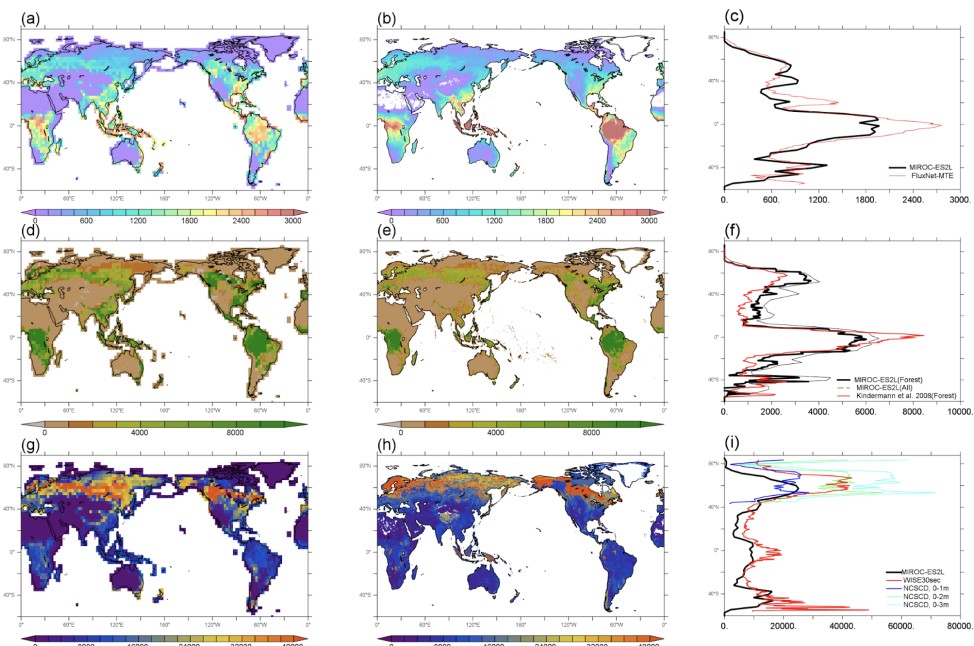






**Figure 10**

Comparison of carbon flux and storage of the land ecosystem between the HIST simulation by MIROC-ES2L and an
observation-based dataset. Upper panels show comparison of GPP (gC m$^{-2}$ yr$^{-1}$) averaged over 1982–2011: (a) model
result, (b) FluxNet-MTE of Jung et al. (2011), and (c) zonally averaged distributions. Middle panels show vegetation
carbon (gC m$^{-2}$): (d) model result of forest carbon (obtained by masking the total vegetation carbon where forest coverage
is <5%), (e) forest carbon estimated by Kindermann et al. (2008), and (f) zonally averaged distributions, where solid black
and red lines represent forest carbon, and the dashed thin line is the total vegetation carbon simulated by the model. Lower
panels show SOC: (g) model result, (h) observation-based product of harmonized soil property values for broadscale
modeling (WISE30sec) by Batjes (2016), and (i) zonally averaged distributions, where the model result and WISE30sec
are shown by black and red lines, respectively. Blue, green, and light blue lines in panel (h) are NCSCDv2 by Hugelius et
al. (2013), which is an independent estimate of SOC in the high-latitude region of the Northern Hemisphere at different
soil depths (blue: 0–1 m, green: 0–2 m, and light blue: 0–3m).

### 3.1.6.   Ocean biogeochemistry

In this section, we evaluate the simulated surface and vertical distributions of nitrate, phosphate, dissolved Fe, NPP,
oxygen, DIC, and alkalinity against observations (Fig. 11). The observations comprise the World Ocean Atlas 2013
(WOA2013; Garcia et al., 2014a, 2014b) for macronutrients and oxygen, GEOTRACES dataset (updated to its 2015
version; Tagliabue et al., 2012) for dissolved iron, Global Ocean Data Analysis Project version 2 (GLODAPv2; Lauvset et
al., 2016) for DIC and alkalinity, and SeaWiFS (Behrenfeld and Falkowski, 1997) satellite observations for NPP.

The simulated surface distributions of nitrate and phosphate are generally in agreement with the WOA2013 datasets
(Fig. 11a and 11b). The surface macronutrient concentrations in high nitrate, low chlorophyll (HNLC) regions (e.g., the
Southern Ocean, North Pacific Ocean, and eastern equatorial Pacific Ocean) are higher than produced by the ocean
biogeochemical component of our previous model (Watanabe et al., 2011) and they are more consistent with observed
values. This increase of macronutrients in HNLC regions is reasonable because implementation of both the iron cycle and
the iron limitation on phytoplankton growth can reduce macronutrient utilization in these regions. Ocean circulation
influences the distribution of nutrient concentrations. In the Southern Ocean, the deep mixed-layer depths simulated by the
model can cause overestimation of entrainment of nutrients to the surface water and thus produce high nutrient bias (Fig.
9). The simulated vertical nitrate concentrations compare reasonably well with observed values (Fig. 11a). This is the
result of the near balance between nitrogen cycle sources (i.e., nitrogen fixation, atmospheric nitrogen deposition, and
riverine nitrogen input) and sinks (i.e., denitrification, N$_2$O emission, and sedimentary loss) over the long spin-up period.

The concentration of dissolved iron in the open ocean is highest in the subtropical North Atlantic Ocean and in the
Arabian Sea (Fig. 11c), which is consistent with the pattern observed in GEOTRACES. Such high concentrations are
caused by enhanced dust deposition from the Sahara Desert. In the remainder of the open ocean, dissolved iron
concentrations are generally <0.2 μmol m$^{-3}$, especially in HNLC regions. The model captures well the main observed
patterns in the surface ocean. The very high iron concentrations (>1 μmol m$^{-3}$) both observed and simulated along coasts
and over continental margins are the result of iron input from sediment. The average simulated dissolved Fe concentration
in the surface ocean (0–100 m) is 0.39 μmol m$^{-3}$, which is lower than observed (0.52 μmol m$^{-3}$) but within the range of the
iron model intercomparison project (FeMIP; Tagliabue et al., 2016). One factor not accounted for in our model is the





variation in the solubility of iron in aerosols, which depend not only on the source chemical composition but also on atmospheric processing during transport (Ito et al., 2019). Consideration of different degrees of atmospheric Fe processing could reduce the overestimations of dissolved Fe concentration in the North Atlantic Ocean and North Pacific Ocean (Ito et al., accepted). Our model also neglects variations in sedimentary iron flux. Observations found iron release or burial in sediment is dependent on the oxygen concentration of bottom water (Noffke et al., 2012), ambient temperature (Sanz-

Lázaro et al., 2011), and amount of OM that reaches the sea floor and is remineralized therein (Elrod et al., 2004). To simulate more realistic iron distributions, these processes should be considered in future studies.

Reproducing the spatial pattern of nutrient limitation on phytoplankton growth is crucial for accurate prediction of primary production and for reflecting in the simulations the consequences of ongoing anthropogenic perturbations to oceanic nutrient cycles (Moore et al., 2013). The model reasonably reproduces the HNLC regions because of the iron

limitation in the subarctic North Pacific Ocean, equatorial Pacific Ocean, and Southern Ocean (Supplementary Fig. 8), although the subarctic North Pacific Ocean and equatorial Pacific have larger HNLC zones than observed upwelling regions. This is likely due to underestimation of surface iron concentrations and/or a relatively high half-saturation constant for iron uptake (Appendix B). Nitrogen limitation occurs throughout much of the surface low-latitude ocean where nitrogen supply from the subsurface is relatively slow.

Based on the distribution pattern of nutrients and the limitations, annual NPP is simulated as 28.6 PgC yr$^{-1}$ (Table 4). This value is lower than a satellite-based estimate of 35–78 PgC yr$^{-1}$ (Carr et al., 2006) and it is also lower than the range of 30.9–78.7 PgC yr$^{-1}$ derived from the CMIP5 models (Bopp et al., 2013). This is likely attributable to the high half-saturation constant for iron uptake, as mentioned above. Although intense primary productivity in coastal regions is not resolved by the coarse grid, the modeled NPP agrees with the basin-scale patterns of observation-based NPP. The values

of both modeled and observed NPP are high in regions of equatorial upwelling, the North Atlantic Ocean, and the Southern Ocean north of the polar front, whereas they are low in subtropical gyres (Fig. 11g). Global export production is estimated as 7.9 PgC yr$^{-1}$, which is the upper bound of the CMIP5 models (4.9–7.9 PgC yr$^{-1}$; Bopp et al., 2013).

Simulated surface distribution of dissolved oxygen compares reasonably well with observations (not shown). This is because the surface oxygen concentration is close to its solubility value and it is strongly constrained by SST. At depth,

oxygen minimum zones (OMZs) in the eastern equatorial Pacific Ocean, eastern tropical Atlantic Ocean, Arabian Sea, and Bay of Bengal are reproduced well (Fig. 11f). However, the model produces oxygen concentration values higher than observed; thus, it underestimates the hypoxic volume ($[O_2] < 80$ mmol m$^{-3}$) by a factor of three in comparison with data-based estimates (Bianchi et al., 2012). Note that existing global ocean biogeochemical models have difficulty in reproducing OMZs owing to coarse resolution and simple globally tuned parameterizations of vertical fluxes of OM

(Cocco et al., 2013; Bopp et al., 2013). The positive bias in oxygen might be driven by wintertime mixing in the Southern Ocean and North Pacific Ocean that is too intense (Fig. 9) and transports too much surface oxygen to depth.

The model also captures the global-scale patterns of observed DIC and alkalinity (Fig. 11d and 11e). High values of these parameters in subtropical gyres (and also in the Southern Ocean for DIC) are found in the model output and observations. Overestimation of alkalinity in subtropical gyres leads to overestimation of DIC because alkalinity affects

the capacity of the ocean to take up and store atmospheric $CO_2$.



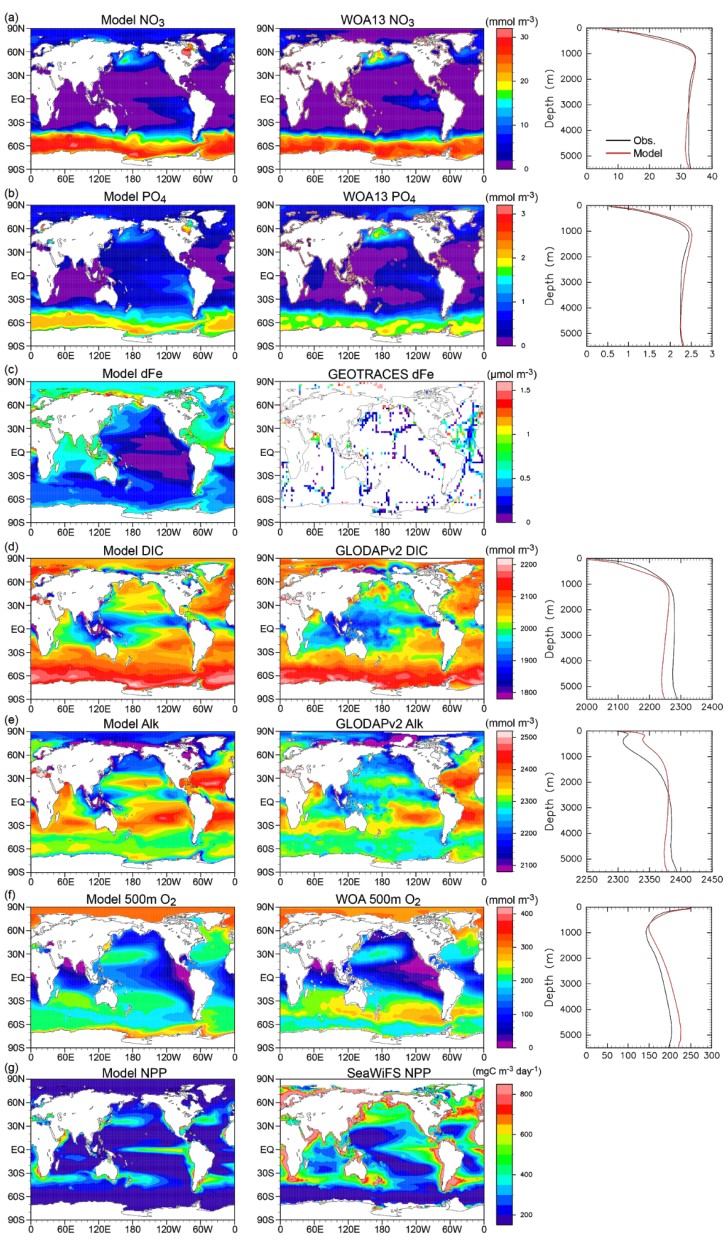

**Figure 11**

Comparison between model output and observations for key oceanic biogeochemical tracers. Simulated annual mean

surface (a) nitrate, (b) phosphate, (c) DIC, (d) alkalinity, (e) dissolved oxygen at 500 m depth, and (f) surface NPP for the

2000s are compared with observations from the WOA2013 (Garcia et al., 2014a, 2014b) and GLODAPv2 datasets

(Lauvset et al., 2016), as well as SeaWiFS (Behrenfeld and Falkowski, 1997) satellite observations. Left and central

panels show horizontal distributions of model output and observations. Right panels show vertical distributions of model

output (red lines) and observations (black lines).






## 3.2. Sensitivity analysis

### 3.2.1. Sensitivity of land biogeochemistry

To evaluate the sensitivities of modeled land biogeochemistry, we focus on GPP and its response to external forcing in the terrestrial system because this carbon flux is the primary driver of land carbon input. GPP change was calculated by
taking the difference of the 2005–2014 averages between the HIST and CTL runs. Then, as diagnosed in Fig. 3c, the GPP change was decomposed into the response to (1) $CO_2$ increase, (2) climate change, and (3) LUC and agricultural change (Fig. 12) based on the simulation results of HIST, HIST-NOLUC, and HIST-BGC (Tables 1 and 2). Additionally, the GPP changes were further decomposed into the contributions from non-crop (i.e., contribution of primary/secondary vegetation, urban, and pasture) and crop tiles by weighting the GPP of each tile by their areal fractions on a grid.

In Fig. 12d–f, it can be seen that $CO_2$ increase in the historical period is the main driver of change in the land carbon cycle, and that the $CO_2$ fertilization effect prevails over most land areas except desert regions. In contrast, GPP response to climate change shows both positive and negative signs (Fig. 12g–i) with relatively smaller magnitudes. Mid- to high-latitude regions of the Northern Hemisphere shows positive change in GPP, likely due to lengthening of the vegetation growth season, enhanced plant growth following accelerated soil mineralization due to warming, and other mechanisms
(e.g., soil water increase via precipitation and permafrost melting). In semiarid regions of the Southern Hemisphere (i.e., Africa, South Asia, Northern Australia, and the eastern side of South America), GPP shows slight reduction. As these regions have less precipitation in comparison with the tropics, the reduction in GPP is likely associated with precipitation change.

In addition to the responses to $CO_2$ increase and climate change, the model demonstrates spatial variation in the
response of GPP to LUC (Fig. 12j). Historical LUC reduces the non-crop GPP contribution (Fig. 12k), while the crop contribution is enhanced (Fig. 12l). In particular, regions with intensive agriculture (Western Europe, East Asia, and Northwest America) show net positive change of GPP as grid averages (Fig. 12j), where increases in the crop contribution overcome reductions in the non-crop contribution (Fig. 12k and 12l). In the model, the crop contribution to GPP can be intensified by the following: 1) increasing the areal fraction of the crop tile following LUC forcing; 2) changing the
vegetation type from natural vegetation to crop, whereby the latter has higher photosynthetic capacity than natural plant functional types (given as parameters that relate photosynthetic capacity with leaf nitrogen concentration, Appendix A); 3) applying nitrogen fertilizer to crop tiles; and 4) increasing nitrogen input via nitrogen-fixing crops, which is considered in the model as a subcategory of crop tiles. Indeed, the total area of cropland increases in the 20th century in the HIST simulation (Supplementary Fig. 2), which is reflected by the model producing an increase of nitrogen input via fertilizer
application and biological fixation on the global scale (Fig. 4a).

By responding to $CO_2$ increase, climate change, and LUC, most land areas show increased GPP in the historical period (Fig. 12a), and regions with intensive agriculture show greater increase in GPP than induced solely by the $CO_2$ fertilization effect (Fig. 12a and 12d). This suggests modeled GPP is sensitive to land use and agricultural management forcing in addition to the increase of $CO_2$, and that this might be one of the reasons for the slowing of LUC-induced land
carbon reduction in the latter half of the 20th century in the HIST simulation (green line in Fig. 3c).

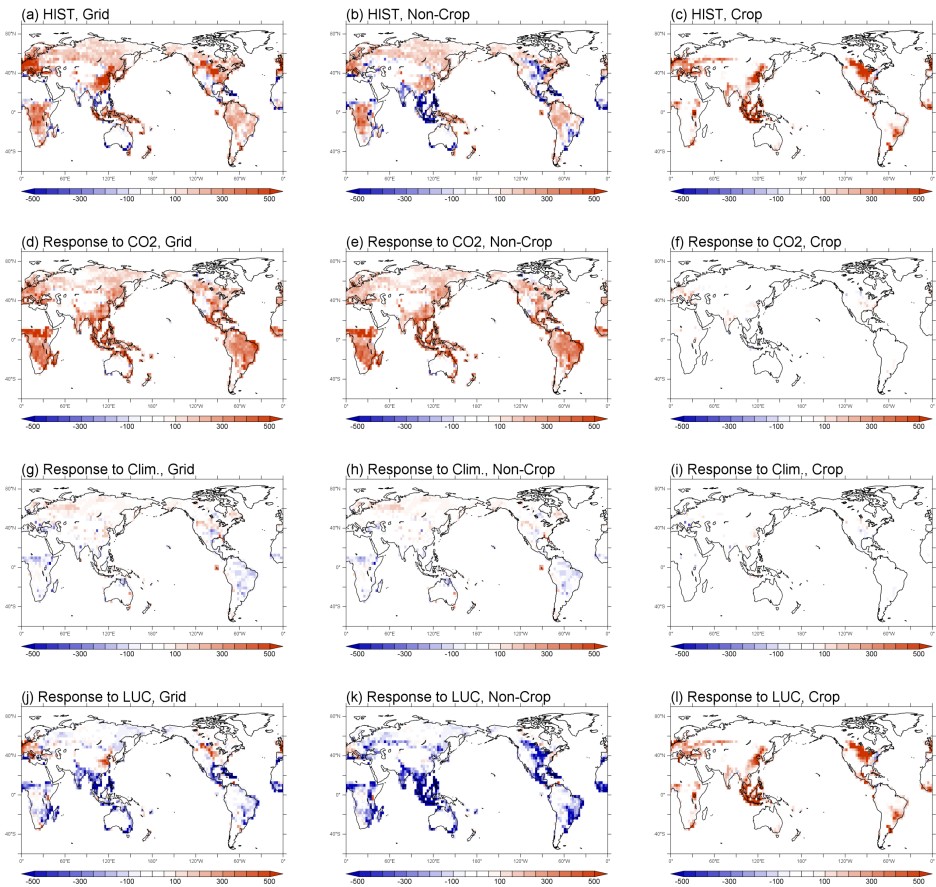

**Figure 12**

(Upper row) Changes in GPP (gC m$^{-2}$ yr$^{-1}$) in HIST derived by taking the difference of the 2005–2014 averages of GPP between HIST and CTL. (Second row) GPP response to CO$_2$ increase diagnosed from simulation results of HIST, HIST-NOLUC, and HIST-BGC. (Third row) GPP response to climate change diagnosed by taking the difference between the simulation results of HIST and HIST-BGC. (Lower row) GPP response to LUC obtained by taking the difference between HIST and HIST-NOLUC. GPP changes in each left-hand panel are further decomposed into contributions from (middle panels) non-crop tiles (primary vegetation, secondary vegetation, urban, and pasture) and (right-hand panels) crop tiles.

### 3.2.2. Sensitivity of ocean biogeochemistry

In this section, we investigate the sensitivity of oceanic NPP to external nutrient inputs from atmospheric deposition and river discharge processes under preindustrial conditions because these processes are newly incorporated into the ESM. Through combination of the simulation results of CTL-D, NO-NR, NO-NRD, and NO-FD (Tables 1 and 2), the impacts of nutrient input on both nutrient concentration and primary productivity are analyzed (Fig. 13 for N input assessment and




Fig. 14 for Fe), and the spatial patterns of simulated nutrient limitation on NPP in the four experiments are examined (Fig. 15).

First, the impacts of riverine N input on the surface nutrient concentration and NPP are assessed by subtracting the zero-input scenario NO-NR from the control experiment CTL-D (Tables 1 and 2). Surface NPP is increased by riverine N

input (by >10 gC m$^{-3}$ yr$^{-1}$) in coastal areas such as the North Brazil Shelf and Gulf of Mexico (Fig. 13a). In comparison with the pattern of distribution of nutrient limitation (Fig. 15a and 15b), it is clear that NPP increase occurs in N-limited regions in the open ocean. Conversely, NPP decreases in Fe-limited regions because the NPP increase in N-limited regions consumes surface dissolved Fe. Surface $NO_3$ concentrations increase only slightly in N-limited regions because $NO_3$ is immediately consumed locally by phytoplankton. Remarkable increase in surface $NO_3$ concentrations is found in Fe-

limited regions such as the Kara Sea, North Atlantic Ocean, Hudson Bay, and Subantarctic Ocean. Global NPP increases by 0.7 PgC yr$^{-1}$ (by 2.5% in comparison with NO-NR). This value is comparable with the finding of da Cunha et al. (2007), who estimated a 5% increase in primary production due to riverine nutrient input. Note that nutrient retention in estuarine areas is not considered in our model. Thus, most nitrogen supplied from river mouths can easily be conveyed to the open ocean. Given that a recent modeling study estimated that approximately 75% of riverine nitrogen globally escapes from

shelf areas to the open ocean (Sharples et al., 2016), our results on the impact of riverine N on NPP should be viewed as an upper limit for the estimation.

Second, the effects of atmospheric N deposition on surface nutrient concentration and NPP are evaluated by subtracting the zero-input scenario NO-NRD from the NO-NR experiment (Tables 1 and 2). Similar to riverine N input, atmospheric N deposition causes an increase of NPP in N-limited regions and a global increase in $NO_3$ (Figs. 13b, 15a, and 15c).

According to deposition flux, significant changes in NPP are found in coastal areas and low-latitude regions of the Pacific Ocean. Global NPP increases by 0.3 PgC yr$^{-1}$ (by 1% in comparison with NO-NR), which is consistent previous estimates (Duce et al., 2008; Moore et al., 2013).

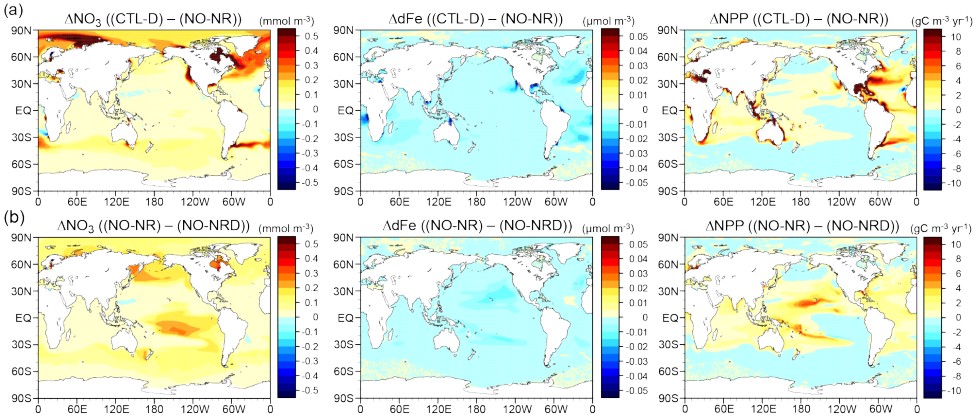


**Figure 13**

Changes in (left) surface nitrate, (center) dissolved iron, and (right) NPP driven by nitrogen input from (a) rivers (CTL-D − NO-NR) and (b) atmospheric deposition (NO-NR − NO-NRD).



Finally, changes in surface nutrient concentration and NPP, driven by atmospheric Fe deposition, are calculated by
subtracting the zero-input scenario NO-FD from the control experiment CTL-D (Tables 1 and 2). In contrast to N input,
atmospheric Fe deposition causes an increase of NPP in Fe-limited regions and a decrease in N-limited regions (Figs. 14,
15a, and 15d). Significant Fe increase is found in N-limited regions. Global NPP and export production increase by 1.8
and 0.8 PgC yr$^{-1}$, respectively (by 6.7% and 11%, respectively, in comparison with NO-FD). These percentage increases

are consistent with previous estimations by Moore et al. (2013). However, the sensitivity of export production to Fe
deposition from dust is higher than reported by Tagliabue et al. (2014), who estimated export production increases by
0.06–0.11 PgC yr$^{-1}$. Therefore, it seems difficult to obtain robust sensitivity both of iron and of the biological cycle to iron
input because of the high uncertainty regarding the iron cycle among models. Although nitrogen input from both
deposition and rivers has little effect on the spatial patterns of distribution of nutrient limitation (Fig. 15a–c), iron input

from the atmosphere changes the pattern in low-latitude regions from one of iron limitation to nitrogen limitation (Fig. 15a
and 15d).

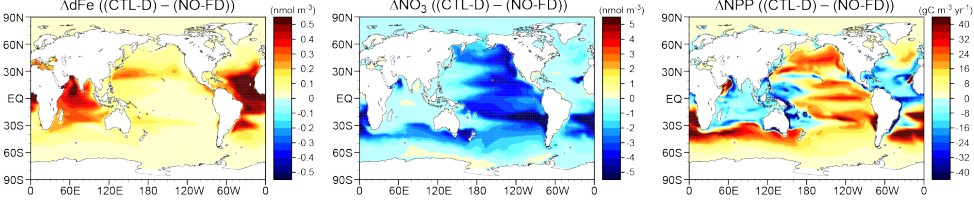

**Figure 14**

Changes in (left) surface dissolved iron, (center) nitrate, and (right) NPP driven by dissolved iron input from dust
(CTL-D − NO-FD).




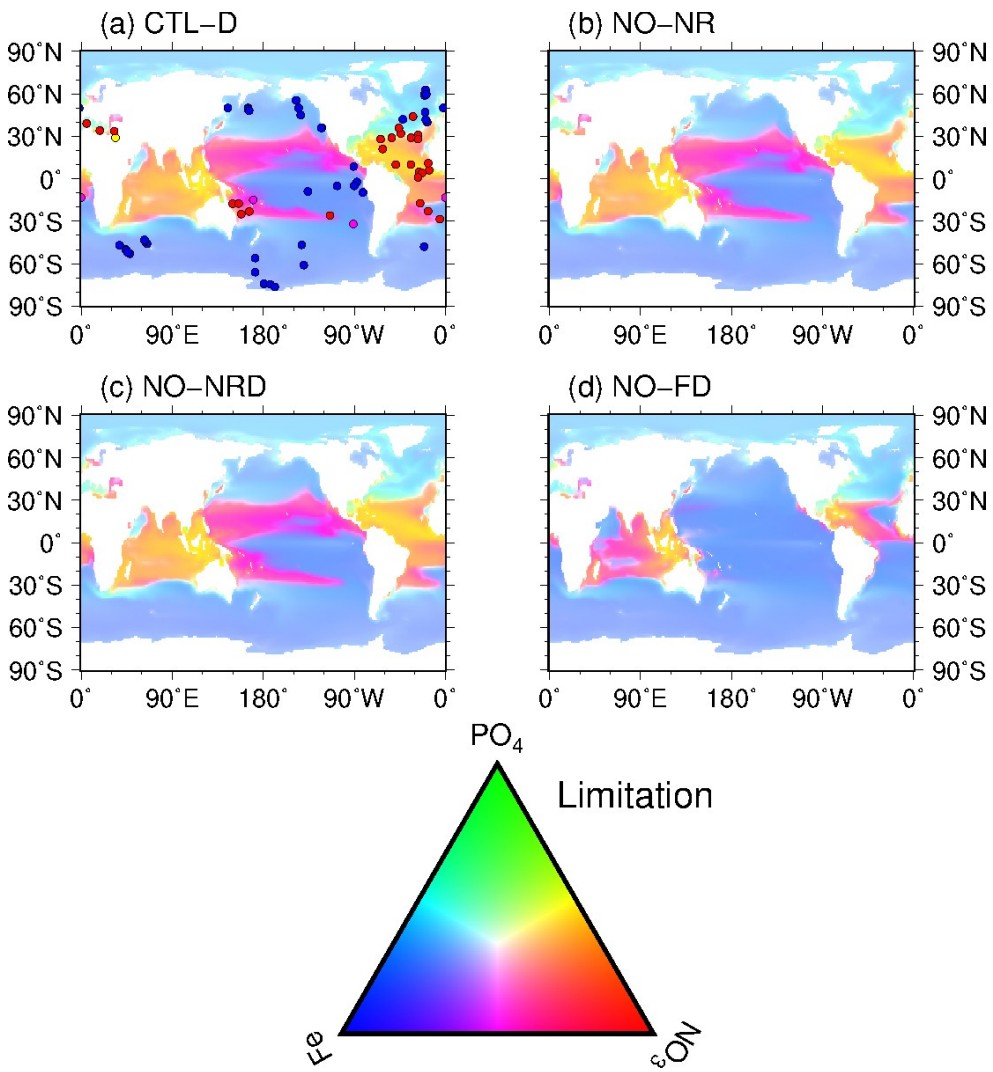

**Figure 15**

Limiting nutrient map for phytoplankton for (a) CTL-D, (b) NO-NR, (c) NO-NRD, and (d) NO-FD. Shading indicates limiting nutrient(s), e.g., red: N limitation, blue: Fe limitation, green: P limitation, magenta: N and Fe limitation, cyan: Fe and P limitation, and yellow: P and N limitation (see bottom color triangle). Circles in (a) represent observed limiting nutrients from nutrient addition experiments (Moore et al., 2013).

Here, we examine model sensitivity against global inputs of both N and Fe into the ocean through atmospheric deposition and river discharge in the preindustrial condition. We note, however, these two types of nutrient input have increased significantly since the preindustrial era because of human activities (Duce et al., 2008; Seitzinger et al., 2010; Krishnamurthy et al., 2010). In addition, ongoing nutrient input increase can lead to future increase in biological production, which might partly negate the production decrease driven by global warming. Conversely, the resultant





increase in export of OM would accelerate warming-induced ocean acidification and deoxygenation in subsurface waters,
leading to major environmental pressures. Thus, the combined effects of global warming and anthropogenic nutrient input
on ocean biogeochemical cycles should be explored in the future.

### 3.2.3. Sensitivity of riverine nitrogen

The coupling of land and ocean ecosystems via riverine nitrogen is one of the new features of MIROC-ES2L, and the
potential impact of the process on ocean biogeochemistry has already been examined and discussed in Sect. 3.2.2. Here,
we examine the response of river nitrogen loading itself against anthropogenic forcing by comparing the results of the
CTL, HIST-NOLUC, and HIST simulations.

As already mentioned in Sect. 3.1.3, the global flux of riverine nitrogen input into the ocean is simulated at 17.5 TgN
yr[-1] in the CTL experiment (Table 4), and the flux is almost doubled in the 2000s at 33.9 TgN yr[-1] in the HIST run. This
number is larger than previous estimates of 19–25 TgN yr[-1] for the present-day condition (Smith et al., 2003; Mayorga et
al., 2010; Dumont et al., 2005). This overestimation might be caused by the inability of the model to simulate all forms of
nitrogen in rivers. For example, the model simulates only dissolved inorganic nitrogen (DIN) flux; thus, the expected
nitrogen flux with non-DIN forms (e.g., dissolved organic and particulate matter) might be partly imposed on the DIN flux
in the simulations. Indeed, global total nitrogen flux, including DIN, dissolved organic nitrogen, and particulate nitrogen is
estimated at 37–66 TgN yr[-1] (Beusen et al., 2016; Mayorga et al., 2010; Boyer et al., 2006; Seitzinger et al., 2005), which
is closer to the result of MIROC-ES2L.

Another possible reason for the above overestimation is precipitation bias and resultant overestimation of BNF on land.
As mentioned in Sect. 3.1.4, the model has positive precipitation bias on land, particularly in arid/desert regions (Fig. 6).
As the scheme for natural BNF flux employed in MIROC-ES2L is modeled to be controlled by the actual
evapotranspiration rate (Cleveland et al., 1999), the precipitation bias in arid regions could easily lead to overestimation of
the BNF flux and produce a resultant increase of riverine nitrogen loading. This is also evident when decomposing the
global riverine flux into river basins and comparing the findings with a previous study by Dumont et al. (2005) (Fig. 16).
MIROC-ES2L overestimates the DIN fluxes of large rivers such as the Amazon, Mississippi, and Yangtze rivers, even in
the CTL experiment where all anthropogenic forcings are fixed at preindustrial levels. This suggests the necessity of
improvement of the baseline flux of riverine nitrogen in the model. For more in-depth discussion, it will be necessary to
simulate explicitly the organic and particulate nitrogen fluxes in rivers, and it might be necessary to simulate the explicit
sedimentary and chemical-reaction processes in freshwater and coastal zone systems.

Although bias exists in the magnitude of riverine nitrogen flux both globally and locally, we confirm the model capable
of capturing qualitatively the changes in riverine nitrogen flux during the historical period. In Fig. 16, the difference
between the results of CTL and HIST-NOLUC mainly reflect the change induced by nitrogen deposition (and historical
climate change) (Table 2), and the model demonstrates that deposition has increased N fluxes in many rivers. In addition,
the difference between HIST-NOLUC and HIST demonstrates the impact of LUC and agricultural management change
(Table 2), and regions that have intensive agriculture within their watersheds (e.g., the basins of the Mississippi, Indus,
Yellow, and Yangtze rivers) are simulated as strongly affected by the forcing change. This simulated trend in the historical
period is qualitatively consistent with previous studies (Gruber and Galloway, 2008). Furthermore, the model simulates
the global riverine flux to be increased by 16.4 TgN yr[-1] in the historical period. This value is quantitatively consistent



with previous estimates, e.g., 16 TgN yr$^{-1}$ by Dumont et al. (2005) for DIN flux, and 18 and 19 TgN yr$^{-1}$ by Beusen et al. (2016) and by Green et al. (2004), respectively, for total N flux.

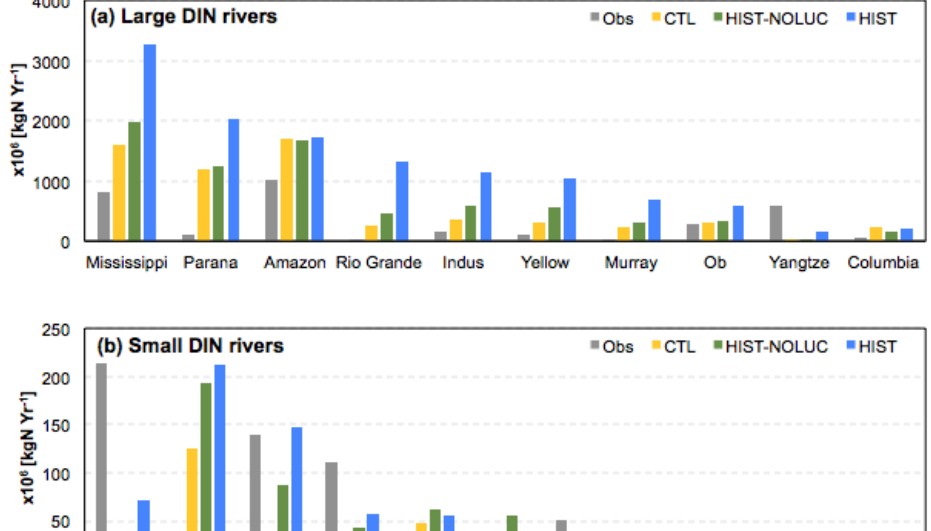


**Figure 16**

Simulated and observed DIN load per river basin: sorted by simulated (a) first 10 largest rivers and (b) second 10 largest rivers. Vertical gray bars represent observations (Dumont et al., 2005). Blue, green, and yellow bars correspond to the results of the HIST, HIST-NOLUC, and CTL experiments, respectively.


### 3.2.4. TCR, AF, and TCRE

We have evaluated the performance of MIROC-ES2L in terms of climate and biogeochemistry using the HIST simulation results (chapter 3.1), and we have explored how the model responds to forcing by comparing the results of different simulations (chapter 3.2). Here, the model sensitivity of the global climate–carbon cycle against $CO_2$ increase is 1025 analyzed by calculating TCR, AF, and TCRE from the results of the 1PPY, 1PPY-BGC, and 1PPY-RAD experiments (see Sect. 2.2.2 for the method). These quantities summarize the total performance of the climate, carbon cycle, and climate–carbon cycle system in the models, which enables us to compare them with existing ESMs.

The TCR, AF, and TCRE derived from the 1PPY simulation are displayed in Table 5 and Fig. 17. The TCR of MIROC-ES2L is 1.5 K, which is lower than the multimodel mean of the CMIP5 ESMs but within the range of spread (1.8 ± 0.5 K; 1030 Gillet et al., 2013). Compared with our previous ESM (i.e., MIROC-ESM; Watanabe et al., 2011), the TCR has reduced by 32% because of the replacement of the physical core of the ESM from the MIROC3-based model to that of MIROC5 (Watanabe et al., 2010). The value of AF, which is a quantity that characterizes the carbon cycle response in an ESM but is dependent on TCR, was simulated at 0.61 in MIROC-ESM. This value is reduced to 0.52 in MIROC-ES2L, i.e., the new





model has a stronger carbon sink than the previous version. The value of AF in the new model is of similar magnitude to

the CMIP5 model average (0.53 ± 0.06; Gillet et al., 2013). The lowered TCR and the moderate AF cause the new model

to have moderate TCRE (1.3 K EgC$^{-1}$), which is smaller than that of the CMIP5 model average (1.6 ± 0.5 K EgC$^{-1}$) by

19%. Using TCRE, we can approximate the value of CE until the global temperature exceeds a specific mitigation target;

CE for the 2°C warming target should be approximately 1540 PgC for MIROC-ES2L, 910 PgC for MIROC-ESM, and

950–1820 PgC for the CMIP5 models.


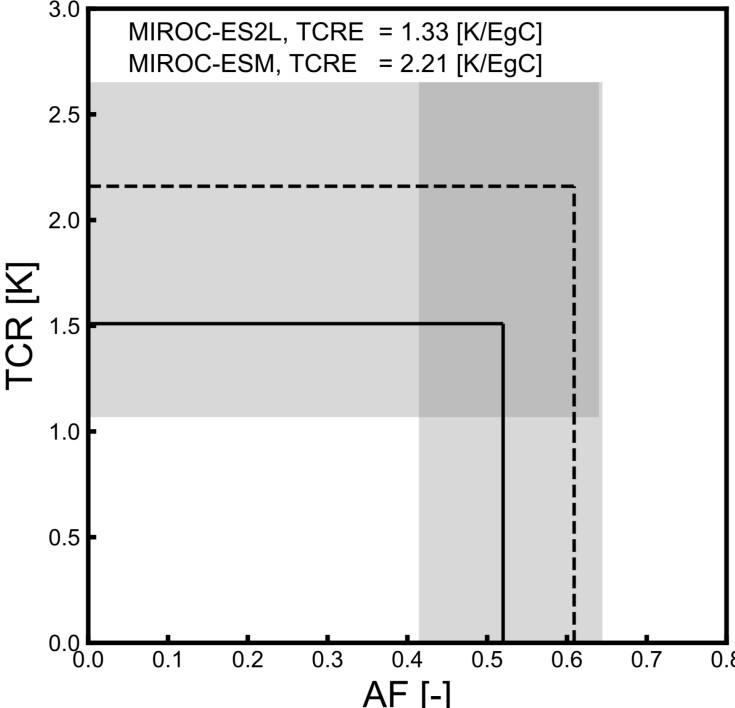

**Figure 17**

Transient climate response (TCR), cumulative airborne fraction (AF), and transient climate response to cumulative

carbon emission (TCRE) for MIROC-ES2L (solid black lines), MIROC-ESM (dashed black lines; Watanabe et al., 2011;

Gillett et al., 2013), and CMIP5-ESMs (gray shading; Gillett et al., 2013) at the doubled CO$_2$ level in the 1PPY

experiment. Vertical and horizontal widths of the shaded areas represent CMIP5 multimodel mean ± 1.65σ. TCRE is

equivalent to (TCR × AF)/CA, where CA is the carbon increase in the atmosphere (which is a constant of approximately

600 PgC in concentration-driven simulations).




**Table 5**

Comparison of TCR, AF, and TCRE between MIROC-ES2L, MIROC-ESM, MIROC5.2, and CMIP5 ESMs in the
1PPY simulation. For MIROC-ES2L, both TCR and AF are calculated based on 20-year means of T2, CL, and CO
centered on the 70th year of the 1PPY simulation (i.e., the time when the $CO_2$ concentration is doubled from the
preindustrial condition), and TCRE is calculated based on TCR and AF. Numbers for the CMIP5-ESMs were obtained
from Gillett et al. (2013) and are presented as the multimodel mean $\pm 1\sigma$.

| | TCR (K) | AF (–) | TCRE (K EgC$^{-1}$) |
|---|---|---|---|
| MIROC-ES2L (This study) | 1.5 | 0.52 | 1.3 |
| MIROC-ESM (Watanabe et al., 2011; Gillett et al., 2013) | 2.2 | 0.61 | 2.2 |
| MIROC5.2 (Tatebe et al., 2018) | 1.6 | – | – |
| CMIP5 (Gillett et al., 2013) | 1.8 ± 0.5 | 0.53 ± 0.06 | 1.6 ± 0.5 |


To further explore the reason why AF is lowered in MIROC-ES2L, the strengths of the carbon cycle feedbacks were
analyzed using the 1PPY-BGC and 1PPY-RAD simulation results (Table 6), and the findings were compared with the
CMIP5 ESMs (Arora et al., 2013). The strength of $CO_2$–carbon feedback ($\beta$) of land is simulated at 0.52 PgC PgC$^{-1}$, which
is slightly higher than the CMIP5 model average (0.43 ± 0.21 PgC PgC$^{-1}$) and larger than that of MIROC-ESM by 48%.
The strength of oceanic $CO_2$–carbon feedback in the CMIP5 ESMs displays less spread among the models (0.38 ± 0.03
PgC PgC$^{-1}$) and the result of MIROC-ES2L is within the spread (0.35 PgC PgC$^{-1}$). The absolute magnitude of the climate–
carbon feedback ($\gamma$) for land and ocean in MIROC-ES2L is −71 and −4.5 PgC K$^{-1}$, respectively, both of which are less
negative than the result of MIROC-ESM by 20% for land and 63% for ocean. Consequently, the land $\gamma$ in MIROC-ES2L
is within the range of the CMIP5 ESMs (−58 ± 29 PgC K$^{-1}$), while the ocean $\gamma$ is slightly larger than the upper range of the
CMIP5 ESMs (−7.8 ± 2.9 PgC K$^{-1}$).

As the quantities $\beta$ and $\gamma$ have different units, it is difficult to conclude which feedback process contributes most to the
AF change. To compare them with the same unit, we used the quantity "u" proposed by Gregory et al. (2009). This
quantity, which is defined as $u_\beta = \beta$ and $u_\gamma = \gamma \times T/CA$, has the unit PgC PgC$^{-1}$, and it can relate the carbon cycle feedback
parameters to AF, as $AF = 1/(1 + u_{\beta L} + u_{\beta O} + u_{\gamma L} + u_{\gamma O})$ (see Appendix E for the derivation). When comparing the u
quantities of MIROC-ES2L with the CMIP5 models (Fig. 18), it is evident that the ocean component of MIROC-ES2L is
less sensitive than the previous model for both $CO_2$–carbon and climate–carbon feedbacks. These two changes almost
counteract each other; thus, the ocean component does not explain the reduced AF in the new model (Table 5, Fig. 17).
For land, the climate–carbon feedback ($u_\gamma$) in MIROC-ES2L is intermediate, while MIROC-ESM was one of the most





sensitive models of the CMIP5 ESMs. In addition, the magnitude of the land $CO_2$–carbon feedback ($u_\beta$) is increased from
        MIROC-ESM to MIROC-ES2L by 48% ($u_\beta = \beta$). Therefore, the land component is the main cause of AF change, making
        the magnitude of both the $CO_2$–carbon and the climate–carbon feedbacks increasingly positive, i.e., strengthening the land
        carbon sink.

**Table 6**

        Comparison of $CO_2$-carbon and climate–carbon feedback parameters between MIROC-ES2L, MIROC-ESM, and the
        CMIP5 ESMs. As presented in Arora et al. (2013), TCR, AF, and TCRE are calculated at the time when $CO_2$
        concentration is quadrupled from the preindustrial condition (i.e., the 140th year in the 1PPY simulation) by taking the
        anomaly from the CTL run. Numbers of CMIP5 ESMs were obtained from Arora et al. (2013) and are presented as the
multimodel mean ± 1σ.

|  | β land (PgC PgC⁻¹) | β ocean (PgC PgC⁻¹) | γ land (PgC K⁻¹) | γ ocean (PgC K⁻¹) |
|---|---|---|---|---|
| MIROC-ES2L (This study) | 0.52 | 0.35 | −71 | −4.5 |
| MIROC-ESM (Watanabe et al., 2011; Arora et al., 2013) | 0.35 | 0.39 | −89 | −12 |
| CMIP5 (Arora et al., 2013) | 0.43 ± 0.21 | 0.38 ± 0.03 | −58 ± 29 | −7.8 ± 2.9 |

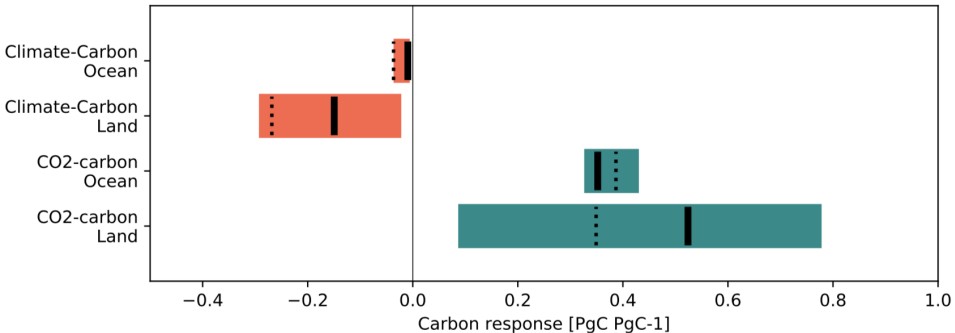

**Figure 18**

        Comparison of strength of $CO_2$–carbon and climate–carbon feedbacks between MIROC-ES2L and the CMIP5 models
        evaluated using the 1PPY, 1PPY-BGC, and 1PPY-RAD experiments. Vertical solid and dotted black bars represent
        MIROC-ES2L and MIROC-ESM, respectively, and the horizontal bars represent the range of the CMIP5 ESMs (mean ±
        1.65σ). To compare the two types of feedback strength with the same unit, land and ocean carbon storage change were
both normalized by dividing the atmospheric carbon change, which corresponds to the "u" quantity proposed by Gregory


et al. (2009): CE = CA (1 + $u_\beta$ + $u_\gamma$), where $u_\beta$ = β, $u_\gamma$ = γ × α. If u > 0 (u < 0), the feedback sign is negative (positive). The calculations were based on the anomaly from the CTL run at the time of quadrupled $CO_2$ concentration from the preindustrial condition (i.e., the 140th year of the 1PPY, 1PPY-BGC, and 1PPY-RAD simulations).


## 4. Summary and conclusions

In this study, a new Earth system model (MIROC-ES2L) was developed using a state-of-the-art climate model
(MIROC5.2) as the physical core. This new ESM embeds a terrestrial biogeochemical component with explicit carbon–
nitrogen interaction (VISIT-e) that accounts for the nutrient limitation of nitrogen on plant growth and the resultant change
in the land carbon sink. In addition, the ocean biogeochemical component (OECO2) is largely updated to simulate the
biogeochemical cycles of carbon, nitrogen, phosphorus, iron, and oxygen such that oceanic primary productivity in the
model is now controlled by multiple nutrient limitations. As a new challenge, land and ocean nitrogen cycles were coupled
via river discharge processes; thus, marine ecosystem productivity can now be controlled by riverine nitrogen input.
Furthermore, iron-related processes such as emission, atmospheric transport, deposition, and utilization in the marine
ecosystem are newly included to represent the micronutrient limitation on phytoplankton productivity. This is necessary
for reproducing the HNLC regions and for simulating ecosystem variability in response to external iron inputs.

To evaluate the performance of the new model, a historical simulation following CMIP6 protocols and forcing datasets
was performed for the 1850–2014 period, and the results were compared with observation-based products. We confirmed
the model reasonably reproduces the global changes in net TOA radiation balance, SAT, SST and upper-ocean
temperature. Considering the few biophysical feedbacks on climate in the model, the good performance in simulating the
physical fields is inherited from the original climate model (MIROC5.2), although persistent problems remain such as the
warm bias in the Southern Ocean, as found in other climate models. Global carbon and nitrogen budgets in the historical
simulation were also examined and discussed by comparing the results with existing studies. It was confirmed that the
model could successfully capture the historical trends of global biogeochemical budgets. The spatial distributions of
fundamental variables of the land carbon cycle were also assessed by comparison with observation-based products, and
the model showed reasonable patterns for primary productivity, forest carbon, and SOC. The spatial patterns of oceanic
macro- and micronutrients, total inorganic carbon, alkalinity, oxygen, and primary productivity were all confirmed
captured well in the historical simulation.

To assess the global climate–carbon cycle feedback in MIROC-ES2L, a sensitivity analysis was performed in which
atmospheric $CO_2$ concentration was prescribed to increase by 1% $yr^{-1}$. Then, the values of TCR, AF, and TCRE were
calculated and compared with the CMIP5 ESMs. TCR in the new model is reduced to 1.5 K, which is approximately 70%
of the previous model used for CMIP5, through the replacement of the physical core from the MIROC3-based model to
that of MIROC5.2. AF is also reduced by 15%. Further feedback analysis of the carbon cycle revealed that most of the AF
reduction should be attributable to the intensified land carbon sink in the new model, resulting in a level of AF that is close
to the average of the CMIP5 ESMs. TCRE, which is a quantity that aggregates the response of the entire climate–carbon
cycle against anthropogenic $CO_2$ emissions, is 1.3 K $EgC^{-1}$ in MIROC-ES2L. This is reduced from value in the model





used for CMIP5 by 32% and it is slightly smaller than the multimodel mean of the CMIP5 ESMs. Thus, MIROC-ES2L

might be an "optimistic" model in terms of simulating global climate and carbon cycle change, considering that some CMIP6-class models are likely to have higher climate sensitivity (Voosen et al., 2019). To discuss whether the climate and carbon cycle sensitivities in the model are realistic, it will be necessary to perform multimodel comparisons on feedback strengths using CMIP6 ESMs and to establish constraints on each feedback process based on observation (e.g., Wenzel et al., 2016).

In the new model, the terrestrial nitrogen cycle processes and the interaction with the carbon cycle are modeled explicitly. By performing several types of simulation, it was demonstrated clearly that agricultural management such as fertilizer application has changed the carbon cycle (GPP) in the historical period, suggesting that the nitrogen cycle in the model actually acts on the land carbon cycle. The model simulated change in the total land carbon during 1850–2014 at 44 PgC, which is within the estimated range of Le Quéré et al. (2018). However, historical terrestrial carbon change is highly

uncertain because the change is processed by multiple responses against the external forcing of $CO_2$, LUC, and climate change, each of which has its own estimation uncertainty. Thus, as performed in this study, decomposition of the impact of these forcings in historical simulations and in multimodel comparisons would be helpful in specifying the processes that produce the large simulation spread of the land carbon budget among the ESMs. Furthermore, although we confirmed the nitrogen cycle actually acts on the carbon cycle in the model, this study did not quantified by how much the soil nutrient

deficit could down-regulate plant growth and reduce the natural carbon sink. For this, a sensitivity analysis associated with the carbon–nitrogen interaction is planned in CMIP6 (Jones et al., 2016), and the multimodel comparison study will reveal the strength of the carbon–nitrogen feedback in MIROC-ES2L relative to other CMIP6-class ESMs.

In the new model, the ocean nitrogen cycle is modified to be an open system and thus the model can reflect the influences of external sources of nitrogen inputs via atmospheric deposition and river discharge. Our sensitivity analyses

under preindustrial condition suggested minor contributions of these two external sources to global primary productivity. However, regions in which primary productivity is constrained by nitrogen availability showed a strong positive NPP response to the relaxation of nitrogen limitation. It accelerates the use of other nutrients within the marine ecosystem in such regions and it reduces iron and phosphorus availability in other regions. Furthermore, by switching on the process of iron deposition into the ocean, the model showed an increase of approximately 7% in GPP under the preindustrial

condition, suggesting that iron input has a relatively stronger impact than nitrogen. Coupling of iron cycle processes in the model led to successful reproduction of HNLC regions, and it will enable the model to project future biogeochemical changes induced by anthropogenic iron emissions associated with the use of fossil fuels and biomass burning. We note, however, as an atmospheric chemistry module is not included in MIROC-ES2L, the atmospheric chemical reaction of iron-containing aerosols is ignored and the iron solubility to seawater is simply assumed constant. Considering the

relatively strong impact of iron deposition on marine primary productivity in the model, we need further detailed evaluation and modification of the iron cycle processes in terms of both aerosol transport and marine biogeochemical responses.

In addition to such improvements in terms of the iron cycle, other factors should be improved/extended in the ESM for future simulation study. First, a freshwater biogeochemistry module is required. In the present model, the chemical form

of riverine nitrogen is assumed inorganic, but actual river flow contains OM and particulate matter that undergo biogeochemical processing during transport. Thus, inclusion of the transport of organic/inorganic matter and the modeling of freshwater biogeochemistry might be necessary. The sensitivity analysis that showed relatively strong impact of riverine nitrogen on regional marine ecosystem productivity supports this conclusion. Second, MIROC-ES2L can simulate



natural emissions of nitrous oxide; however, the emissions did not change the radiative balance of the atmosphere in this study. Nitrous oxide is one of the strongest greenhouse gases and it has a long lifetime. However, as diagnosed in this study, future nitrous oxide emissions could be controlled by land use and agriculture, as well as climate change. Therefore, full coupling of the nitrous oxide cycle with other associated atmospheric chemical processes should be incorporated in the next-generation ESM, together with the methane cycle, as suggested in previous studies (e.g., Collins et al., 2018). Third, a mechanistic model for the denitrification process in ocean sediment should be included in a future model. The present model simulates only the denitrification rate of water column, and the flux from sediment is likely imposed on the water-column denitrification. As the timescale of the sedimentary process is likely longer than that of water-column denitrification, explicit modeling of sedimentary denitrification will be important, particularly for long-term simulations over timescales of millennia. Finally, we partly demonstrated the importance of external sources of nutrients for marine productivity, although its evaluation was performed under the preindustrial condition. As anthropogenic nutrient inputs under that condition are much smaller than under the present-day condition and they could be amplified or mitigated in the future, a similar simulation study should be undertaken for present-day and future conditions.

ESMs represent powerful tools with which to investigate interactions between climate, biogeochemistry, and human activities, and they have facilitated climate projections and explorations of future emissions of greenhouse gases for achieving climate change mitigation goals. Such models are also valuable for examining how Earth system components might respond to different levels of mitigation policies/scenarios spanning from the business-as-usual scenario to one employing intensive measures such as geoengineering techniques. Furthermore, state-of-the-art ESMs can now partly simulate environmental problems on Earth that are becoming evident/doubted in association with climate change, e.g., acidification and hypoxia in the ocean, global nitrogen loading, air pollution, and habitable zone changes in ecosystems. ESMs can simulate such problems and their interactions in a holistic and consistent manner. Such simulations have potential to elucidate sustainable ways to mitigate climate change with less environmental stress. To support such applications, further efforts should be made to evolve ESMs and to constrain model performance in collaboration with observation studies.

## 5. Appendices

### Appendix A. Land ecosystem/biogeochemical component

#### A.1 Nitrogen cycle

The structure of carbon and nitrogen compartments and the flux calculations in VISIT-e mostly follow the original version of the model (Ito and Inatomi, 2012a). For N cycle and LUC processes, some major changes were brought to VISIT-e to couple the model with MIROC-ES2L; the details are described below.

#### A.1.1. N compartment structure in VISIT-e

Terrestrial N dynamics in VISIT are simulated based on three major compartment groups of N storage: vegetation N ($N_{VEG}$), soil organic matter ($N_{SOM}$), and soil inorganic matter ($N_{IOM}$). The component $N_{VEG}$ is composed of canopy N ($N_{CAN}$) and storage N ($N_{STG}$):

$N_{VEG} = N_{CAN} + N_{STG}.$





The mass conservation equations for $N_{CAN}$ and $N_{STG}$ are as follows:

$$dN_{CAN}/dt = FN_{SBNF, CAN} + FN_{UPTK, CAN} + FN_{RALC} - FN_{MORT, CAN}, \tag{A1}$$

$$dN_{STG}/dt = FN_{SBNF, STG} + FN_{UPTK, STG} + FN_{WTHD} - FN_{MORT, STG}, \tag{A2}$$

where $FN$ represents nitrogen flux, and the subscripts SBNF, UPTK, RALC, WTHD, and MORT represent symbiotic biological N fixation, N uptake by plants, reallocation of storage N to the canopy, withdrawal of canopy N to storage, and loss of N by mortality, respectively. In this study, biological N input into vegetation (represented by $FN_{SBNF}$) is modified from the original model; the detail is described in Sect. A.1.2.

The component $N_{SOM}$ is composed of the three nitrogen pools of litter ($N_{LIT}$), humus ($N_{HUM}$), and microbes ($N_{LIT}$):

$$N_{SOM} = N_{LIT} + N_{HUM} + N_{MCR}. \tag{A3}$$

The N conservation equations for the pools are as follows:

$$dN_{LIT}/dt = FN_{MORT, CAN} + FN_{MORT, STG} + FN_{NBNF} - FN_{HUMF} - FN_{MNRL, LIT}, \tag{A4}$$

$$dN_{HUM}/dt = FN_{HUMF} + FN_{MORT, MCR} - FN_{MNRL, HUM}, \tag{A5}$$

$$dN_{MCR}/dt = FN_{IMBL} - FN_{MORT, MCR}, \tag{A6}$$

where subscripts NBNF, HUMF, MNRL, and IMBL represent nonsymbiotic BNF, humification of litter, mineralization of litter/humus, and immobilization by microbes, respectively. The components $FN_{NBNF}$ and $FN_{HUMF}$ are new components of flux, which are described in Sect. A.1.2 and Sect. A.1.3, respectively.

The inorganic nitrogen is assumed to consist of N pools of $NH_4^+$ ($N_{NH4}$) and $NO_3^-$ ($N_{NO3}$):

$$N_{IOM} = N_{NH4} + N_{NO3}. \tag{A7}$$

The budget equation for $N_{NH4}$ is as follows:

$$dN_{NH4}/dt = FN_{DEPO, NH4} + FN_{FRTL, NH4} + FN_{MNRL, LIT} + FN_{MNRL, HUM}$$
$$- FN_{UPTK, NH4} - FN_{IMBL} - FN_{N2ON} - FN_{NTRF} - FN_{NH3V} - FN_{ALOS, NH4}, \tag{A8}$$

where subscripts DEPO, FRTL, N2ON, NTRF, NH3V, and ALOS represent deposition, fertilizer, N₂O emission of nitrification process, nitrification of $NH_4^+$, $NH_3$ volatilization, and abiotic N loss, respectively.

The budget equation for $N_{NO3}$ is as follows:

$$dN_{NO3}/dt = FN_{DEPO, NO3} + FN_{FRTL, NO3} + FN_{NTRF}$$
$$- FN_{UPTK, NO3} - FN_{N2OD} - FN_{N2} - FN_{LECH} - FN_{ALOS, NO3}, \tag{A9}$$

where subscripts N2OD and N2 represent N₂O and N₂ emissions in the denitrification process, respectively and LECH presents N leaching.

In the above two equations, $FN_{DEPO}$ and $FN_{FRTL}$ are forced by external datasets, while $FN_{ALOS}$ is the process newly introduced in this study, which is described in Sect. A.1.4.

**A.1.2. Biological N fixation**





BNF is calculated based on the actual evapotranspiration rate (Cleveland et al., 1999). In the original version of VISIT, all nitrogen fixed through BNF ($FN_{BNF}$) was assumed available for plants. As this assumption makes vegetation in the

1250 model less dependent on soil nutrient availability, the model is modified in that only a portion of BNF-N is made directly available for plant. For this, $FN_{BNF}$ is decomposed into symbiotic BNF ($FN_{SBNF}$) and nonsymbiotic BNF ($FN_{NBNF}$):

$$FN_{BNF} = FN_{SBNF} + FN_{NBNF} \tag{A10}$$

and

$$FN_{SBNF} = \alpha_{SBNF} \times FN_{BNF}, \tag{A11}$$

$$FN_{NBNF} = (1 - \alpha_{SBNF}) \times FN_{BNF}, \tag{A12}$$

where $\alpha_{SBNF}$ is the portion of N of symbiotic BNF. Here, $\alpha_{SBNF}$ is assumed as 0.5 as the landscape-level parameter. Nitrogen fixed by the symbiotic process is used directly by plants, while N fixed by nonsymbiotic microbes is assumed to directly form part of the litter. The BNF in cropland is modeled differently, as shown in Sect. A.2.3.

**A.1.3. Mineralization, humification, and immobilization**

The mineralization rate of litter is same as that in the original version, and it is calculated as follows:

$$FN_{MNRL, LIT} = N_{LIT} \times (FC_{MNRL, LIT}/C_{LIT}), \tag{A13}$$

where $FC_{MNRL, LIT}$ is the C mineralization rate of litter and $C_{LIT}$ is the amount of C in the litter pool.

The humus N mineralization rate is similar to that of litter but it is modified to be dependent on the humus CN ratio

($CN_{HUM}$):

$$FN_{MNRL, HUM} = N_{HUM} \times (FC_{MNRL, HUM}/C_{HUM}) \times (1 - f_{CN}(CN_{HUM})) \tag{A14}$$

and

$$f_{CN}(CN_{HUM}) = S_{min} \times \exp((\log S_{max} - \log S_{min})/(R_{max} - R_{min}) \times (CN_{HUM} - R_{min})). \tag{A15}$$

Here, $S_{max}$ and $S_{min}$ are the maximum and minimum fractions of mineralized N that eventually move to the inorganic N

pool ($N_{NH4}$), respectively. $R_{max}$ and $R_{min}$ are the maximum and minimum CN ratios in the humus pool, respectively. The term $1 - f_{CN}(CN_{HUM})$ controls the humus CN ratio to be between $R_{max}$ and $R_{min}$, by accelerating humus N mineralization under a lower CN ratio and decreasing it under a higher CN ratio. Here, the values of $S_{max} = 0.95$ and $S_{min} = 0.05$ are assumed, and $R_{max}$ and $R_{min}$ are set to the values of 40 and 10, respectively.

Immobilization rate is simplified in VISIT-e and it is modeled as a function of the mineralization rate of litter N,

depending on the CN status in the humus:

$$FN_{IMBL} = FN_{MNRL, LIT} \times f_{CN}(CN_{HUM}). \tag{A16}$$

Thus, N immobilization is accelerated if the humus has a high CN ratio and it decreases under a lower CN condition.

N flux by humification (N flow from litter to humus, $FN_{HUMF, LIT}$) is newly introduced in VISIT-e and it is modeled as follows:

$$FN_{HUMF, LIT} = N_{LIT} \times (FC_{HUMF, LIT}/C_{LIT}), \tag{A17}$$

where $FC_{HUMF, LIT}$ is the rate of C flux in the humification process, which is simulated in the C cycle part of the model.





### A.1.4. Abiotic N loss

Abiotic N loss from soil ($FN_{ALOSS, NH4}$ and $FN_{ALOSS, NO3}$) is newly introduced in VISIT-e to prevent infinite N
accumulation in deserts and arid regions, where much N removal thorough biotic and hydrological processes cannot be
expected. This new scheme is based on the findings of McCalley and Sparks (2009) and it is modeled as follows:

$$FN_{ALOSS, NH4} = S_{ALOSS} \times \exp(K_{ALOSS}(T_{sfc} - 50)) \times N_{NH4}, \tag{A18}$$

$$FN_{ALOSS, NO3} = S_{ALOSS} \times \exp(K_{ALOSS}(T_{sfc} - 50)) \times N_{NO3}, \tag{A19}$$

where $S_{ALOSS}$ is a specific rate of abiotic loss that is set to the value of $7.26 \times 10^{-3}$ (ngN m$^{-2}$ s$^{-1}$) (Schaeffer et al., (2003)),
and $K_{ALOSS}$ is a constant to normalize the rate at 50°C. Here, the emitted gas is assumed an inert form of N.

### A.1.5. N limitation on plant productivity

To simulate soil nutrient (soil inorganic nitrogen) control on plant growth, VISIT-e is modified from the original model
as follows.

First, the photosynthetic capacity ($P_{CSAT}$), which used to be given as the fixed parameter, is modified such that it is
controlled by N concentration in the leaf ($N_{FOL}$):

$$P_{CSAT} = K_{PSAT1} \times N_{FOL} + K_{PSAT2} \tag{A20}$$

and

$$N_{FOL} = N_{CAN}/LAI, \tag{A21}$$

where $K_{PSAT1}$ and $K_{PSAT2}$ are the slope and intercept, respectively, of the empirical relationship between $N_{FOL}$ and $P_{CSAT}$,
and $LAI$ is the leaf area index. In this study, the parameters $K_{PSAT1}$ and $K_{PSAT2}$ were obtained from a meta-analysis study of
Kattge et al., (2009). The leaf-level photosynthetic capacity is upscaled using the analytical method of the Monsi–Saeki
theory, assuming a vertically uniform distribution of canopy N.

Second, actual N uptake by plants ($FN_{UPTK}$) is determined by the balance between N demand by plants ($FN_{DMND}$) and
the potential supply from the soil ($FN_{SPPL}$), which allows the model to have a flexible CN ratio in plant organs:

$$FN_{UPTK} = \min\{FN_{SPPL}, FN_{DMND}\}. \tag{A22}$$

Here, $FN_{SPPL}$ is assumed simply as the total amount of inorganic N in soil ($=N_{NH4} + N_{NO3}$). The component $FN_{DMND}$ is
the sum of the demand from plant organs:

$$FN_{DMND} = FN_{DMND, CAN} + FN_{DMND, ROT} + FN_{DMND, STM} \tag{A23}$$

and

$$FN_{DMND, CAN} = (FC_{TRNS, CAN} - FC_{GRSP, CAN})/(C_{CAN}/N'_{CAN}), \tag{A24}$$

$$FN_{DMND, ROT} = (FC_{TRNS, ROT} - FC_{GRSP, ROT})/(R_{ROT}), \tag{A25}$$

$$FN_{DMND, STM} = (FC_{TRNS, STM} - FC_{GRSP, STM})/(R_{STM}). \tag{A26}$$

In the above, $FCs$ represents the carbon flux of translocation of primary production (with subscript TRNS) and the
carbon lost by growth respiration (GRSP). Subscripts CAN, ROT, and STM represent canopy, root, and stem, respectively.





$R_{\text{ROT}}$ and $R_{\text{STM}}$ are fixed parameters used as reference CN ratios in the root and stem, respectively, obtained from White et al., (2000). $N'_{\text{CAN}}$ is the canopy N that maximizes canopy productivity, which is determined numerically by considering the balance between GPP and canopy (foliage) respiration.

**A.2. Land use change**

**A.2.1. Structure of LUC tiles**

LUC forced by external forcing and its impact on land biogeochemistry are simulated with five main types of tile (primary vegetation, secondary vegetation, urban, cropland, and pasture) in each land grid. The same structure of C and N compartments is shared among the tiles and each tile has its own areal fraction in a grid ($f_{\text{LUC}}$):

$$f_{\text{LUC, PV}} + f_{\text{LUC, SV}} + f_{\text{LUC, UR}} + f_{\text{LUC, CR}} + f_{\text{LUC, PS}} = 1 \tag{A27}$$

The crop tile further holds two subtiles and their areal fractions: nitrogen-fixing crops and others.

$$f_{\text{LUC, CR}} = f_{\text{LUC, CRN}} + f_{\text{LUC, CRO}} \tag{A28}$$

where $f_{\text{LUC, CRN}}$ is the areal fraction for the N-fixing crop and $f_{\text{LUC, CRO}}$ is for the others. This subtile-level fraction is used for the estimation of nitrogen fixation by crops (see Sect. A.2.3).

**A.2.2. Product pool and decomposition**

The carbon and nitrogen in biomass removed by crop harvesting and by land use conversion ($P$) are allocated to three product pools with different turnover rates (1 year, 10 years, and 100 years):

$$dM_{\text{PROD, 1yr}}/dt = \varepsilon_{\text{1yr}} \times P - FM_{\text{LUCE, 1yr}}, \tag{A29}$$

$$dM_{\text{PROD, 10yr}}/dt = \varepsilon_{\text{10yr}} \times P - FM_{\text{LUCE, 10yr}}, \tag{A30}$$

$$dM_{\text{PROD, 100yr}}/dt = \varepsilon_{\text{100yr}} \times P - FM_{\text{LUCE, 100yr}}, \tag{A31}$$

where $M_{\text{PROD}}$ is the harvested biomass of C or N stored in the three product pools and $P$ is harvested mass of C or N. Here, $\varepsilon$ is the allocation fraction among the product pools (set in this study as $\varepsilon_{\text{1yr}} = 0.5$, $\varepsilon_{\text{10yr}} = 0.45$, and $\varepsilon_{\text{100yr}} = 0.05$). $FM_{\text{LUCE}}$ represents the volatilization rates of carbon (as $CO_2$) or nitrogen (as an inert form) from the three pools, which are

1340 calculated as follows:

$$FM_{\text{LUCE, 1yr}} = K_{\text{LUCE, 1yr}} \times M_{\text{PROD, 1yr}}, \tag{A32}$$

$$FM_{\text{LUCE, 10yr}} = K_{\text{LUCE, 10yr}} \times M_{\text{PROD, 10yr}}, \tag{A33}$$

$$FM_{\text{LUCE, 100yr}} = K_{\text{LUCE, 100yr}} \times M_{\text{PROD, 100yr}}, \tag{A34}$$

where $K_{\text{LUCE}}$ is the specific emission rate in each product pool, which is set to reduce the carbon/nitrogen in each pool by

1345 99.9% within 1 year, 10 years, and 100 years.

**A.2.3. LUC status-driven impact on biogeochemistry**



Even if the areal fraction of each land use tile were fixed in a simulation, there could still be impacts of land use on land biogeochemistry, referred to here as the status-driven impact. This impact is specific to each tile and it is summarized as follows:

(1) prohibition of plant growth on an urban tile;

(2) increased mortality of plants by grazing pressure on pasture tiles, assuming a 20% increase of mortality rate for foliage;

(3) annual crop harvesting on crop tiles (assuming 10% of foliage is harvested) and loss of C and N from the product pools;

(4) nitrogen fixation by N-fixing crop on crop tiles.

For (4), the total BNF rate on crop tiles ($FN_{\text{SBNF}}$) is modeled as follows:

$$FN_{\text{SBNF}} = FN_{\text{SBNF, CRO}} \times f_{\text{LUC, CRO}} + FN_{\text{SBNF, CRN}} \times f_{\text{LUC, CRN}}, \tag{A35}$$

where $FN_{\text{SBNF, CRO}}$ represents the rate of nitrogen fixation on non-N-fixing crop tiles, which is assumed the same as that in natural vegetation. $FN_{\text{SBNF, CRN}}$ is the rate of nitrogen fixation on N-fixing crop tiles, which is calculated simply to satisfy a fixed ratio of BNF-derived N to all N taken up by N-fixing crops (=0.66; from Herridge et al. (2008)).

### A.2.4. LUC transition-driven impact on biogeochemistry

When the areal fractions of tiles are made to change following the forcing dataset, the apparent mass densities of C and N on a grid can be changed. For example, when a portion of a grid area is converted from category X to category Y in a year, the mass conservation between the "before (t)" and "after (t+1)" on a grid should be as follows:

$$M_X^t \times f_X^t + M_Y^t \times f_Y^t = M_X^{t+1} \times f_X^{t+1} + M_Y^{t+1} \times f_Y^{t+1} + P, \tag{A36}$$

and

$$M_X^t = M_X^{t+1}, \tag{A37}$$

where $M$ is the mass density per unit tile area, subscripts X and Y represent categories of land use type, and superscript t denotes time. By presenting the areal fraction change as $\Delta f$ and the change in apparent mass density in category Y as $\Delta M_Y$, these equations can be written as follows:

$$M_X^t \times f_X^t + M_Y^t \times f_Y^t = M_X^t \times (f_X^t - \Delta f) + (M_Y^t + \Delta M_Y) \times (f_Y^t + \Delta f) + P, \tag{A38}$$

and

$$P = \Delta f \times M_X^t \times K_{\text{HARV}}, \tag{A39}$$

where $K_{\text{HARV}}$ determines the fraction of mass that enters the product pools instead of the tile of category Y. Here, $K_{\text{HARV}}$ is always set to zero for litter and soil pools and $K_{\text{HARV}} = 1$ for vegetation pools in specific transition patterns (e.g., $K_{\text{HARV}} = 1$ if the LUC transition type is urbanization, whereas $K_{\text{HARV}} = 0$ if the LUC conversion is pasture abandonment). By solving the equations for $\Delta M_Y$, we obtain the following:

$$\Delta M_Y = (\Delta f \times (M_X^t - M_Y^t) - P)/(f_Y^t + \Delta f). \tag{A40}$$



If $\Delta M_Y > 0$ ($<0$), the apparent mass density in tile Y is increased (decreased). The changes in apparent mass density lead to mass imbalance of C and N and therefore the storage of both C and N starts to move toward a rebalanced status under the given environmental conditions.





### Appendix B. Ocean ecosystem/biogeochemical component

#### B.1. Governing equations

The ocean ecosystem component (OECO2) embedded within the ocean circulation model is based on nutrient–
1390 phytoplankton–zooplankton–detritus (NPZD) type with four prognostic variables: nitrate (NO3), "ordinary"
nondiazotrophic phytoplankton (Phy), zooplankton (Zoo), and particulate detritus (Det). In addition, phosphate (PO4),
dissolved oxygen (O2), dissolved iron (Fe), nitrous oxide (N2O), and diazotrophic phytoplankton (nitrogen fixers, Diaz)
are included. Biogeochemical tracers associated with the carbon cycle, i.e., dissolved inorganic carbon (DIC), alkalinity
(Alk), calcium carbonate (CaCO3), and calcium (Ca) are also included. Constant (~Redfield) stoichiometry relates the C,
1395 N, P, Fe, and O content of the biological variables and their exchanges with the inorganic variables (NO3, PO4, Fe, O2,
N2O, Alk, and DIC).

Each variable changes its concentration $C$ according to the following equation:

$$\frac{\partial C}{\partial t} = Tr + S, \tag{B1}$$

where $Tr$ represents all transport terms associated with the physical processes, including advection, isopycnal and
1400 diapycnal diffusion, and convection, and $S$ denotes the source minus sink terms that include the surface and bottom fluxes.
Using the variables and parameters listed in Tables B1 and B2, the source minus sink terms for each prognostic variable
can be obtained as follows.

First, the source minus sink term for NO3 $S(NO3)$ is given by the following:

$$S(NO3) = G_{\text{NO3}}(1 - 0.8 R_{O:N} \Gamma_{\text{NO3}} r_{\text{sox}}^{\text{NO3}}) + \text{Dep}_{\text{NO3}} + \text{Riv}_{\text{NO3}}, \tag{B2}$$

where $\text{Dep}_{\text{NO3}}$ ($\text{Riv}_{\text{NO3}}$) represents nitrogen deposition from the atmosphere (riverine input) and

$$G_{\text{NO3}} = (\mu_D \text{Det} + \mu_P^* \text{Phy} + E_z \text{Zoo} - J_o \text{Phy} - u_N J_D \text{Diaz}),$$

$$\Gamma_{\text{NO3}} = \begin{cases} 1 \text{ if NO3} > \text{NO3}_{\text{crit}}, \\ 0 \text{ if NO3} < \text{NO3}_{\text{crit}}, \end{cases}$$

where $J_O$ ($J_D$) is the growth rate of "ordinary" nondiazotrophic (diazotrophic) phytoplankton (see Appendix B2). The
nitrate uptake rate is given by $u_N = \text{NO3}/(k_N^{\text{Diaz}} + \text{NO3})$ (Schmittner et al., 2005). Denitrification (Denit) can be
expressed as follows:

$$\text{Denit} = G_{\text{NO3}}(-0.8 R_{O:N} \Gamma_{\text{NO3}} r_{\text{sox}}^{\text{NO3}}) - P_{\text{N2O}},$$

where $P_{\text{N2O}}$ is the source term of N2O, which is discussed later. The source minus sink terms for Phy and Diaz, i.e.,
$S$(Phy) and S(Diaz), respectively, can be expressed as follows:

$$S(\text{Phy}) = J_o \text{Phy} - \mu_P^* \text{Phy} - m_{\text{Phy}} \text{Phy}^2 - \text{Graze}_{\text{Phy}}, \tag{B3}$$

$$S(\text{Diaz}) = J_D \text{Diaz} - m_{\text{Diaz}} \text{Diaz} - \text{Graze}_{\text{Diaz}}. \tag{B4}$$

The term $S$(zoo) is estimated as follows:

$$S(\text{Zoo}) = \gamma(\text{Graze}_{\text{Phy}} + \text{Graze}_{\text{Diaz}}) - E_z \text{Zoo} - m_{\text{Zoo}} \text{Zoo}^2. \tag{B5}$$

Then, $S$(Det) is given by the following:

$$S(\text{Det}) = (1 - \gamma)\big(\text{Graze}_{\text{Phy}} + \text{Graze}_{\text{Diaz}}\big) + m_{\text{Phy}}\text{Phy}^2 + m_{\text{Diaz}}\text{Diaz} + m_{\text{Zoo}}\text{Zoo}^2$$

$$-\mu_D \text{Det} - \text{Fsed}_{\text{Det}} - \frac{\partial \,\text{Sink}_{\text{Det}}}{\partial z}, \tag{B6}$$

$$\text{Sink}_{\text{Det}} = \begin{cases} w_D\,\text{Det} & \text{if } z < 200\text{m} \\ \text{Sink}_{\text{Det200}}\left(\frac{z}{200}\right)^{0.875} & \text{if } z > 200\text{m} \end{cases},$$

where $\text{Fsed}_{\text{Det}}$ represents the net flux of detritus between the ocean and ocean sediment (Kobayashi and Oka, 2018) and $\text{Sink}_{\text{Det200}}$ is the flux of sinking detritus at the depth of 200 m (Kawamiya et al., 2000).

Using the molar P:N ratio of organic matter, $R_{P:N}$ , and the riverine input of phosphate ($\text{Riv}_{\text{PO4}}$), the source minus sink term for PO4 becomes:

$$S(\text{PO4}) = R_{P:N}\,G_{\text{NO3}} + \text{Riv}_{\text{PO4}}. \tag{B7}$$

As the land ecosystem model cannot simulate the phosphorus cycle, it is assumed that phosphate is brought to the river mouth at a rate to satisfy $\text{Riv}_{\text{NO3}}:\text{Riv}_{\text{PO4}} = 16:1$, similar to the Redfield ratio. The term $S(\text{O2})$ can be estimated as follows:

$$S(\text{O2}) = -\Gamma_{\text{O2}}\,R_{O:N}\,G_{\text{NO3}} + \text{Fsfc}_{\text{O2}}, \tag{B8}$$

$$\Gamma_{\text{O2}} = \begin{cases} 1 & \text{if O2} > \text{O2}_{\text{crit}}, \\ 0 & \text{if O2} < \text{O2}_{\text{crit}}, \end{cases}$$

where $\text{Fsfc}_{\text{O2}}$ is the dissolved oxygen exchange with the atmosphere, according to the OMIP protocol (Orr et al., 2017). The term S(Fe) can be expressed as follows:

$$S(\text{Fe}) = R_{Fe:N}\,G_{\text{NO3}} + \text{Scav} + \text{Dustin} + \text{Sedin} + \text{HTin}, \tag{B9}$$

where Scav represents scavenging (Moore et al., 2004; Moore and Braucher, 2008), Dustin is the iron input from dust, Sedin is the iron input from sediment following both Moore et al. (2004) and Aumont and Bopp (2006), and HTin is the hydrothermal dissolved iron flux following Tagliabue et al. (2010).

The source minus sink term for N2O is linked to the consumption of oxygen during the remineralization of OM (Ilyina et al., 2013):

$$S(\text{N2O}) = r_{N2O}\Gamma_{O2}R_{O:N}(\mu_D\text{Det} + \mu_P^*\text{Phy} + E_Z\text{Zoo}) + Fsfc_{N2O}, \tag{B10}$$

where $Fsfc_{N2O}$ is the N$_2$O exchange with the atmosphere according to Orr et al. (2017).

The source minus sink term for DIC can be expressed as follows:

$$S(\text{DIC}) = R_{C:N}G_{\text{NO3}}(1 - 0.8R_{O:N}r_{\text{sox}}^{\text{NO3}}) - G_{\text{CaCO3}} + \text{Fsfc}_{\text{DIC}}, \tag{B11}$$

where $\text{Fsfc}_{\text{DIC}}$ is the DIC exchange with the atmosphere according to the OMIP protocol (Orr et al., 2017) and $G_{\text{CaCO3}} = Pr_{\text{CaCO3}} - Di_{\text{CaCO3}}$.

Then, $S(\text{Alk})$, $S(\text{CaCO3})$, and $S(\text{Ca})$ can be estimated, respectively, as follows:

$$S(\text{Alk}) = -2G_{\text{CaCO3}} - G_{\text{NO3}}, \tag{B12}$$

$$S(\text{CaCO3}) = G_{\text{CaCO3}}, \tag{B13}$$

$$S(\text{Ca}) = -G_{\text{CaCO3}}. \tag{B14}$$






**B.2. Growth rate of nondiazotrophic and diazotrophic phytoplankton**

To simply evaluate the effect of iron limitation on the growth of "ordinary" nondiazotrophic phytoplankton and diazotrophic phytoplankton (nitrogen fixers), we modify the equations of phytoplankton growth rate by Keller et al. (2012) as follows. First, we estimate the maximum potential growth rate of phytoplankton ($J_O{}^{\max}$) and diazotrophic

plankton ($J_D{}^{\max}$) that depend on temperature ($T$):

$$J_O^{\max} = ae^{(T/T_b)}, \tag{B15}$$

$$J_D^{\max} = c_D \max\left(0, a(e^{(T/T_b)} - 2.61)\right) \tag{B16}$$

(Schmittner et al., 2008).

Once the maximum potential growth rate has been calculated, the realized growth rate of phytoplankton ($J_O$) is then

determined by irradiance ($I$) and the concentrations of NO3, Fe and PO4, while the growth rate of diazotrophic plankton ($J_D$) is determined by irradiance ($I$) and the concentrations of Fe and PO4:

$$J_O = \min\left(J_{OI}, J_O^{\max}\frac{\text{NO}_3}{k_N+\text{NO}_3}, J_O^{\max}\frac{\text{Fe}}{k_{Fe}+\text{Fe}}, J_O^{\max}\frac{\text{PO}_4}{k_P+\text{PO}_4}\right), \tag{B17}$$

$$J_D = \min\left(J_{DI}, J_D^{\max}\frac{\text{Fe}}{k_{Fe}+\text{Fe}}, J_D^{\max}\frac{\text{PO}_4}{k_P+\text{PO}_4}\right). \tag{B18}$$

$J_{OI}$ and $J_{DI}$ in (B17) and (B18) represent the light-limited growth rate of phytoplankton and diazotrophic phytoplankton, respectively, given by $J_{OI} = \frac{J_O^{\max}\alpha I}{\sqrt{(J_O^{\max})^2+(\alpha I)^2}}$ and $J_{DI} = \frac{J_D^{\max}\alpha I}{\sqrt{(J_D^{\max})^2+(\alpha I)^2}}$, where $\alpha = 0.1$ d$^{-1}$ and $I$ is shortwave radiation at each depth (see (14) of Keller et al. (2012)).

**Table B1**.

Model parameters.

| Parameter | Symbol | Value | Unit |
|---|---|---|---|
| Fast recycling term (microbial loop) | $\mu_P^*$ | 0.05 | d$^{-1}$ |
| Excretion of zooplankton | $E_Z$ | 0.03 | d$^{-1}$ |
| Critical NO3 concentration of denitirification | $\text{NO3}_{\text{crit}}$ | 1 | μmol L$^{-1}$ |
| Critical O2 concentration of remineralization | $\text{O2}_{\text{crit}}$ | 4 | μmol L$^{-1}$ |
| Molar O:N ratio | $R_{O:N}$ | 8.625 | N.D. |





| Molar P:N ratio | $R_{P:N}$ | 0.0625 | N.D. |
|---|---|---|---|
| Molar Fe:N ratio | $R_{Fe:N}$ | $4.4167 \times 10^{-5}$ | N.D. |
| Molar C:N ratio | $R_{C:N}$ | 6.625 | N.D. |
| Half-saturation constant for N uptake | $k_N^{Diaz}$ | 0.05 | µmol L$^{-1}$ |
| Phytoplankton mortality rate | $m_{Phy}$ | 0.05 | d$^{-1}$ (µmol L$^{-1}$)$^{-1}$ |
| Diazotroph mortality rate | $m_{Diaz}$ | 0.025 | d$^{-1}$ |
| Zooplankton mortality rate | $m_{Zoo}$ | 0.2 | d$^{-1}$ (µmol L$^{-1}$)$^{-1}$ |
| Assimilation efficiency coefficient | $\gamma$ | 0.75 | N.D. |
| Sinking speed at the depth of 0–200 m | $w_D$ | 5 | m d$^{-1}$ |
| Maximum potential growth rate of nondiazotrophic phytoplankton at 0°C | $a$ | 0.8 | d$^{-1}$ |
| Diazotroph handicap | $c_D$ | 0.5 | N.D. |
| E-folding temperature of biological rates | $T_b$ | 15.65 | °C |
| Half-saturation constants for NO3 uptake | $k_N$ | 0.5 | µmol L$^{-1}$ |
| Half-saturation constant for PO4 uptake | $k_P$ | 0.5 | µmol L$^{-1}$ |
| Half-saturation constant for iron uptake | $k_{Fe}$ | $10^{-3}$ | nmol L$^{-1}$ |

**Table B2**.

Definitions of parameters and variables not mentioned specifically in the text.

| Parameter or variable | Definition | Reference |
|---|---|---|
| $r_{sox}^{NO3}$ | Oxygen-equivalent oxidation of nitrate in suboxic waters (i.e., denitrification) | Equation (A18) in Schmittner et al. (2008) |
| $\mu_D$ | Temperature and O2 dependent rate of detritus remineralization | Equation (A16) in Schmittner et al. (2008) |
| $\alpha$ | Initial slope of P–I curve | Table A1 in Schmittner et al. |





| | | | (2008) |
|---|---|---|---|
| Graze$_{Phy}$ | Grazing rate of zooplankton on nondiazotrophic phytoplankton | Schmitter et al. (2005) | |
| Graze$_{Diaz}$ | Grazing rate of zooplankton on diazotrophic phytoplankton | Schmitter et al. (2005) | |
| $Pr_{CaCO3}$ | Production of calcium carbonate | Schmittner et al. (2008) | |
| $Di_{CaCO3}$ | Dissolution of calcium carbonate | Schmittner et al. (2008) | |
| $I$ | Shortwave radiation at each depth | Equation (14) in Keller et al. (2012) | |
| $r_{N2O}$ | N$_2$O production rate | Broecker and Peng (1982) | |

**Appendix C. Forcing data**

External forcing used for HIST experiment is summarized in Table C1.

**Table C1**.

List of forcing datasets for HIST simulation: categories, variables, and references for the data creation and description of how the datasets are applied in the HIST simulation in MIROC-ES2L.


| Category | Variables | Reference | Treatment in MIROC-ES2L |
|---|---|---|---|
| GHG concentration | CO$_2$, CH$_4$, N$_2$O, CFC11, CFC12, CFC113, CFC114, CFC115, HCFC22, HCFC123, HCFC141b, HCFC142b, HFC32, HFC125, HFC134a, HFC143a, SF$_6$, CCl$_4$, C$_2$F$_6$ | Meinshausen et al. (2017) | Same as Tatebe et al. (2019): given as globally averaged annual concentration |
| Anthropogenic SLCF emission | BC, OC, SO$_2$ | Hoesly et al. (2018) | Same as Tatebe et al. (2019): given as monthly emissions |
| Open biomass burning emission | BC, OC, SO$_2$ | van Marle et al. (2017) | Same as Tatebe et al. (2019): given as monthly emissions |
| Atmospheric chemical composition for aerosol scheme | H$_2$O$_2$, OH radical, NO$_3$ | Precalculated from atmospheric chemistry model CHASER: Sudo et al. (2002) | Same as Tatebe et al. (2019): given as three-dimensional concentration with monthly interval |
| Anthropogenic dissolved iron emission | Dissolved Fe | Biomass burning emission diagnosed from BC emission (van Marle et al., 2017; Ito, 2011); fossil fuel and biofuel emission (Hoesly et al., 2018; Ito et al., 2018) | Given as monthly emission of biomass burning emission and fossil fuel/biofuel emissions |
| Nitrogen deposition | NOy (wet and dry), NHy (wet and dry) | IGAC/SPARC CCMI: http://blogs.reading.ac.uk/ccmi/forcing-databases-in-support-of-cmip6/ | Given as wet plus dry monthly deposition for both NOy and NHy |
| Land use | Status, transition, fertilizer | Ma et al. (2019) | Given as two types of land use status (non-agriculture and |



| | | | agriculture) for energy/hydrology processes; given as transition matrix among five land use types (primary, secondary, urban, crop, and pasture) for biogeochemistry; given as cropland fertilizer |
|---|---|---|---|
| Stratospheric aerosol | Extinction coefficient | Thomason et al. (https://www.wcrp-climate.org/) | Same as Tatebe et al. (2019): monthly vertically integrated extinction coefficients for each radiation band |
| Ozone concentration | $O_3$ | Hegglin et al. (in prep.) | Same as Tatebe et al. (2019): given as three-dimensional concentration with monthly interval |
| Solar | Solar spectral irradiance | Matthes et al. (2017) | Same as Tatebe et al. (2019): given as monthly solar irradiance spectra |

### Appendix D. Diagnosis of atmospheric $CO_2$ concentration

The global carbon budget can be written as follows:

$$CE = CA + CL + CO,$$

where CE is the cumulative emission derived from fossil fuel and industry. CA, CL, and CO represent the changes in carbon amount in the atmosphere, land, and ocean, respectively. When models are forced with prescribed $CO_2$ concentration (CA), both CL and CO are diagnosed in the simulations. By expressing the prescribed CA as $CA^P$, the

budget equation can be described as:

$$CE^D = CA^P + CL + CO, \tag{C1}$$

where $CE^D$ is a diagnosed fossil fuel/industrial carbon emission. This diagnosis of $CE^D$ was used in the analysis of Jones et al. (2013). If we can obtain the prescribed emission ($CE^D$) that is consistent with historical atmospheric $CO_2$ concentration change, we can diagnose $CO_2$ concentration ($CA^D$) as follows:

$$CA^D = CE^P - CL - CO. \tag{C2}$$

For CMIP6, $CE^P$ during 1850–2014 is approximately 403 PgC, and the values of CL and CO in this study were 44 and 163 PgC, respectively. Thus, $CA^D$ in this study was 193 PgC. This is equivalent to the $CO_2$ concentration change of 91 ppmv determined using a conversion factor of 2.12 (PgC ppmv$^{-1}$). Consequently, we can obtain the diagnosed $CO_2$ concentration at the end of simulation (2014) of 376 ppmv. We note the estimate of anthropogenic $CO_2$ emission of fossil

fuel and industry has its uncertainty range, e.g., Le Quéré et al. (2018) estimate the cumulative emission as $400 \pm 20$ PgC for 1850–2014; however, it was not considered in this study. In addition, there is a budget imbalance of 25 PgC in Le Quéré et al. (2018), which was also ignored in this study.

### Appendix E. Feedback parameters of carbon cycle with same unit

As in Appendix D, the global carbon budget can be written as follows:

$$CE = CA + CL + CO. \tag{E1}$$





Following Gregory et al. (2009), this carbon budget equation can relate the feedback parameters of land and ocean to AF. First, following the definition, CL and CO can be expressed by the feedback parameters of $CO_2$–carbon and climate–carbon feedbacks ($\beta$ and $\gamma$, respectively) as follows:

$CL = \beta_L\, CA + \gamma_L\, T,$ (E2)

$CO = \beta_O\, CA + \gamma_O\, T,$ (E3)

where CA is the carbon increase in the atmosphere and T is global temperature change (T). Using Eqs. E1–E3, the global carbon budget equation can be written as follows:

$CE = CA + CA\,(\beta_L + \beta_O) + T\,(\gamma_L + \gamma_O).$ (E4)

Dividing both sides of the equation by CA leads to the following:

$CE/CA = 1 + (\beta_L + \beta_O) + T\,(\gamma_L + \gamma_O)/CA.$ (E5)

Then, we define $T/CA = \alpha$, as used by Friedlingstein et al. (2006) or Arora et al. (2013), and we replace CE/CA by 1/AF (because AF = CA/CE):

$1/AF = 1 + (\beta_L + \beta_O) + \alpha\,(\gamma_L + \gamma_O).$ (E6)

The "u" quantity proposed by Gregory et al. (2009) is $u_{\beta L} = \beta_L$; $u_{\beta O} = \beta_O$; $u_{\gamma L} = \alpha\,\gamma_L$; and $u_{\gamma O} = \alpha\,\gamma_O$. Through replacement with the u terms, Eq. E6 can be expressed as follows:

$1/AF = 1 + u_{\beta L} + u_{\beta O} + u_{\gamma L} + u_{\gamma O},$ (E7)

and thus we obtain the following:

$AF = 1/(1 + u_{\beta L} + u_{\beta O} + u_{\gamma L} + u_{\gamma O}).$ (E8)

As AF has the unit of PgC PgC$^{-1}$, the unit of the u parameters is also dimensionless.



## 6. Author contributions

TH was responsible for the development and description of MIROC-ES2L and VISIT-e, executed the spin-up and experiments, and undertook global analyses of climate–biogeochemistry and the terrestrial analysis. MW, AY, and MN
contributed to the development and description of OECO2, as well as the analysis of ocean biogeochemistry. HT developed MIROC5.2 and supervised the physical modeling and engineering. MA contributed to the DMS emission modeling, preparation of the forcing dataset, and conversion and archiving of the output. RO contributed to the examination of model performance, postprocessing of the output, and analysis of the physical fields. AI[1] contributed to the development of atmospheric iron transport, preparation of iron emission forcing, and its description. DY contributed to
river nitrogen modeling and its analysis. HO contributed to the coupling of OECO2. AI[3] provided the original model VISIT and supervised the modeling and analysis of the terrestrial biogeochemistry. KT supervised the modeling of the terrestrial physical processes. KO supervised and supported the software engineering. SW determined the primitive design of MIROC-ES2L and supervised the entire system. MK organized the project, supervised the entire system, and contributed to the background section.

## 7. Competing interests

The authors declare that they have no conflict of interest.

## 8. Code and data availability

The code of MIROC-ES2L are not publicly archived because of the copyright policy of MIROC community. Readers are
requested to contact the corresponding author if they wish to validate the model configurations of MIROC-ES2L and conduct replication experiments. The source codes, required input data, and simulation results will be provided by the modeling community to which the author belongs. The model output of the control, historical, and 1%CO2 increase simulations performed in this study will be distributed and made freely available through the Earth System Grid Federation (ESGF). Details on the ESGF can be found on the website of the CMIP Panel (https://www.wcrp-
climate.org/wgcm-cmip/wgcm-cmip6, last access: 28 August 2019).

## 9.Acknowledgments

This work was supported by TOUGOU/SOUSEI, the "Integrated Research Program for Advancing Climate Models"/"Program for Risk Information on Climate Change", by the Ministry of Education, Culture, Sports, Science, and Technology of Japan. This work was also partly supported by JSPS KAKENHI Grant Number 17K12820 and by scientific
collaboration in GCOM-C RA (JX-PSPC-500211). The Earth Simulator and JAMSTEC Super Computing System were used for the simulations, and the administration staff provided much supports. The authors are grateful for the programming support provided by Tsuyoshi Hasegawa and Shinichi Toshimitsu and for the engineering advice offered by Hiroaki Kanai. Osamu Arakawa provided powerful support and services on data archiving and server management. Kengo Sudo and Tomoko Nitta kindly provided the forcing data and the forcing preparation system, respectively. Kaoru Tachiiri
provided helpful and encouraging comments. This work was based on the long-term endeavor of members of the MIROC





community. We thank James Buxton MSc from Edanz Group (www.edanzediting.com./ac) for editing a draft of this manuscript.



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
