# Peer review of "Development of the MIROC-ES2L Earth system model and the evaluation of biogeochemical processes and feedbacks"

_Geoscientific Model Development, 2019_

## Referee Comment (RC1) · Jerry Tjiputra (Referee) · 26 Nov 2019

**General comments**

The manuscript by Hajima et al., describes the update in MIROC-ES2L (CMIP6) ESM relative to previous MIROC (CMIP5) ESMs, with emphasis on the land and ocean biogeochemistry components. In addition to the standard piControl and transient Historical simulations, the authors also performed additional sensitivity simulations with the updated model to analyze the impact of new parameterizations and new model improvements on the simulated biogeochemical metrics such as the land carbon content, ocean primary production, nutrient limitations, etc. They also use a set of C4MIP 1%CO2 simulations to quantify the simulated carbon fluxes sensitivity to climate change and atmospheric CO2 increase, and compare these quantities with that from previous MIROC ESM and other CMIP5 ESMs.

The paper is well structured and nicely written, providing a good overview of the new model with key information that would be of interest for GMD readers who need reference to the carbon cycle components in MIROC-ES2L model. Below I have listed several suggestions and comments that the author can consider to further improve the manuscript prior to publication in GMD. I also disclose that I am not an expert in land carbon cycle and therefore can only offer limited reviews on the land biogeochemical part of the paper. It would therefore be good to have additional referee(s) who can offer better perspectives on the land discussions of the paper.

**Specific comments**

The title gives away an impression that the paper fully describes the MIROC-ES2L model. While it does briefly, most of the content is focused on the land and carbon cycle description and assessments of the performed (biogeochemical-focused sensitivity) simulations. I therefore suggest modifying the title to something like "Description and sensitivity analysis of the carbon cycle components of MIROC-ES2L Earth system model."

The performed sensitivity experiments are very useful to characterize the model's response to (some are newly implemented) external forcing and future climate change. Many of the simulated changes in the key metrics, such as the ocean primary production and export production were quantified and discussed (e.g., Page 35). In addition to this useful discussion, I recommend the authors to expand (add additional columns) Tables 3 and 4 to include the same metrics but from the relevant sensitivity experiments

Some information on how stable (or any drift, if exist) the ocean biogeochemistry budgets in the preindustrial control run would be useful. Also, one of the uniqueness of the MIROC-ES2L among other CMIP6 models is the fact that it has a very long model spin up (i.e., >3500 years; albeit some are offline), which is thought to be crucial to reduce model bias (e.g., in the interior ocean biogeochemical tracers such as oxygen; Seferian et al., 2015; GMD). I think it would be of interest to many readers to know whether in fact interior biases is improved through this computationally costly process.

On Page 7, the authors wrote "a static biome distribution". Please briefly clarify what you meant by this? Do the prescribed PFTs change annually or or static in the sense that it is constant from preindustrial states?

The Figures' resolutions and qualities need some improvements. Namely, Fig. 2,3,4,10,12: x-labels and y-labels are difficult read in my printouts. Legends in Fig. 10 are also too small to read.

On Page 17, there is a statement that the model might overestimate net carbon uptake by land and/or ocean, or underestimate LUC emissions. This was implied through comparing the diagnosed

atmospheric CO2 concentration with the observed values. I found this strange. Since the simulation was performed with prescribed CO2 concentrations, the authors should instead compute the diagnosed anthropogenic emissions, and compared that with the observations (as in Jones et al. 2013), e.g., 403PgC. Based on this diagnosed emissions, the authors can then make a statement whether or not the land and/or ocean sinks are under/overestimated. Furthermore, if the authors have completed the prognostic CO2 simulations, it would be interesting to compare the diagnosed atmospheric CO2 concentrations and determine if the analysis on land/ocean carbon uptake strengths is consistent with the above approach.

Ocean carbon uptakes. While there's discussion on the cumulative carbon sinks over the historical period, there was no discussion on the simulated contemporary CO2 sinks, i.e., annual mean and spatial distributions (only stated in in Table 4). Given the importance of carbon sinks and its application in ESM simulations, I recommend adding some air-sea CO2 flux comparison between the model with e.g., spatial patterns from Landschutzer et al., 2014-https://doi.org/10.1002/2014GB004853; or Takahashi et al., 2009- https://doi.org/10.1016/j.dsr2.2008.12.009), and this discussion can be linked to the currently still limited surface DIC/alkalinity discussions, as shown in Fig. 11.

In addition to the contemporary spatial air-sea CO2 flux patterns, several studies have shown the importance of simulating proper regional seasonal cycle for constraining long-term spread in ESM projections (Kessler and Tjiputra, 2016-ESD; Goris et al., 2018-J. Climate). If the authors agree, it would be great to see how the regional CO2 fluxes in key world ocean regions (North Atlantic, Southern Ocean, etc.) are simulated in MIROC-ES2L and compare that with results from the previous model version.

Similarly for land, if seasonal cycle is an important criteria for constraining future projections (I am not fully aware of such studies on the terrestrial carbon side), then such presentation could be considered as well, e.g., similar to that shown in Tjiputra et al. (2013 - GMD) for GPP across different latitudinal bands.

In Section 3.1.4, MLD bias (too deep) is described as the partial reason for the SST cold bias. One could argue the other way around. On the other hand, in the Southern Ocean, there is a general warm bias, yet the MLD in the model also appears to be too deep. This should be elaborated. In this regard, i would also suggest adding Southern Hemisphere sea ice extent to Fig. 8.

Figure 15 shows that in many regions, there are two (instead of one) nutrients limit the primary production. It is not clear to me how the authors arrive with two limiting nutrients. Shouldn't it be only one, which is the minimum of the three? Some clarification on how it is derived would be useful.

Figure 17 and Table 5 essentially show the same information and appear redundant. I suggest removing Fig. 17 and keep the table.

**Technical corrections**
In addition to the above comments, please find below a list of minor comments that needs to be clarified as well as edits to improve the manuscript readability.

L25: Comparison … the model could reproduce well the transient global climate change and carbon cycle as well as reproduce the observed large-scale spatial patterns of land carbon cycle and upper ocean biogeochemistry.
L29: … revealed that the simulated ocean biogeochemistry could be altered regionally (and substantially) by …
L35: Suggest removing "model performance in"
L36-7: The MIROC-ES2L could further improve our understanding of climate …
L42: … on simulations using atmosphere- …
L43: had evolved
L48: future climate due to processes …
L52: semicolon between Watanabe and Collins references.
L55-6: As ESMs explicitly simulate …, they can simulate the temporal …
L60: Furthermore, their simulations can be …
L66: … on climate manifests through … OR … on climate is manifested through …
L69: … ocean, and eventually …
L77: transport instead of transportation
L100: you can also cite Kessler and Tjiputra (2016; https://doi.org/10.5194/esd-7-295-2016)
L111: remove 'nutrient'
L117: as well as the physical response
L162: with existing studies.
L170: used in the CMIP5
L182: … ocean physical model … land physical model …
L183: river routing (?)
L201: what is the ocean horizontal resolution?
L236: prescribed in the forcing data
L236-7: The fluxes of nitrogen out of the land ecosystem are simulated through N2 and N2O production during nitrification …
L277: is transported by rivers
L281: erosion and dissolution of organic carbon in the ocean, these processes are not activated to close the global mass conservation of carbon and nitrogen.
L283: replace 'simply diagnosed' with 'only for diagnostic purpose'
L290: … are simulated with 13 biogeochemical tracers.
L294: .. the Redfield ratio of C.N:P:O= …
L295: remove 'types of'
L298: The nitrogen cycle in OECO2 is similar ..
L301: denitrification is also …
L330: assumed to be
L334: remove 'oxygen' (it is not remineralized, it is consumed)
L347: this is only true when the model simulations is configured to be fully interactive, right? i.e., prognostic (not prescribed) atmospheric CO2.
L376: carbon cycle variation relative to the preindustrial control (CTL). …. climate-carbon feedback (Arora et al. 2013; Schwinger et al., 2014-https://doi.org/10.1175/JCLI-D-13-00452.1)
L386: these experiments
L387: I don't understand 'which would be noise in the analysis'. Maybe clarify or remove it.
Table 1 caption: Summary of experimental details.

Table 1, row NO-NRD: replace combination with combining; replace ' doesn't get impact from' with ' is not impacted by'

Table 1, row NO-FD: replace ' doesn't get impact from de' with ' is not impacted by Fe'

L414: replace status with states

L415: … is calculated independently of ocean biogeochemical states.

L427: The first term on the right

L432: replace 'which is because' with ' since'

L437: second term

L438: This sentence is a bit confusing. TCRE quantifies only the global temperature change in response to  emissions, and not the 'entire climate-carbon cycle response', right?

L441: … response to atmospheric CO2 increase …

Eqs (8) and (9) are wrong.

L456: … (TOA), and anomalies of new-surface …

L473-4: … internal climate variability (Kosaka et al., 2016).

L492: … positive), and anomalies of (b) …

L508: 166Pg, it looks closer to 200PgC in the figure.

L526: replace alleviated with weakened

L527: … within the independent estimates range of …

L528: … where the estimation of uncertainties take into account both …

L529: remove ' by Le Quere et al. (2018) are considered'

L531: … the model shows an increase in carbon …

L542: … conservation in the ocean biogeochemical component.

L548: … would not induce significant global-scale impact …

L570: remove global

L574: In the lower …

L575: replace 'against' with 'driven only by'

L597: .. reveals the annual inputs of …

L598: … increase to 46 …

L606: Shouldn't 4.5 be 4.2? (Table 3: 13.7-9.5)

L615: … shows a net …

L616: "… stimulates ecosystem nitrogen demand." Is this implied from the increasing atmospheric N2 fixation in the model?

L623: suggests that the historical …

Fig. 4 caption: Rate of change of global nitrogen budget in (a) land and in (b) ocean …

L660: … attributed to the model bias in simulating cloud cover ….

Fig 6: Why not show the difference between HIST and GPCP as Figs. 5 and 7?

L684-5: suggest revising it to something like:

When projecting future climate change, it is important for an ESM to reproduce the observed climatological patterns of key physical and biogeochemical tracers (Ohgaito and Abe-Ouchi, 2009).

L701: cover. Sea ice …

L701: … summertime minimum concentration is slightly …

L718: Briefly describe how the MLD is computed in the model.

L7738: replace higher with high

L740: is generally still underestimated

L749: Suggest replacing the reference Anav et al. with Jung et al., 2011.

L756: vegetation carbon content including …

L761: define NCSCDv2

L780: replace panels with rows

L784: add units ti the bottom row figures, presumably (g C m^-2)

L797: replace nitrate with nutrient
L799: consistent with the observed
L800: because of the implementation
L801: Ocean circulation also …
L803: … over estimation of nutrients entrainment to the surface and thus …
L804: The simulated global mean vertical profile of nitrate concentration …
L804-6: I am not convince this is the main or only reason of the well fitted model nitrate with observations. Model can simulate correct sources and sinks but still compare poorly with observations if the circulation fields is correct. There are several ways to get analyze whether the circulation fields is reasonable or not, e.g., using apparent oxygen utilization tracer. Nevertheless, you can get correct distribution for the wrong reason (Duteil et al., 2012 - Biogeosciences)
L825: … the equatorial Pacific Ocean, and the Southern Ocean …
L826: … the equatorial Pacific … than the observed …
L828: … much of the low-latitude surface ocean …
L846: … (Fig. 9), which also transports …
L848: replace parameters with tracers
L847-50: You mention alkalinity bias leads to DIC bias. Can you elaborate what cause the alkalinity bias in the first place?
L882: replace Northwest with 'path of north'
L950: remove 'one of'
L991: .. bias, which results in overestimation …
L995: remove 'produce a resultant'
L1002: .. the model is capable …
Fig. 16b shows that for the Neva, Yukon, and Churchill rivers, the DIN flux is larger in HIST-NOLUC than in HIST. This is in contrast to other rivers. Can the authors clarify the mechanism behind these patterns?
L1064: … is simulated to be 0.52 …
L1067: … is within this spread …
L1081: … is the main cause for the lower AF, making …
L1082: … feedbacks more positive and less negative, respectively, …
L1111-2: … plant growth and therefore the change in the land carbon fluxes.
L1115: … thus, the marine productivity is now also affected by the riverine nitrogen input.
L1118: … in response to changes in external iron inputs.
L1120: … We confirmed that …
L1122-3: … the model, the MIROC-ES2L's good performance in simulating … is inherited from its original …
L1114: … as found in some climate models.
L1126: Actually, there is no comparison of global biogeochemical (nitrate, phosphate, oxygen, etc.) budget trends, except for carbon. Suggest revising it to: … capture the observations-based estimates of contemporary air-sea and
air-land carbon fluxes.
L1127: … assessed through comparison …
L1228: the model produced reasonable …
L1133: …. compared with those of the CMIP5 ESMs.
L1138: … This is reduced from the value seen in the model …
L1141-4: Suggest rephrasing the sentence to: A multimodal comparison on feedback strengths using CMIP6 ESMs is necessary to potentially determine whether the climate and carbon cycle sensitivities in MIROC-ES2L are realistic, and furthermore to establish constraints on each feedback process based on observations (e.g., Wenzel et al., 2016; Goris et al., 2018 - J. Climate).

L1148: model alters the land carbon cycle. … carbon content during 1850-2014 is 44
L1154: nitrogen cycle alters the carbon cycle … did not quantify to what extent the soil nutrient ….
L1159: remove 'inputs'. 'external sources' already implies 'inputs'
L1164: replace GPP with primary production. You have used NPP in section 3.1.6 and 3.2.2.
Generally, in ESMs, we refer it to simply 'primary production'
L1173: … should also be …
L1177: here you said strongly impact, but on L1160, you states minor contributions. Please clarify.
L1179-80: … radiative balance in the atmosphere. Nitrous …. gases with a long ….
L1191: … a similar set of sensitivity simulations should be …
L1192: remove 'with which'
L1193: replace explorations with quantifications
L1196-7: … ESMs can reproduce some of the dominant long-term environmental changes on Earth …
L1198: ocean acidification …
L1201: replace evolve with improve
L1447-9: Shouldn't GCaCO3 be PrCaCO3 instead (based on Table B2)?
L1467: see eq. (14) of
L1498: (CE^D) should be (CE^P)
L1596: … Cycles, 26, GB2009, doi …
L2022-4: this citation needs to be updated.

Supp Fig 3: If I understood correctly, the 'N2 fixation' box should be labeled 'Diazotroph'.
There should also be arrows from PHY and ZOO to CaCO3?
Arrows from PHY to nutrients should only be labeled (Fast) remineralization, and the arrows from
PHY to "DNO3, DFe, (and DPO4)" should then be labelled 'mortality'
What is the purpose of the green label 'Definition of Alkalinity'? I suggest removing it.
Should the 'Nitrification' arrow be reversed?

Finally, if any of my comments are unclear, please feel free to contact me.

---

## Referee Comment (RC2) · Anonymous Referee #2 · 31 Dec 2019

This article presents the CMIP6 version of the MIROC Earth system model. I appreciate this work, because a detailed description and evaluation of the CMIP6 models helps to interpret their results and raises the scientific value of this major community effort. The article is written clearly and is quite extended (73 pages), but I guess that just a complete description of the complex model would require several hundred pages. Therefore, the authors concentrate on those aspects in the biogeochemical part of the model that are new in comparison to the CMIP5 version of the model. I completely agree with this approach, but it should be followed more consequently. I suggest two steps in this direction. First, the results of the physical model should be evaluated at one place (i.e. merge subsections 3.1.1 and 3.1.4). This can go along with a comment

that these results are presented first as a basis for the assessment of the biogeochemical results and the climate-carbon cycle feedbacks. Second, I see some potential to shorten the subsection 3.1.4, e.g. fig. 7 (SST) can be skipped as fig. 5 (2m temperature) is very similar. Also the title may be adapted towards the biogeochemical focus.

Scientific questions - section 3.1.1: For a more complete view on the simulated climate, please specify the strength of the AMOC and the amplitude of ENSO as these two features of the physical system also affect the simulated carbon cycle quite profoundly. Just mention the numbers. - line 527-530: Uncertainty in land carbon uptake (estimated from data) is smaller if calculated from the global carbon budgets (following $CL = CE - CA - CO$, s. line 435) than if it is calculated from the uncertainties in LUC emissions and the natural land sink. The uncertainty in CO is 20 PgC (s. line 550). The uncertainties in CE (derived directly from inventory data) and CA (derived from precise and representative measurements) are even smaller. Thus, the uncertainty in CL is much smaller than 90 PgC. - line 531-542: I'm not an expert in ocean biogeochemistry, but as far as I understand, a buildup of the ocean sediment reduces alkalinity in the ocean water, so that the ocean on the long term will outgas CO2 to the atmosphere and ocean water + ocean sediment looses carbon after the sedimentation process has been switched on (and the loss in alkalinity is not compensated by riverine input). By contrast, in the manuscript it is mentioned, that the ocean carbon uptake in the control run is partly explained by the sediment extracting carbon from the ocean bottom. - line 766-769: I don't understand, why the different treatment of the vertical SOC profile in the model and WISE30sec explains the large difference in the amount of SOC in the boreal range. I think, that has to do with permafrost. It should be mentioned here, if the model includes freezing in the soil and how this affects SOC. - section 3.2.2: The simulations NO-NR, NO-NRD, and NO-FD are only 100 years long. Do you've analyzed, if the signals that are based on these simulations and discussed in section 3.2.2 are already stable after 100 years? Or are they still very transient?

Minor corrections - line 18: "article describes" instead of "study developed" - line 176: Figure 1 is not helpful. Please specify for each model component whether it represents atmosphere, ocean, or land. You can also add for each model component the elements that are handled prognostically (C,N for VISIT-e, C,N,P,Fe,Ca,O for OECO-v2?, Fe,S for SPRINTARS?) and indicating by labeled arrows which elements are passed from each component to others (e.g. N,P from VISIT-e to OECO-v2). I think, this would result in a nice overview schematic, how the components are coupled concerning biogeochemistry. - line 201: Please indicate the horizontal resolution of the ocean model (e.g. average size of a grid box in km or the number of grid boxes of the global field). - line 210: What is a "snow-derived wetland"? - line 257: "of vegetation (each represented on a separate tile) in each land grid box" instead of "of tile in each land grid" - line 277: "transported by rivers" instead of "transported rivers" - line 305: The phosphorous cycle has also no analog to denitrification. - line 346: It would be nice to mention how the DMS affects the climate (I guess as sulfate aerosol in MATSIRO that affects radiation). - line 364, 372, 373: I would not use the word "detect" in that way. Please substitute it by e.g. "except that the prescribed $CO_2$ increase affects only the carbon cycle processes". - line 403: last 4 lines of table 1, NO-NRD Configurations "N depositions" instead of "Fe depositions", NO-FD Configurations "Fe depositions" instead of "de depositions". - line 438: "coupled" instead of "entire" - line 448,449: The denominator should be T. The common unit of Gamma is PgC/K (s. also table 6). - line 471: "deviations of the model results from HadCRUT4" instead of "discrepancies between the model result and HadCRUT4" - line 555: "in the HIST run" instead of "at the end of the HIST run" - line 606: "4.2 TgN yr-1" instead of "4.5 TgN yr-1" - compare with line 11 of table 3 - line 644: "decay of biomass in the LUC-product pools" instead of "decay of LUC-product pools" - line 701: "concentration minimum" instead of "concentration peak" - line 721: Please mention that the model (obviously) simulates no deep water formation in the Labrador Sea. - line 738: "high" instead of "higher" - line 755: "GPP in these regions is captured reasonably well by the model (Fig. 10a and 10b). Thus, the overestimation" instead of "Considering the GPP in these regions is

captured reasonably well by the model (Fig. 10a and 10b), the overestimation" - line 760: "products" instead of "product" - line 784: unit of SOC is missing - line 875: I see also some regions north of the equator, where GPP is reduced by climate change (e.g. South Asia). Please remove "of the Southern Hemisphere". - line 879-890: Please comment on the strong reduction of GPP by LUC in the tropics. - line 917: I think, it would be good to mention that the NPP increase in the open ocean by N input from rivers mainly occurs in the Atlantic. - line 973: "$CO_2$-induced ocean acidification and warming-induced deoxygenation" instead of "warming-induced ocean acidification and deoxygenation" - line 1002,1003: This sentence is better placed at the end of the paragraph. - line 1022,1023: This sentence is just repetition. You can remove it to shorten the article. - line 1026: "climate, carbon cycle, and coupled climate-carbon cycle system" instead of "climate, carbon cycle, and climate-carbon cycle system" - line 1130: "confirmed to be captured well" instead of "confirmed captured well"

---

## Author Comment (AC1) · 12 Mar 2020

**To Reviewer #1 [Dr. Jerry Tjiputra]**

**First, we greatly appreciate that you agreed to review our paper, and we thank you for your careful reading, positive comments, and many helpful suggestions. Following consideration of your insightful remarks, together with those of the other referee, we have revised our manuscript accordingly. Please find below (in bold) our detailed responses to your specific comments.**

General comments

The manuscript by Hajima et al., describes the update in MIROC-ES2L (CMIP6) ESM relative to previous MIROC (CMIP5) ESMs, with emphasis on the land and ocean biogeochemistry components. In addition to the standard piControl and transient Historical simulations, the authors also performed additional sensitivity simulations with the updated model to analyze the impact of new parameterizations and new model improvements on the simulated biogeochemical metrics such as the land carbon content, ocean primary production, nutrient limitations, etc. They also use a set of C4MIP 1%CO2 simulations to quantify the simulated carbon fluxes sensitivity to climate change and atmospheric CO2 increase, and compare these quantities with that from previous MIROC ESM and other CMIP5 ESMs.

The paper is well structured and nicely written, providing a good overview of the new model with key information that would be of interest for GMD readers who need reference to the carbon cycle components in MIROC-ES2L model. Below I have listed several suggestions and comments that the author can consider to further improve the manuscript prior to publication in GMD. I also disclose that I am not an expert in land carbon cycle and therefore can only offer limited reviews on the land biogeochemical part of the paper. It would therefore be good to have additional referee(s) who can offer better perspectives on the land discussions of the paper.

**Thank you for your positive comments regarding this research. The other reviewer of our paper is expert in the field of land biogeochemistry, and thus we believe the manuscript has been reviewed by peers with expertise in both the land and the ocean elements of the subject area.**

**The structure of the revised manuscript has been changed slightly from that of the first draft in accordance with the comments from Reviewer #2. In the first draft, the results on the physical climate were split into two subsections (i.e., "3.1.1. Global climate: net radiation balance and global temperature" and "3.1.4. Climate: atmosphere and ocean physical fields"), whereas these two subsections have been merged into a single unit ("3.1.1. Global climate: atmosphere and ocean physical fields") in the revised version.**

Specific comments

The title gives away an impression that the paper fully describes the MIROC-ES2L model. While it does briefly, most of the content is focused on the land and carbon cycle description and assessments of the performed (biogeochemical-focused sensitivity) simulations. I therefore suggest modifying the title to something like "Description and sensitivity analysis of the carbon cycle components of MIROC- ES2L Earth system model."

**Thank you for your suggestion regarding the article title. Yes, we agree that the sensitivity analysis and model evaluation performed in this study focused mainly on biogeochemical processes (and the feedbacks on climate via $CO_2$ exchange). To reflect this fully, we have therefore changed the title to** *"Development of the MIROC-ES2L Earth system model and the evaluation of biogeochemical processes and feedbacks"***.**

The performed sensitivity experiments are very useful to characterize the model's response to (some are newly implemented) external forcing and future climate change. Many of the simulated changes in the key metrics, such as the ocean primary production and export production were quantified and discussed (e.g., Page 35). In addition to this useful discussion, I recommend the authors to expand (add additional columns) Tables 3 and 4 to include the same metrics but from the relevant sensitivity experiments

**We thank you for this good idea to expand the tables to include other simulation results and we agree it would be informative for the readers. To keep the manuscript as concise as possible, we have included the results of the other nine experiments in Supplementary Tables 1 and 2.**

**Additionally, we found an inconsistency between land (Table 3) and ocean (Table 4) with regard to the averaging period for the CTL experiment. In the revised manuscript, this inconsistency has been resolved and thus the numbers for land in Table 3 have been updated. These minor modifications have not changed the fundamental conclusions of this study.**

Some information on how stable (or any drift, if exist) the ocean biogeochemistry budgets in the preindustrial control run would be useful. Also, one of the uniqueness of the MIROC-ES2L among other CMIP6 models is the fact that it has a very long model spin up (i.e., >3500 years; albeit some are offline), which is thought to be crucial to reduce model bias (e.g., in the interior ocean biogeochemical tracers such as oxygen; Seferian et al., 2015; GMD). I think it would be of interest

to many readers to know whether in fact interior biases is improved through this computationally costly process.

**We agree that information on bias would be useful to the readers. Following your suggestion, we have mentioned model bias in the main text (as below) and added a table in the Supplementary Material.**

*"Owing to the long spin-up, the drift in global averaged concentrations of biogeochemical tracers becomes close to zero. The linear drift of dissolved oxygen, NO3, and Alk-DIC over the final 250 years of the spin-up is less than 3% $kyr^{-1}$ (Supplementary Table 3). This small bias is significant in providing results on ocean biogeochemistry and carbon cycle feedbacks that are quantitatively more correct (Séférian et al., 2016)."*

On Page 7, the authors wrote "a static biome distribution". Please briefly clarify what you meant by this? Do the prescribed PFTs change annually or or static in the sense that it is constant from preindustrial states?

**We meant the distribution of PFTs is fixed (constant) throughout the simulations because the terrestrial model is not a dynamic vegetation model. We have reworded this passage in the main text as follows:**

*"…the biogeochemical processes are simulated under a fixed biome distribution (Supplementary Fig. 2)…"*

The Figures' resolutions and qualities need some improvements. Namely, Fig. 2,3,4,10,12: x-labels and y-labels are difficult read in my printouts. Legends in Fig. 10 are also too small to read.

**We apologize for this inconvenience. In the revised manuscript, we have increased the font size used in the figures mentioned.**

On Page 17, there is a statement that the model might overestimate net carbon uptake by land and/or ocean, or underestimate LUC emissions. This was implied through comparing the diagnosed atmospheric CO2 concentration with the observed values. I found this strange. Since the simulation was performed with prescribed CO2 concentrations, the authors should instead compute the diagnosed anthropogenic emissions, and compared that with the observations (as in Jones et al. 2013), e.g., 403PgC. Based on this diagnosed emissions, the authors can then make a statement whether or not the land and/or ocean sinks are under/overestimated. Furthermore, if the authors have completed the prognostic CO2 simulations, it would be interesting to compare the diagnosed atmospheric CO2 concentrations and determine if the analysis on land/ocean carbon uptake strengths is consistent with the above approach.

**Yes, we agree the comparison of anthropogenic (fossil fuel) emission, as in Jones et al. (2012) would be straightforward and informative for the readers. Following your suggestion, we have also compared the diagnosed fossil fuel emission with the GCP estimation in the revised manuscript.**

**We also agree it would be a good idea to mention the prognostic $CO_2$ concentration in the emission-driven historical run. As we had finished the emission-driven runs (esm-historical) during the reviewing process, we have mentioned the prognostic $CO_2$ concentration in the revised manuscript as follows:**

*"However, the model might overestimate net carbon uptake by the land and/or ocean or underestimate LUC emissions. This is because the cumulative fossil fuel emission, diagnosed from the simulated atmosphere–land/ocean $CO_2$ fluxes and prescribed $CO_2$ concentration change (FF = CA + CL + CO; Appendix D), was 447 PgC, i.e., larger than the estimate of 400 ± 20 PgC of Le Quéré et al. (2018). Additionally, this speculation is also supported by the diagnosed $CO_2$ concentration at the end of the HIST run (Appendix D); the diagnosed concentration is 376 ppmv, which is lower (by 22 ppmv) than that actually monitored. We note, however, the likely biases in land/ocean carbon uptake, suggested by the larger diagnosed emission/lower diagnosed $CO_2$ concentration, could be alleviated partially if the model were driven by anthropogenic $CO_2$ emissions. This is because in emission-driven mode, the relatively stronger land/ocean carbon uptake leads to lower atmospheric $CO_2$ concentration, which could weaken the land and ocean sink through negative $CO_2$–carbon feedback. Indeed, in emission-driven mode, the atmospheric $CO_2$ concentration in the historical run ("esm-historical", Jones et al., 2016) is simulated to be 384 ppmv in 2014 (as an average of three ensemble experiments; data not shown but available via the Earth System Grid Federation servers), which is closer to the actual level monitored (but still lower by 14 ppmv)."*

Ocean carbon uptakes. While there's discussion on the cumulative carbon sinks over the historical period, there was no discussion on the simulated contemporary CO2 sinks, i.e., annual mean and spatial distributions (only stated in in Table 4). Given the importance of carbon sinks and its application in ESM simulations, I recommend adding some air-sea CO2 flux comparison between the model with e.g., spatial patterns from Landschutzer et al., 2014-https://doi.org/10.1002/2014GB004853; or Takahashi et al., 2009-https://doi.org/10.1016/j.dsr2.2008.12.009), and this discussion can be linked to the currently still limited surface DIC/alkalinity discussions, as shown in Fig. 11.

In addition to the contemporary spatial air-sea CO2 flux patterns, several studies have shown the importance of simulating proper regional seasonal cycle for constraining long-term spread in ESM projections (Kessler and Tjiputra, 2016-ESD; Goris et al., 2018-J. Climate). If the authors agree, it would be great to see how the regional CO2 fluxes in key world ocean regions (North Atlantic, Southern Ocean, etc.) are simulated in MIROC-ES2L and compare that with results from the previous model version.

**We thank you for your useful suggestion. Accordingly, we have added a comparison between the spatial distribution of observed air–sea $CO_2$ flux and that simulated by MIROC-ES2L to the revised manuscript (Fig. 12a). We have also included comparison of the regional seasonal cycle of $CO_2$ fluxes between the model and observations (Fig. 12b and 12c).**

Similarly for land, if seasonal cycle is an important criteria for constraining future projections (I am not fully aware of such studies on the terrestrial carbon side), then such presentation could be considered as well, e.g., similar to that shown in Tjiputra et al. (2013 - GMD) for GPP across different latitudinal bands.

**This is an important suggestion and just such a multimodel comparison study on contemporary land GPP has been reported by Anav et al. (2015), although the relationship between the seasonal cycle and the long-term trend of global GPP remains unclear. Thus, following your suggestion, we have analyzed GPP seasonality and a relevant figure is included in the Supplementary Material. We have mentioned it in the revised manuscript as follows:**
***"The simulated GPP seasonality is also compared with that of Jung et al. (2011) (Supplementary Fig. 9). It reveals a reasonable summertime peak and the seasonality of GPP in the extratropical Northern/Southern Hemisphere, where vegetation phenology is controlled primarily by air temperature. However, the region around 40°N displays a longer growing season than that of Jung et al. (2011), and the tropics (20°S–20°N) show less seasonality, suggesting room for improvement of the phenology-related processes and surface climate fields in the corresponding region/biome types."***

In Section 3.1.4, MLD bias (too deep) is described as the partial reason for the SST cold bias. One could argue the other way around. On the other hand, in the Southern Ocean, there is a general warm bias, yet the MLD in the model also appears to be too deep. This should be elaborated. In this regard, i would also suggest adding Southern Hemisphere sea ice extent to Fig. 8.

**Thank you for this important suggestion. We agree that we cannot conclude that only deep MLD bias causes the SST cold bias, and careful discussion should address the SST bias. In particular, we think the mechanisms responsible for producing the SST biases are potentially different between the Southern Ocean and western North Pacific Ocean. To clarify the possible mechanisms, we have revised the manuscript (Section 3.1.4) as follows:**

**(1) We have added a figure showing the sea ice extent in the Southern Hemisphere (Fig. 6), in accordance with your suggestion.**

**(2) The mechanism that possibly produces the warm (T2) bias in the Southern Ocean is discussed more carefully as follows:**

***"The warm bias in the Southern Ocean can be attributed mainly to poor representation of cloud radiative processes (Bodas-Salcedo et al., 2012; Williams et al., 2013; Hyder et al., 2018), but also to poor representations of the mixed-layer depth and deep convection in the open ocean attributable to the lack of modeled mesoscale processes in the Antarctic Circumpolar Current (Tatebe et al., 2019). A related warm bias in SST over the Southern Ocean is also confirmed, which is discussed later."***

**(3) The mechanism that possibly produces the cold SST bias in the western North Pacific Ocean is also mentioned in the revised manuscript as follows:**

***"A cold bias is also evident over the western North Pacific Ocean, which is attributable to the lack of narrow and swift western boundary currents owing to the coarse horizontal resolution in the ocean parts of the present ESM."***

Figure 15 shows that in many regions, there are two (instead of one) nutrients limit the primary production. It is not clear to me how the authors arrive with two limiting nutrients. Shouldn't it be only one, which is the minimum of the three? Some clarification on how it is derived would be useful.

**As you correctly commented, the estimation of the limiting (two) nutrients was not clear in the original manuscript. We have clarified it in the main part of the revised text as follows:**

***"Here, NO3, Fe, and PO4 limitation is diagnosed using the equations NO3/(kN + NO3), Fe/(kFe + Fe), and PO4/(kP + PO4), respectively, as simulated in MIROC-ES2L (see Equation (B17)); Fig.16 presents the strength of each limitation visualized by the intensity of each of the three primary colors (red, blue, and green)."***

Figure 17 and Table 5 essentially show the same information and appear redundant. I suggest removing Fig. 17 and keep the table.

**Following your suggestion, Fig. 17 (the figure for TCR, AF, and TCRE) has been removed from the revised manuscript and Table 5 has been retained.**

Technical corrections

In addition to the above comments, please find below a list of minor comments that needs to be clarified as well as edits to improve the manuscript readability.

**We appreciate your careful reading of our manuscript and the listing of identified technical corrections.**

L25: Comparison ... the model could reproduce well the transient global climate change and carbon cycle as well as reproduce the observed large-scale spatial patterns of land carbon cycle and upper ocean biogeochemistry.

**We have changed the manuscript accordingly.**

L29: ... revealed that the simulated ocean biogeochemistry could be altered regionally (and substantially) by ...

**We have changed the manuscript accordingly.**

L35: Suggest removing "model performance in"

**We have removed the corresponding words.**

L36-7: The MIROC-ES2L could further improve our understanding of climate ...

**We have changed the manuscript following your suggestion.**

L42: ... on simulations using atmosphere- …

**Corrected.**

L43: had evolved

**Corrected.**

L48: future climate due to processes …

**Corrected.**

L52: semicolon between Watanabe and Collins references.

**Corrected.**

L55-6: As ESMs explicitly simulate ..., they can simulate the temporal ...

**Corrected.**

L60: Furthermore, their simulations can be …

**Corrected.**

L66: ... on climate manifests through ... OR ... on climate is manifested through ...

**Corrected to "…climate is manifested through..."**

L69: ... ocean, and eventually …

**Reworded as follows:**

*"…and the absorbed $CO_2$ is transported into the deeper ocean…"*

L77: transport instead of transportation

**Corrected.**

L100: you can also cite Kessler and Tjiputra (2016; https://doi.org/10.5194/esd-7-295-2016)

**Thank you for suggesting we include this useful reference; the work has been cited in the revised manuscript.**

L111: remove 'nutrient'

**Corrected.**

L117: as well as the physical response

**Corrected.**

L162: with existing studies.

**Corrected.**

L170: used in the CMIP5

**Corrected.**

L182: ... ocean physical model ... land physical model …

**Corrected.**

L183: river routing (?)

**Replaced with "river submodel".**

L201: what is the ocean horizontal resolution?
**In the revised manuscript, we have specified the ocean horizontal resolution as follows:**
**"…that is divided horizontally into 360 × 256 grids. (To the south of 63°N, the longitudinal grid spacing is 1° and the meridional spacing becomes fine near the Equator. In the central Arctic Ocean, the grid spacing is finer than 1° because of the tripolar system.)"**

L236: prescribed in the forcing data
**Corrected.**

L236-7: The fluxes of nitrogen out of the land ecosystem are simulated through N2 and N2O production during nitrification ...
**Corrected.**

L277: is transported by rivers
**Corrected.**

L281: erosion and dissolution of organic carbon in the ocean, these processes are not activated to close the global mass conservation of carbon and nitrogen.
**Corrected.**

L283: replace 'simply diagnosed' with 'only for diagnostic purpose'
**Replaced as suggested, albeit as "…only for diagnostic purposes…"**

L290: ... are simulated with 13 biogeochemical tracers.
**Replaced by the suggested words.**

L294: .. the Redfield ratio of C.N:P:O= ...
**Corrected.**

L295: remove 'types of'
**Corrected.**

L298: The nitrogen cycle in OECO2 is similar ..

**Corrected.**

L301: denitrification is also ...

**Corrected.**

L330: assumed to be

**Corrected.**

L334: remove 'oxygen' (it is not remineralized, it is consumed)

**Removed.**

L347: this is only true when the model simulations is configured to be fully interactive, right? i.e., prognostic (not prescribed) atmospheric CO2.

**Yes, you are correct. The sentence has been modified in the revised manuscript:**

***"…this is the only pathway via which ocean biogeochemistry affects climate if the model is driven by prescribed $CO_2$ concentration."***

L376: carbon cycle variation relative to the preindustrial control (CTL). .... climate-carbon feedback (Arora et al. 2013; Schwinger et al., 2014-https://doi.org/10.1175/JCLI-D-13-00452.1)

**The two references are now cited in the corresponding section of text.**

L386: these experiments

**Corrected.**

L387: I don't understand 'which would be noise in the analysis'. Maybe clarify or remove it.

**To avoid confusion, this text has been removed.**

Table 1 caption: Summary of experimental details.

**Corrected.**

Table 1, row NO-NRD: replace combination with combining; replace ' doesn't get impact from' with ' is not impacted by'

**Corrected.**

Table 1, row NO-FD: replace ' doesn't get impact from de' with ' is not impacted by Fe'

**Corrected.**

L414: replace status with states
**Corrected.**

L415: ... is calculated independently of ocean biogeochemical states.
**Corrected.**

L427: The first term on the right
**Corrected.**

L432: replace 'which is because' with ' since'
**Corrected.**

L437: second term
**Corrected.**

L438: This sentence is a bit confusing. TCRE quantifies only the global temperature change in response to emissions, and not the 'entire climate-carbon cycle response', right?
**Yes, your understanding is correct. For clarity, we have reworded the sentence as follows:**
**"*As shown in Matthews et al. (2009), AF summarizes the carbon cycle response to anthropogenic CE; the second term in Eq. 5 (TCR/CA) captures the global temperature response to CO$_2$ increase in the models; and TCRE thus summarizes the two, i.e., the global temperature response to anthropogenic CO$_2$ emission in the model.*"**

L441: ... response to atmospheric CO2 increase ...
**Corrected.**

Eqs (8) and (9) are wrong.
**Thank you for drawing our attention to this error; the equations have been corrected by replacing CA with T.**

L456: ... (TOA), and anomalies of new-surface …
**Corrected.**

L473-4: ... internal climate variability (Kosaka et al., 2016).

**Corrected.**

L492: ... positive), and anomalies of (b) …

**Corrected as follows:**

*"…anomalies of (a) net radiation balance at the top of the atmosphere (TOA; upward positive), (b) global mean surface air temperature, (c) global mean sea surface temperature, and (d) global mean ocean temperature at 0–700 m depth…"*

L508: 166Pg, it looks closer to 200PgC in the figure.

**Thank you for highlighting this typo. The erroneous number has been replaced with the correct one (200 PgC) in the revised manuscript.**

L526: replace alleviated with weakened

**Corrected.**

L527: ... within the independent estimates range of …

**Corrected.**

L528: ... where the estimation of uncertainties take into account both ...

**Corrected.**

L529: remove ' by Le Quere et al. (2018) are considered'

**Corrected.**

L531: ... the model shows an increase in carbon ...

**Corrected.**

L542: ... conservation in the ocean biogeochemical component.

**Corrected.**

L548: ... would not induce significant global-scale impact ...

**Corrected.**

L570: remove global

**Corrected.**

L574: In the lower ...
**Corrected.**

L575: replace 'against' with 'driven only by'
**Corrected.**

L597: .. reveals the annual inputs of …
**Corrected.**

L598: ... increase to 46 ...
**Corrected.**

L606: Shouldn't 4.5 be 4.2? (Table 3: 13.7-9.5)
**The incorrect number has been replaced with 4.3 TgN yr$^{-1}$.**

L615: ... shows a net ...
**Because of the rewording applied in response to the following point, these words have been removed.**

L616: "... stimulates ecosystem nitrogen demand." Is this implied from the increasing atmospheric N2 fixation in the model?
**This is implied from the net increase of land N uptake in 1PPY-BGC.**
**In 1PPY-BGC, N inputs (deposition, fertilizer, and BNF) are almost unchanged from the CTL; thus, atmospheric $CO_2$ increase is the main driver of change in land N cycles. However, the 1PPY-BGC run indicated land N uptake, suggesting that atmospheric $CO_2$ increase alters the C:N ratio in land organic materials and that a higher C:N ratio in the organic materials promotes N uptake. We have clarified this in the revised manuscript as follows:**
***"In addition to the increasing N input, the net positive N uptake by land is likely accelerated by the increased nitrogen demand by plants and soils that have higher C:N ratios under elevated $CO_2$ concentrations. This is because the net increase of land N uptake is also found in 1PPY-BGC (Supplementary Table 1), even though the N inputs such as BNF, fertilizer, deposition, and climate condition in the 1PPY-BGC simulation are almost unchanged from the CTL run. It suggests atmospheric $CO_2$***

*increase in HIST has changed the C:N ratios in plants and soil and hence stimulated ecosystem nitrogen demand."*

L623: suggests that the historical ...
**Corrected.**

Fig. 4 caption: Rate of change of global nitrogen budget in (a) land and in (b) ocean ...
**Corrected.**

L660: ... attributed to the model bias in simulating cloud cover ....
**The corresponding sentence has been changed as follows to clarify the mechanism that produces the warm bias:**
*"The warm bias in the Southern Ocean can be attributed mainly to poor representation of cloud radiative processes (Bodas-Salcedo et al., 2012; Williams et al., 2013; Hyder et al., 2018)..."*

Fig 6: Why not show the difference between HIST and GPCP as Figs. 5 and 7?
**Following your suggestion, the map of GPCP precipitation has been replaced with a map of the bias (HIST − GPCP). The map of raw GPCP precipitation has been moved to the Supplementary Material because it remains useful regarding the discussion of the problems in land biogeochemistry.**

L684-5: suggest revising it to something like:
When projecting future climate change, it is important for an ESM to reproduce the observed climatological patterns of key physical and biogeochemical tracers (Ohgaito
**Following your suggestion, we have included this text with some modifications:**
*"When projecting future climate change, it is important for a model to reproduce the observed climatological patterns of key physical variables, as suggested in Ohgaito and Abe-Ouchi 2009. The biogeochemical tracers are also affected by the representation of the physical fields."*

L701: cover. Sea ice ...
**Corrected.**

L701: ... summertime minimum concentration is slightly ...
**Corrected.**

L718: Briefly describe how the MLD is computed in the model.

**We have clarified this in Section 3.1.1 as follows:**

*"The mixed-layer depth is defined as the depth where the potential density becomes larger than that of the sea surface by 0.125 kg m$^{-3}$."*

L738: replace higher with high

**Corrected.**

L740: is generally still underestimated

**Corrected.**

L749: Suggest replacing the reference Anav et al. with Jung et al., 2011.

**Replaced as suggested.**

L756: vegetation carbon content including ...

**Corrected (we supposed you intended L758).**

L761: define NCSCDv2

**The abbreviation has now been expanded.**

L780: replace panels with rows

**Corrected.**

L784: add units ti the bottom row figures, presumably (g C m^-2)

**The unit for SOC has now been specified in the caption.**

L797: replace nitrate with nutrient

**Corrected.**

L799: consistent with the observed

**Corrected.**

L800: because of the implementation

**The relevant sentence was changed as follows:**

**"*This increase of macronutrients in HNLC regions is reasonable because the*"**

*implementation of the iron cycle and the iron limitation on phytoplankton growth can reduce macronutrient utilization in these regions.*"

L801: Ocean circulation also …
**Corrected.**

L803: ... over estimation of nutrients entrainment to the surface and thus ...
**Corrected.**

L804: The simulated global mean vertical profile of nitrate concentration ...
**Corrected.**

L804-6: I am not convince this is the main or only reason of the well fitted model nitrate with observations. Model can simulate correct sources and sinks but still compare poorly with observations if the circulation fields is correct. There are several ways to get analyze whether the circulation fields is reasonable or not, e.g., using apparent oxygen utilization tracer. Nevertheless, you can get correct distribution for the wrong reason (Duteil et al., 2012 - Biogeosciences)

**We agree with your suggestion. We have mentioned the significance of the ocean circulation with regard to the simulated tracer distributions in the revised manuscript (as below) and we have added a model–data comparison of the AOU in Supplementary Fig. 10.**
*"To check the influence of ocean circulation on the tracer distributions, we compared the AOU between the model and observations (Supplementary Fig. 10). Although the model captures the observed AOU distributions, the strong and deep AMOC causes underestimation of AOU values in the Atlantic Ocean deep water. The largest bias is underestimation in the North Pacific Ocean, which is caused by the strong Pacific Ocean deep circulation. It should be noted that the difficulty of simulating the Pacific Ocean deep circulation appears to be a general problem in present coarse-resolution models (Hasumi et al., 2010)."*

L825: ... the equatorial Pacific Ocean, and the Southern Ocean ... L826: ... the equatorial Pacific ... than the observed …
**Corrected.**

L828: ... much of the low-latitude surface ocean ...

**Corrected.**

L846: ... (Fig. 9), which also transports ...
**Corrected.**

L848: replace parameters with tracers
**Replaced as suggested.**

L847-50: You mention alkalinity bias leads to DIC bias. Can you elaborate what cause the alkalinity bias in the first place?
**We have added the reason for the alkalinity bias to the revised manuscript in accordance with your comment:**
***"Salinity bias as well as parameterization of calcium carbonate production in the model can contribute to the alkalinity bias."***

L882: replace Northwest with 'path of north'
**Corrected.**

L950: remove 'one of'
**Removed.**

L991: .. bias, which results in overestimation ...
**Corrected.**

L995: remove 'produce a resultant'
**Corrected.**

L1002: .. the model is capable ...
**Corrected.**

Fig. 16b shows that for the Neva, Yukon, and Churchill rivers, the DIN flux is larger in HIST-NOLUC than in HIST. This is in contrast to other rivers. Can the authors clarify the mechanism behind these patterns?
**One of the possible reasons for the larger DIN in HIST-NOLUC than in HIST is the difference in leaf area index (LAI). In HIST-NOLUC, land use maintains the preindustrial condition, while in the HIST simulation LAI is affected by forcing**

associate with land use change (LUC). Consequently, LAI in the HIST-NOLUC experiment should be different from that of HIST (the left-hand panel in the figure below), producing a slightly different spatial pattern of temperature (the right-hand panel). In particular, we can confirm the temperature anomaly in each of the three river basins is positive, which suggests the decomposition rate of soil organic matter is likely accelerated in HIST-NOLUC compared with HIST. This could lead to more inorganic soil N and allow the rivers to transport more DIN to the coastal region. Given that the three river basins are affected little by agriculture (e.g., see Fig. 13), the difference in DIN between HIST-NOLUC and HIST is likely attributable to the indirect LUC impact via temperature change. In addition, the averaging period for the analysis might affect the result, particularly for a region with little LUC.

Although this is an interesting point to analyze in detail, we believe this issue is beyond the scope of this paper. Thus, rather than specifically explaining the possible mechanism, we briefly acknowledge it in the revised manuscript:

*"We note that the DIN discharge in each river is not always smaller in HIST-NOLUC than in HIST. This is because LAI in HIST-NOLUC is different to that in HIST, which sometimes is accompanied by slight change in the surface climate via biophysical feedback. If soil temperature is slightly warmer in HIST-NOLUC than in HIST, the soil mineralization rate in HIST-NOLUC should be accelerated and thus the DIN loadings of rivers could be increased."*

[Figure]

**Figure for Reviewer: the difference in leaf area index (left) and air temperature (right) between HIST-NOLUC and HIST (HIST-NOLUC − HIST). Averages over 30 years (1985–2014) in each experiment were used for the calculation.**

L1064: ... is simulated to be 0.52 ...

**Corrected.**

L1067: ... is within this spread ...
**Corrected.**

L1081: ... is the main cause for the lower AF, making ...
**Corrected.**

L1082: ... feedbacks more positive and less negative, respectively, ...
**Corrected.**

L1111-2: ... plant growth and therefore the change in the land carbon fluxes**.**
**Corrected.**

L1115: ... thus, the marine productivity is now also affected by the riverine nitrogen input.
**Corrected.**

L1118: ... in response to changes in external iron inputs.
**Corrected.**

L1120: ... We confirmed that ...
**Corrected.**

L1122-3: ... the model, the MIROC-ES2L's good performance in simulating ... is inherited from its original ...
**Corrected.**

L1114: ... as found in some climate models.
**Corrected (at L1124 in the first draft).**

L1126: Actually, there is no comparison of global biogeochemical (nitrate, phosphate, oxygen, etc.) budget trends, except for carbon. Suggest revising it to: ... capture the observations-based estimates of contemporary air-sea and air-land carbon fluxes.
**Thank you for your suggestion. We have changed the sentence as you recommended. In addition, as we also discuss the global nitrogen budget (section 3.1.3), we have also mentioned this in the revised manuscript:**

*"It was confirmed that the model could successfully capture the observation-based estimates of contemporary air–sea and air–land carbon fluxes in terms of cumulative values. We also confirmed that the component fluxes of global nitrogen between land, atmosphere, and ocean are reproduced reasonably by the model."*

L1127: ... assessed through comparison ...
**Corrected.**

L1228: the model produced reasonable ...
**Corrected. (at L1128 in the first draft)**

L1133: .... compared with those of the CMIP5 ESMs.
**Corrected.**

L1138: ... This is reduced from the value seen in the model ...
**Corrected.**

L1141-4: Suggest rephrasing the sentence to: A multimodal comparison on feedback strengths using CMIP6 ESMs is necessary to potentially determine whether the climate and carbon cycle sensitivities in MIROC-ES2L are realistic, and furthermore to establish constraints on each feedback process based on observations (e.g., Wenzel et al., 2016; Goris et al., 2018 - J. Climate).
**Thank you for this useful suggestion. We have rephrased the sentence and added the reference to the work of Goris et al. (2018) in the revised version.**

L1148: model alters the land carbon cycle. ... carbon content during 1850-2014 is 44
**Corrected, as you suggested.**

L1154: nitrogen cycle alters the carbon cycle ... did not quantify to what extent the soil nutrient ....
**Corrected.**

L1159: remove 'inputs'. 'external sources' already implies 'inputs'
**Corrected.**

L1164: replace GPP with primary production. You have used NPP in section 3.1.6 and 3.2.2. Generally, in ESMs, we refer it to simply 'primary production'
**This correction has been made, i.e., it has been replaced with "primary production."**

L1173: ... should also be ...
**Corrected.**

L1177: here you said strongly impact, but on L1160, you states minor contributions. Please clarify.
**We meant to emphasize that river input is minor on the global scale (L1160 in the previous manuscript), but significant on the regional scale (L1177 in the previous manuscript). In the revised manuscript, this has been clarified as follows:**
**"*Our sensitivity analyses under the preindustrial condition suggested minor contributions of these two external sources to primary productivity on the global scale*" and**
**"*This conclusion is supported by the sensitivity analysis that showed relatively strong regional-scale impact of riverine nitrogen on marine primary productivity, although the global-scale impact was demonstrated to be minor.*"**

L1179-80: ... radiative balance in the atmosphere. Nitrous .... gases with a long ....
**Corrected.**

L1191: ... a similar set of sensitivity simulations should be ...
**Corrected.**

L1192: remove 'with which'
**Corrected.**

L1193: replace explorations with quantifications
**Corrected.**

L1196-7: ... ESMs can reproduce some of the dominant long-term environmental changes on Earth ...
**Corrected.**

L1198: ocean acidification ...
**Corrected.**

L1201: replace evolve with improve
**Corrected.**

L1447-9: Shouldn't GCaCO3 be PrCaCO3 instead (based on Table B2)?

**The S terms in Eqs. (B12)–(B14) represent the "source minus sink" and thus the Eqs. (B12)–(B14) are correct as presented.**

L1467: see eq. (14) of

**Corrected.**

L1498: (CE^D) should be (CE^P)

**Corrected.**

L1596: ... Cycles, 26, GB2009, doi ...

**Corrected.**

L2022-4: this citation needs to be updated.

**This reference has been updated in the revised text.**

Supp Fig 3: If I understood correctly, the 'N2 fixation' box should be labeled 'Diazotroph'.

There should also be arrows from PHY and ZOO to CaCO3?

Arrows from PHY to nutrients should only be labeled (Fast) remineralization, and the arrows from PHY to "DNO3, DFe, (and DPO4)" should then be labelled 'mortality'

What is the purpose of the green label 'Definition of Alkalinity'? I suggest removing it. Should the 'Nitrification' arrow be reversed?

**Your suggested points are correct. In the new figure:**

- **"N2 fixation" has been replaced by "Diazotroph"**
- **Arrows from PHY/ZOO to CaCO3 have been added**
- **The label from PHY to detritus has been changed to "mortality" only**
- **The label "Definition of Alkalinity" has been removed**
- **The "Nitrification" arrow has been removed**

Finally, if any of my comments are unclear, please feel free to contact me.

**Thank you very much again for your careful reading and many suggestions. All your suggestions and comments were very helpful to us in improving our manuscript. We believe we have fully reflected all of your comments properly in the revised manuscript.**

---

## Author Comment (AC2) · 12 Mar 2020

**To Reviewer #2**

This article presents the CMIP6 version of the MIROC Earth system model. I appreciate this work, because a detailed description and evaluation of the CMIP6 models helps to interpret their results and raises the scientific value of this major community effort. The article is written clearly and is quite extended (73 pages), but I guess that just a complete description of the complex model would require several hundred pages. Therefore, the authors concentrate on those aspects in the biogeochemical part of the model that are new in comparison to the CMIP5 version of the model. I completely agree with this approach

**First, we greatly appreciate that you agreed to review our paper and we thank you for your positive comments on this work. Following consideration of your comments, together with those of the other referee, which we found very helpful, we have revised the manuscript accordingly. Please find below (in bold) our detailed responses to your specific comments.**

it should be followed more consequently. I suggest two steps in this direction. First, the results of the physical model should be evaluated at one place (i.e. merge subsections 3.1.1 and 3.1.4). This can go along with a comment that these results are presented first as a basis for the assessment of the biogeochem-ical results and the climate-carbon cycle feedbacks. Second, I see some potential to shorten the subsection 3.1.4, e.g. fig. 7 (SST) can be skipped as fig. 5 (2m tem-perature) is very similar. Also the title may be adapted towards the biogeochemical focus.

**Thank you for this observation on the structure of the paper. Following your suggestion, subsections 3.1.1 and 3.1.4 have been merged into one, titled "Global climate: atmosphere and ocean physical fields".**

**As you pointed out, we agree that the two maps (2 m temperature and SST) look similar in some part. However, the detailed discussion on ocean biogeochemical fields relies on the SST map, as discussed between Reviewer #1 and the authors. We believe its inclusion would be informative for the readers and thus we have decided to retain the SST figure in the revised manuscript.**

Scientific questions
- section 3.1.1: For a more complete view on the simulated climate, please specify the strength of the AMOC and the amplitude of ENSO as these two features of the physical system also affect the simulated carbon cycle quite profoundly. Just mention the numbers.

**Following your suggestion, we briefly mentioned the amplitude of both AMOC and**

**ENSO in the revised manuscript without any additional figures as follows:**

*"In addition to the radiation/temperature responses against historical external forcing, we briefly describe here the El Niño–Southern Oscillation (ENSO) and Atlantic meridional overturning circulation (AMOC) strength in MIROC-ES2L, both of which can affect interannual–multidecadal carbon cycle processes (Zickfeld et al., 2008; Pérez et al., 2013; Friedlingstein 2015). In HIST experiment, the standard deviation of monthly SST anomaly in the Niño-3 region (5°S-5°N; 90°–150°W) was 1.57 K in 1950–2009, which is larger than that of HadSST (0.94 K). This unrealistically large ENSO amplitude tends to influence the simulated interannual global temperature variability (Fig. 2b), which is suggestive of further effect on the interannual variability in biogeochemical fields (e.g., $CO_2$ flux in the tropics). The AMOC intensity, quantified by the North Atlantic Deep Water transport across 26.5°N, was approximately 13 Sv (1 Sv = 106 $m^3$ $s^{-1}$) as the 1850–2014 average, which is smaller than the observational estimates of 17.2 Sv (McCarthy et al., 2015). In the HIST run, the AMOC strength was weakened at a rate of 0.01 Sv $yr^{-1}$ (i.e., reduction of 1.7 Sv during the 165 years of HIST), which seems slightly smaller than the recent estimates of AMOC weakening of 3 ± 1 Sv from the mid-twentieth century (Caesar et al., 2018)."*

- line 527-530: Uncertainty in land carbon uptake (estimated from data) is smaller if calculated from the global carbon budgets (following CL = CE – CA – CO, s. line 435) than if it is calculated from the uncertainties in LUC emissions and the natural land sink. The uncertainty in CO is 20 PgC (s. line 550). The uncertainties in CE (derived directly from inventory data) and CA (derived from precise and representative measurements) are even smaller. Thus, the uncertainty in CL is much smaller than 90 PgC.

**Thank you for the important suggestion regarding the range of uncertainty for CL. As you pointed out, if we calculate the possible CL range from the combination of CE (fossil fuel), CA, and CO, the CL range is changed to 15 ± 29 PgC, where the uncertainty range (±29 PgC) is much smaller than that of the bottom-up approach (i.e., the sum of the natural land sink + land use change, i.e., ±90 PgC). Thus, in the revised manuscript, we have mentioned both estimation ranges as follows:**

*"The possible range for CL can be changed if we estimate it as the residual of other global carbon budgets (i.e., CL = FF − CA − CO, where FF is the cumulative fossil fuel carbon emission). Using the estimated ranges of FF, CA, and CO reported by Le Quéré et al. (2018) (i.e., 400 ± 20, 235 ± 5, and 150 ± 20 PgC, respectively; the budget imbalance of 25 PgC is ignored here), the CL range is suggested to be 15 ± 29 PgC. In this case, the result of MIROC-ES2L (44 PgC) is still within the estimation boundaries*

*although it is at the upper end of the suggested range.***"**

- line 531-542: I'm not an expert in ocean biogeochemistry, but as far as I understand, a buildup of the ocean sediment reduces alkalinity in the ocean water, so that the ocean on the long term will outgas CO2 to the atmosphere and ocean water + ocean sediment looses carbon after the sedimentation process has been switched on (and the loss in alkalinity is not compensated by riverine input). By contrast, in the manuscript it is mentioned, that the ocean carbon uptake in the control run is partly explained by the sediment extracting carbon from the ocean bottom.

**Thank you for your inquiry regarding the effect of the sedimentation process on the carbon cycle. As you indicated, $CaCO_3$ burial reduces alkalinity, leading to ocean carbon release. However, organic matter burial decreases DIC, resulting in the ocean carbon uptake. In our model, the ocean absorbed $CO_2$ during the spin-up period because the latter process is dominant.**

- line 766-769: I don't understand, why the different treatment of the vertical SOC profile in the model and WISE30sec explains the large difference in the amount of SOC in the boreal range. I think, that has to do with permafrost. It should be mentioned here, if the model includes freezing in the soil and how this affects SOC.

**We apologize for this confusion. Yes, as you correctly pointed out, the direct reason for the difference between WISE30sec and the model is the frozen carbon within the permafrost region; it is considered in WISE30sec but not in the model. We have clarified this point in the revised text as follows:**

*"This is likely attributable to different treatment of frozen carbon in deeper soils in permafrost regions, i.e., WISE30sec covers the total SOC down to 2 m depth including frozen carbon, while the model does not consider the frozen carbon and instead simulates only upper SOC as litter form and lower SOC as humus."*

- section 3.2.2: The simulations NO-NR, NO-NRD, and NO-FD are only 100 years long. Do you've analyzed, if the signals that are based on these simulations and discussed in section 3.2.2 are already stable after 100 years? Or are they still very transient?

**Thank you for your suggestion. In our model, NPP changes in response to the changes in nutrient input within several decades, consistent with Somes et al. (2016). We have added this information to the revised manuscript as follows.**

*"In the simulations, because changes in NPP and surface nutrient concentrations continued to change over several decades after the abrupt switching-off*

*manipulation, the average over the final 10 years is used for the following analysis. The rapid response of NPP to changes in nutrient input is consistent with that found in previous research (Somes et al., 2016)."*

Minor corrections
- line 18: "article describes" instead of "study developed"
**We have made this change as you suggest.**

- line 176: Figure 1 is not helpful. Please specify for each model component whether it represents atmosphere, ocean, or land. You can also add for each model component the ele-ments that are handled prognostically (C,N for VISIT-e, C,N,P,Fe,Ca,O for OECO-v2?, Fe,S for SPRINTARS?) and indicating by labeled arrows which elements are passed from each component to others (e.g. N,P from VISIT-e to OECO-v2). I think, this would result in a nice overview schematic, how the components are coupled concern-ing biogeochemistry.
**Thank you for this very good suggestion. The schematic has been modified accordingly and we agree that the revised figure is much more powerful and more informative for the readers.**

- line 201: Please indicate the horizontal resolution of the ocean model (e.g. average size of a grid box in km or the number of grid boxes of the global field).
**This has been documented in the revised manuscript as follows:**
*"The horizontal coordination for the ocean is changed from the bipolar system employed in MIROC5 to a tripolar system in MIROC5.2 that is divided horizontally into 360 × 256 grids. (To the south of 63°N, the longitudinal grid spacing is 1° and the meridional spacing becomes fine near the Equator. In the central Arctic Ocean, the grid spacing is finer than 1° because of the tripolar system.)"*

- line 210: What is a "snow-derived wetland"?
**In the revised manuscript, this has been clarified as follows:**
*"…wetland formed temporarily in the snowmelt season is newly considered to reduce…"*

- line 257: "of vegetation (each represented on a separate tile) in each land grid box" instead of "of tile in each land grid"
**Thank you for suggesting this correction. We have changed it as follows in the revised manuscript:**

*"The model assumes five types of land cover (each represented on a separate tile) in each land grid box (i.e., primary vegetation, secondary vegetation, urban, cropland, and pasture)…"*

- line 277: "transported by rivers" instead of "transported rivers"
**Corrected.**

- line 305: The phosphorous cycle has also no analog to denitrification.
**Following your comment, we have rephrased it as follows:**
*"The structure of the phosphorus cycle is generally similar to that of nitrogen except in two respects: 1) the riverine input of phosphate is the only process that introduces phosphorus into the ocean, and 2) there is no process of outgassing from the ocean, unlike the denitrification process in the nitrogen cycle."*

- line 346: It would be nice to mention how the DMS affects the climate (I guess as sulfate aerosol in MATSIRO that affects radiation).
**In response to your comment, we have added the following sentences at the end of this paragraph:**
*"This modification of the DMS emission scheme increases the sulfate aerosol amount, particularly over high-latitude oceans during winter and in regions where strong surface wind speed occurs. Solar irradiance of the surface decreases in such regions; however, this effect is not sufficiently significant to change the mean physical climate states."*

- line 364, 372, 373: I would not use the word "detect" in that way. Please substitute it by e.g. "except that the prescribed CO2 increase affects only the carbon cycle processes".
**Corrected.**

- line 403: last 4 lines of table 1, NO-NRD Configurations "N depositions" instead of "Fe depositions", NO-FD Configurations "Fe depositions" in-stead of "de depositions".
**Corrected.**

- line 438: "coupled" instead of "entire"
**The corresponding sentence has been reworded following the comment from Reviewer #1.**

- line 448,449: The denominator should be T. The common unit of Gamma is PgC/K (s. also table 6).

**Thank you for identifying this error. The equation was originally presented incorrectly and in the revised manuscript, "CA$^{1PPY}$" has been replaced with "T$^{1PPY-RAD}$".**

-line 471: "deviations of the model results from HadCRUT4" instead of "discrepancies between the model result and HadCRUT4"

**Corrected.**

- line 555: "in the HIST run" instead of "at the end of the HIST run"

**We are sorry but we cannot find the corresponding text in the first manuscript. We suppose you meant <"at the end of the HIST run" instead of "in the HIST run">. Based on this idea, we have rephrased L555 as follows:**

**"…this speculation is also supported by the diagnosed CO$_2$ concentration at the end of the HIST run…"**

- line 606: "4.2 TgN yr-1" instead of "4.5 TgN yr-1" - compare with line 11 of table 3

**This has been replaced with the correct number (4.3 TgN yr$^{-1}$) in the revised manuscript.**

- line 644: "decay of biomass in the LUC-product pools" instead of "decay of LUC-product pools"

**Corrected.**

- line 701: "concentration minimum" instead of "con-centration peak"

**Corrected.**

- line 721: Please mention that the model (obviously) simulates no deep water formation in the Labrador Sea.

**Thank you for your suggestion. The problem in relation to the Labrador Sea has been mentioned in the revised manuscript.**

- line 738: "high" instead of "higher"

**Corrected.**

- line 755: "GPP in these regions is captured reasonably well by the model (Fig. 10a and

10b). Thus, the overestimation" instead of "Considering the GPP in these regions is captured reasonably well by the model (Fig. 10a and 10b), the overestimation"

**Corrected.**

- line 760: "products" instead of "product"

**Corrected.**

- line 784: unit of SOC is missing

**The unit for SOC has been added (gC m$^{-2}$).**

- line 875: I see also some regions north of the equator, where GPP is reduced by climate change (e.g. South Asia). Please remove "of the Southern Hemisphere".

**Corrected.**

- line 879-890: Please comment on the strong reduction of GPP by LUC in the tropics.

**Following your comment, we have mentioned the GPP reduction in the tropics as follows:**

**_"In the tropics, LUC reduces the non-crop GPP but weakly increases crop GPP, which results in net negative reduction of GPP as grid averages (Fig. 13j). Meanwhile, regions with intensive agriculture with nitrogen fertilizer input (e.g., Western Europe, East Asia, and parts of North America) show net positive change of GPP as grid averages, where increases in the crop contribution overcome reductions in the non-crop contribution (Fig. 13k and 12l)."_**

- line 917: I think, it would be good to mention that the NPP increase in the open ocean by N input from rivers mainly occurs in the Atlantic.

**We agree with your comment. Significant NPP increase due to riverine N input in the Atlantic has been mentioned in the revised manuscript.**

- line 973: "CO2-induced ocean acidification and warming-induced deoxygenation" instead of "warming-induced ocean acidification and deoxygenation"

**The suggested change has been made.**

- line 1002,1003: This sentence is better placed at the end of the para-graph.

**The suggested change has been made.**

- line 1022,1023: This sentence is just repetition. You can remove it to shorten the article.

**The sentence has been removed from the revised manuscript.**

- line 1026: "climate, carbon cycle, and coupled climate-carbon cycle sys-tem" instead of "climate, carbon cycle, and climate-carbon cycle system"

**The suggested change has been made.**

- line 1130: "confirmed to be captured well" instead of "confirmed captured well"

**The suggested change has been made.**